# Transcriptomics of Hirschsprung disease patient-derived enteric neural crest cells reveals a role for oxidative phosphorylation

Zhixin Li [1,6], Kathy Nga-Chu Lui [1,6], Sin-Ting Lau[1,6], Frank Pui-Ling Lai[1], Peng Li [2,3], Patrick Ho-Yu Chung[1], Kenneth Kak-Yuen Wong [1], Paul Kwong-Hing Tam[1], Maria-Mercedes Garica-Barcelo[1], Chi-Chung Hui [4], Pak Chung Sham[5] & Elly Sau-Wai Ngan [1]✉

Hirschsprung disease is characterized by the absence of enteric neurons caused by the defects of enteric neural crest cells, leading to intestinal obstruction. Here, using induced pluripotent stem cell-based models of Hirschsprung and single-cell transcriptomic analysis, we identify a gene set of 118 genes commonly dysregulated in all patient enteric neural crest cells, and suggest *HDAC1* may be a key regulator of these genes. Furthermore, upregulation of RNA splicing mediators and enhanced alternative splicing events are associated with severe form of Hirschsprung. In particular, the higher inclusion rate of exon 9 in *PTBP1* and the perturbed expression of a PTBP1-target, *PKM*, are significantly enriched in these patient cells, and associated with the defective oxidative phosphorylation and impaired neurogenesis. Hedgehog-induced oxidative phosphorylation significantly enhances the survival and differentiation capacity of patient cells. In sum, we define various factors associated with Hirschsprung pathogenesis and demonstrate the implications of oxidative phosphorylation in enteric neural crest development and HSCR pathogenesis.

Hirschsprung disease (HSCR), the most common congenital enteric neuropathy, is characterized by the absence of nerve cells along a variable length of the colon (short segment, S-HSCR; long segment, L-HSCR; total colonic aganglionosis, TCA). It is caused by the problems of enteric neural crest cells (ENCCs) in growth, migration and/or differentiation, leading to intestinal obstruction.

Glial cell-derived neurotrophic factor and RET tyrosine receptor kinase constitute the key signaling pathways for the development of the enteric nervous system (ENS)[1]. *RET* variants and mutations that are not necessarily deleterious have been found to be associated with most HSCR cases. Specifically, the risk allele "*T*" in *RET* rs2435357 is most frequently associated with S-HSCR[2]. The BACE2/APP/BACE1[3], NRG1/ERBB[4,5], and Semaphorin C/3D[6] signaling pathways have also been identified through genome-wide genetic screens in various sub-groups of HSCR patients. Nevertheless, the underlying causes of phenotypic heterogeneity are still not well-understood.

It is generally believed that the majority of HSCR cases are caused by cumulative actions of multiple genetic variants, each with minor effects. While each individual may carry different constellations of genetic variants, they likely converge on shared pathogenic pathways,

[1]Department of Surgery, Li Ka Shing Faculty of Medicine, The University of Hong Kong, Pokfulam, Hong Kong. [2]Scientific Research Center, The Seventh Affiliated Hospital of Sun Yat-sen University, Shenzhen 518107 Guangdong, People's Republic of China. [3]Guangdong Provincial Key Laboratory of Digestive Cancer Research, The Seventh Affiliated Hospital of Sun Yat-sen University, No. 628 Zhenyuan Road, Shenzhen 518107 Guangdong, People's Republic of China. [4]Program in Developmental & Stem Cell Biology, The Hospital for Sick Children and Department of Molecular Genetics, University of Toronto, Toronto M5G1L7 ON, Canada. [5]Department of Psychiatry, Li Ka Shing Faculty of Medicine, The University of Hong Kong, Pokfulam, Hong Kong. [6]These authors contributed equally: Zhixin Li, Kathy Nga-Chu Lui, Sin-Ting Lau. ✉e-mail: engan@hku.hk

leading to HSCR disease. On the other hand, additional modifiers likely determine disease severity and account for the disease spectrum. Since ENCC samples from patients cannot be collected at or before disease onset, most transcriptome analyses were performed on resected diseased bowels. Thus, these transcriptomic data mainly reflect end-stage disease status, and molecular events occurring prior to disease onset remain unclear. To date, various directed differentiation protocols have been established for the derivation of neural crest (NC) cells from human induced pluripotent stem cells (iPSC) and to model ENS development and HSCR disease states[3,7–10]. More importantly, subsequent scRNA-seq analysis further revealed that these iPSC-derived NC cells and their neuronal derivatives highly resemble the developing ENS in E13.5 mouse embryonic guts[8]. It opens new opportunities for understanding this complex congenital disease. The use of a single-cell approach exploits cell heterogeneity and provides insight into the involved cellular changes of gene expression across the disease severity axis, allowing exploration of the molecular pathways, genes, and gene regulatory networks involved in specific disease states and cellular phenotypes.

Increasing evidence suggests that metabolic rewiring plays an active role in the regulation of migration and development of neural crest cells (NCCs)[11,12]. Hedgehog-mediated metabolic reset has been observed in cancers, muscle, and adipose tissue, and during stem cell renewal and cell fate transition. It is achieved mainly through the activation of GLI-independent non-canonical Hedgehog signaling, which can rapidly activate the machineries of primary energy metabolism through the activation of AMP kinase (AMPK)[13]. The non-canonical pathway driving the metabolic rewiring is SMO- and cilium-dependent, and that can be uncoupled with the canonical Hedgehog signaling cascades at the level of SMO by selective partial agonism with a low dosage of the Hedgehog antagonist (such as cyclopamine) that allows SMO translocation to the cilium. Hedgehog signaling has been implicated in various stages of ENS development[14–19]. Nevertheless, till now, how the non-canonical arm of the Hedgehog pathway influences ENS development remains largely unknown.

In the present study, we established seven iPSC lines from HSCR patients demonstrating the full disease spectrum and recapitulating HSCR-associated phenotypes in vitro. We then performed a high-resolution transcriptomic analysis of control- and HSCR-ENCCs to identify major changes in molecular pathways and networks associated with disease severity. Defective oxidative phosphorylation (OXPHOS) was found to be highly associated with neuronal differentiation defects of HSCR-ENCCs. Activation of the fatty acid oxidation (FAO) pathway and Hedgehog-induced metabolic rewiring restored the differentiation capacity of HSCR-ENCCs, highlighting the relevance of OXPHOS in HSCR pathogenesis.

## Results

### Establishment and characterization of an iPSC-based model of HSCR

Seven patients with HSCR from different subgroups, including four S-HSCR, two L-HSCR, and one TCA, were recruited for this study. Each patient was given a disease score according to the extent of the aganglionosis (0–5; control = 0 and TCA = 5). The *RET* alleles in rs2435357 and variants in the coding region of *RET* gene were identified by sequencing the genomic DNA from the patient's blood. The clinical and genetic features of each patient are summarized in Fig. 1a, b. Briefly, among the four S-HSCR patients, HSCR#5, #10, and #20 showed relatively mild phenotypes with disease scores of 1 or 2. In contrast, HSCR#1 demonstrated a longer aganglionic segment up to the lower sigmoid, representing the intermediate phenotype between S- and L-HSCR.

Skin fibroblasts from these HSCR patients were used to generate iPSC lines and the corresponding genotype of *RET* alleles in each hiPSC line was confirmed by Sanger sequencing. Two control iPSC lines

which carry the non-risk allele "C/C" in rs2435357 were also included in this study. NCCs carrying the exact genetic makeup of the patients were derived from iPSC lines using an in vitro differentiation protocol[7] (Fig. 1c). The migration and neuronal differentiation capabilities of the FACS-enriched control- and HSCR-NCCs were assessed using scratch (Fig. 1d) and in vitro differentiation (Fig. 1f) assays, respectively, as previously described in refs. 7, 8, 19. The control- and HSCR-iPSC lines generated comparable yields of NCCs (Supplementary Fig. 1a, b). However, the HSCR-NCCs exhibited lower expression levels of *RET* (Supplementary Fig. 1c) and were less competent to migrate, and generate neurons than those of the control NCs (Fig. 1d–g). Interestingly, among these S-HSCR-NCCs, HSCR#1-NCCs exhibited the most severe defects in migration (Fig. 1e), which may have contributed to their higher disease score. Furthermore, the neuronal differentiation defects of the L- and TCA-HSCR NCs were found to be more severe than those of the S-HSCR NCs, as observed in the formation of neuron-like processes and the expression levels of pan-neuronal markers (neurofilament [NF] and protein gene product 9.5 [PGP9.5]) (Fig. 1g). These data showed that the disease phenotypes and severities are highly associated with the combined effects of migration and differentiation defects of iPSC-derived NCs.

We then sequenced the control and HSCR-NCCs at high resolutions using Smart-Seq. A total of 3342 individual cells were sequenced, and 6344 average genes, as well as 770,870 mean confidently mapped reads per cell were identified (Supplementary Data 1 and Supplementary Fig. 2a–c). We projected all single cells on a *t*-distributed stochastic neighbor embedding (*t*-SNE) plot (Fig. 1h) and identified five transcriptionally distinct clusters. Cluster 1 was identified to be the major cluster showing an expression profile highly resembles the major population of ENS cells identified in E12.5 mouse embryos, which express glial and progenitor marker genes, such as *Sox10*, *Erbb3*, and *Plp1*, without any neuronal gene expression[20] (Fig. 1i, j). We also compared this major cluster (Cluster 1) of our iPSC-derived cells to mouse NCCs on enteric and non-enteric lineages at E13.5 obtained from three independent datasets. They included our own dataset where two subpopulations of vagal NCCs: mouse enteric (mENCCs)[19] and cardiac NCCs (mCNCCs), were sequenced at single cell level using 10 × Genomics; and another set of published scRNA-seq sequence data of mouse NCCs on enteric (mENCCs) and peripheral nervous system (mPNS) lineages derived from sciRNA-seq3 platform[21]. All NCCs from different platforms and lineages were integrated together by Seurat software[22] to remove technical and batch effects. As demonstrated by the correlation analysis and principal component analysis (PCA), the iPSC-derived cells showed the highest similarity to the mouse ENCCs when compared to the NCCs on other lineages (Fig. 1k and Supplementary Fig. 2d). Additional lineage comparisons using the published scRNA-seq datasets of mouse ENCCs[20,23] confirmed the iPSC-derived cells highly resembled the bipotent progenitors (BP), but not the committed glial progenitors (GP) nor the neuronal progenitors (NP) (Supplementary Fig. 2e, f). In addition, both control and patient iPSC-derived cells expressed various *HOX* genes (*HOXB2*, *HOX2B3*, *HOXB4*, and *HOXB5*) resembling the vagal/enteric NCCs (Supplementary Fig. 2g). More importantly, the iPSC-derived-NCCs could give rise to neurons with molecular signatures highly comparable to enteric, but not with the sympathetic nor sensory neurons in the peripheral nervous system (PNS)[24] (Supplementary Fig. 2h, i). Therefore, hereafter, the NCCs derived from the iPSC lines will be termed ENCCs. RNA velocity was proven to be a powerful method to predict cell state based on unspliced and spliced mRNAs in single cell[25]. Based on the RNA velocity analysis, Cluster 1 was predicted as the progenitor cells which had great differentiation potential towards clusters 2-5 (Supplementary Fig. 2j), further suggesting that the majority of iPSC-derived cells are at the progenitor state. It is also noteworthy that the five clusters demonstrated different, uneven proportions of cells in the control and HSCR samples, reflecting the heterogeneity between samples

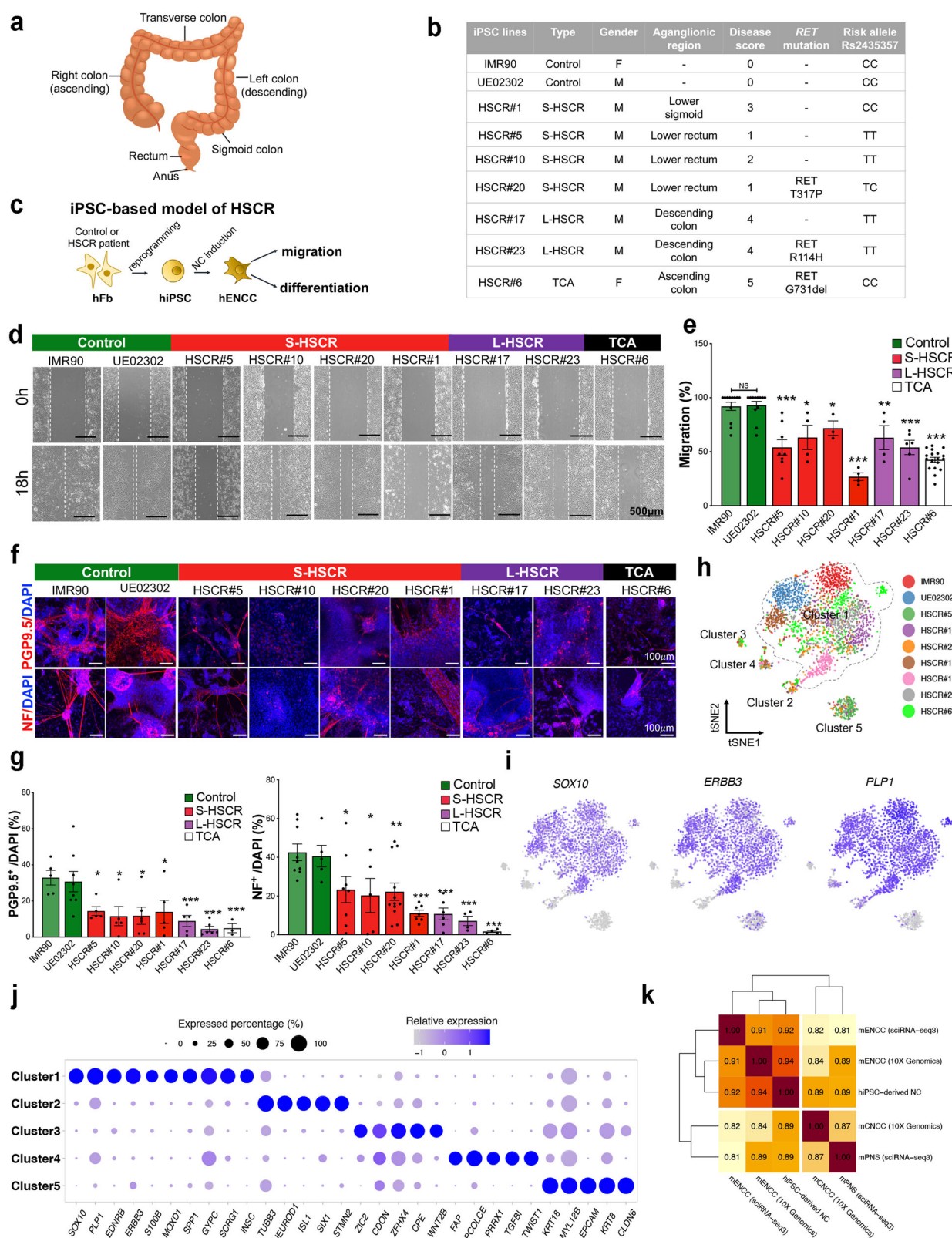

(Supplementary Fig. 2k) and that may also be associated with the disease phenotypes.

## Stratification of HSCR patients and reconstruction of the disease severity axis based on single-cell transcriptomes

Cluster 1 represents the major population of cells found in all iPSC-derived ENCCs and highly resembled the major progenitor population of ENS as identified in E12.5 and E13.5 mouse embryos, so cells in Cluster 1 were used for the subsequent single-cell transcriptome analyses. We first examined whether the transcriptome profiles of HSCR-ENCCs are associated with the disease severity. Genes that were uniquely expressed in one sample were excluded for downstream analysis to remove individual effects. A cladogram was constructed to estimate the overall transcriptional similarities between cells in Cluster

**Fig. 1 | Functional and molecular characterization of an iPSC-based model of HSCR. a** Anatomy of the human colon. **b** An overview of healthy and HSCR-iPSC lines used in this study. **c** Schematic shows the recapitulation of disease phenotypes using HSCR-iPSC lines. **d** In vitro scratch assay. **e** Bar chart shows the quantitative data of migration. Data were presented as mean values ± SEM from 3–18 independent experiments. **f** In vitro differentiation assay with two neuronal markers (PGP9.5 and NF) and **g** the corresponding quantitative analyses. Data are presented as mean values ± SEM from 3–12 independent experiments. **e, g** One-way ANOVA was used for the statistical analyses. Bars marked by "***", "**", and "*" represent they are statistically different from the controls of *P* values <0.001, <0.01, and <0.05,

respectively. NS not significantly different. **h** *t*-SNE projection of all 3342 individual cells based on the expression of markers of shared clusters, colored by iPSC lines. Five main clusters are labeled and Cluster 1 is marked with a dotted line. **i** Canonical markers expressed in Cluster 1. **j** Dot-plot shows top key markers expressed in each cluster. The color of the dot indicates the relative expression and the size of the dot indicates the expressed percentage. **k** Correlation heatmap shows the similarities between human iPSC-derived NC cells and mouse in vivo enteric/non-enteric NC cells at E13.5 from two independent datasets. Sequencing platforms of datasets are noted. Source data are provided as a Source Data file.

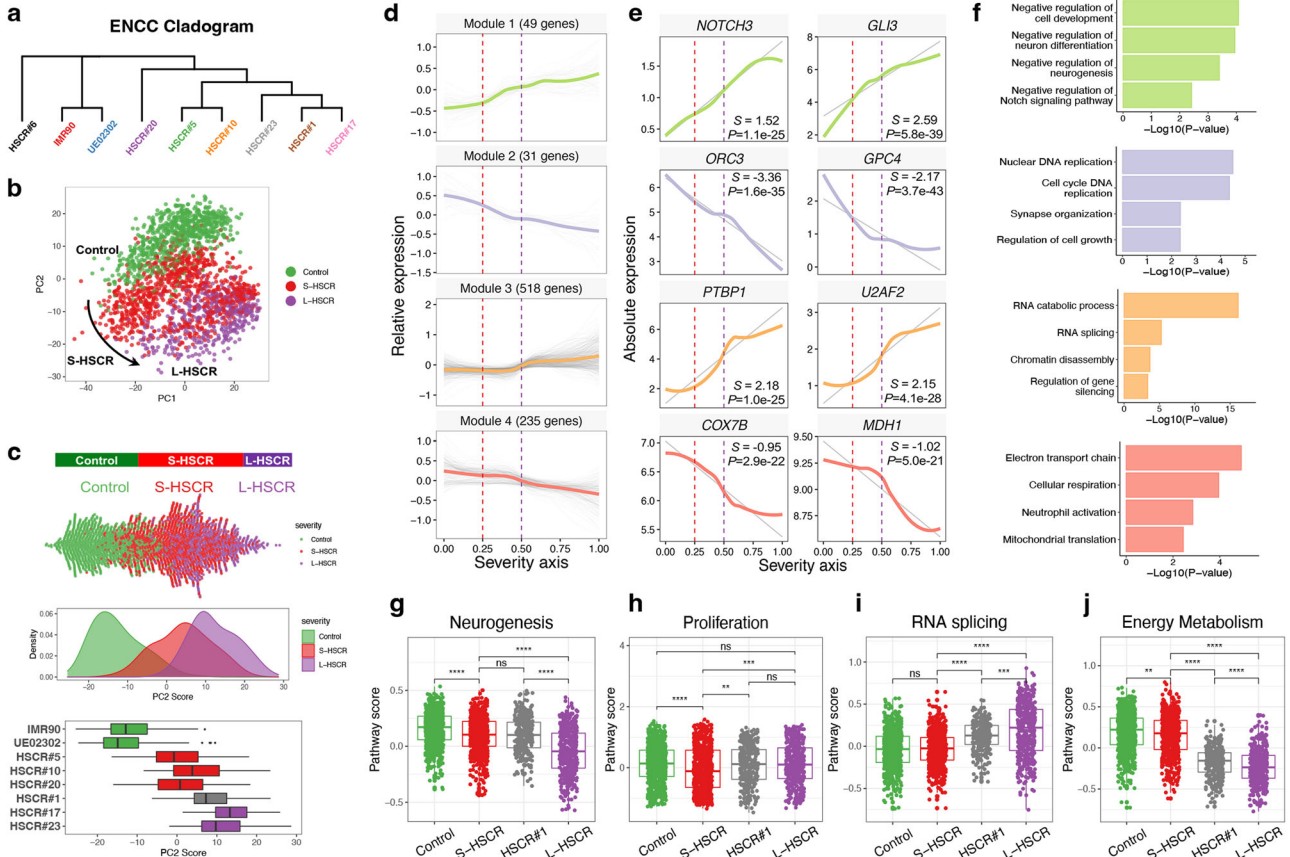

**Fig. 2 | Reconstruction of disease severity axis integrating different HSCR states. a** Hierarchical clustering by average gene expression. **b** PCA projection of cells in Cluster 1 based on gene expression profile after excluding the TCA-HSCR-iPSC-line (HSCR#6). **c** Distribution of cells along PC2 is highly associated with disease severity, as shown in beeswarm, density, and boxplots. Colored by type of HSCR. 379 IMR90 cells, 375 UE02302 cells, 161 HSCR#5 cells, 296 HSCR#10 cells, 186 HSCR#20 cells, 306 HSCR#1 cells, 187 HSCR#17 cells, and 246 HSCR#23 cells were included. **d** The expression dynamics of 833 top DEGs were cataloged into four major modules, colored by modules. Thick lines indicate the average gene expression patterns of each module. All 2136 cells ordered by severity axis were included to fit the expression curve. **e** Gene signatures and expression dynamics of representative genes in each gene module. The relative expression levels of these genes are shown as a LOESS smooth fit line (colored line) and the best-fit lines (gray

lines) in linear regression. *S* slope, *P P* value of linear regression. **f** Gene ontology analyses of each gene module. *P* value and FDR were calculated by clusterProfiler software. **g–j** Overall pathway scores of neurogenesis, proliferation, RNA splicing, and energy metabolism. A two-sided Wilcoxon rank-sum test was applied to calculate the significance of the difference between the two groups. All samples (two controls with 754 cells, three S-HSCR with 643 cells, intermediated HSCR#1 with 306 cells, and two L-HSCR with 433 cells) were included for comparison and presented as groups. Tests marked by "****", "***", "**", and "*" represent they are statistically different from the controls of *P* values <0.0001, <0.001, <0.01, and <0.05, respectively. ns not significantly different. In all the boxplots, each box represents the interquartile range (IQR, the range between the 25th and 75th percentile) with the mid-point of the data, whiskers indicate the upper and lower value within 1.5 times the IQR.

1 for all samples. Samples with similar clinical severity scores tended to be clustered together (Fig. 2a). HSCR#6 was located at a unique branch in the cladogram, suggesting that the pathogenic pathways involved in TCA are different from those in S- and L-HSCR. Although HSCR#1 was classified as S-HSCR based on clinical stratification, it showed similarities with the L-HSCR specimens (HSCR#17 and #23). Next, we compared each HSCR-ENCCs with those from the two controls and genes in HSCR-ENCCs significantly different from that of the cells from both of

these two control lines were considered as true DEGs. At a false discovery rate (FDR) of 1%, we identified 879, 909, 2,301, 3,635, 6,445, 1,790, and 818 differentially expressed genes (DEGs) in HSCR#5, #10, #20, #1, #17, #23, and #6, respectively. Except for those in HSCR#6, which was derived from a patient with TCA carrying a deletion mutation in the *RET* gene (RET G731del), the ENCCs with higher clinical scores demonstrated significantly higher numbers of DEGs in both threshold-free (Supplementary Fig. 2l) and fixed-threshold DEG

analysis and comparison ($R^2 = 0.464$, $P < 0.01$) (Supplementary Fig. 2m). TCA-ENCCs exhibited unique transcription profiles with enriched molecular pathways associated with GDNF/RET signaling as described previously in ref. 7 and in Supplementary Fig. 3. To identify the RET-independent pathways implicated in HSCR pathogenesis, we excluded HSCR#6 from subsequent analyses.

Since the S- and L-HSCR patients show different extents of aganglionosis, we then determined whether there is a pseudo-axis that can reflect the clinical phenotypic heterogeneity and/or the extent of aganglionosis based on single-cell transcriptomes from patients and control iPSC-derived ENCCs. To this end, we employed PCA to re-cluster the control- and HSCR- ENCCs. A disease severity axis was observed along the PC2 axis, with the ENCCs distributed according to the extent of aganglionosis (Fig. 2b). In particular, S-HSCR#1, which had a relatively longer aganglionic segment, ENCCs derived from this iPSC-line fell between the S-HSCR- and L-HSCR-ENCCs in the PC2 axis (Fig. 2c). Therefore, the stratification of disease severity based on single-cell profiling of iPSC-derived ENCCs was consistent with clinical stratification. Furthermore, the case-control divergence in this component may reflect cells at varying severity states along a shared disease process. The ENCCs of S-HSCR#5, #10, and #20 were more similar to the control group, demonstrating they were less severe than the S-HSCR#1 ENCCs which was at the intermediate state, and the L-HSCR-ENCCs (Fig. 2c).

To further delineate the genes implicated in the disease severity, all 2136 iPSC-derived cells from two control and six S/L-HSCR patient lines were included to perform the differential expression analysis using Monocle[26], identifying 833 genes as the significant DEGs ($q$ value < 0.01) along the disease severity axis. When we examined their expression dynamics along this pseudo-disease severity axis, we observed four major categories of transcriptional gene modules in characterized patterns (Fig. 2d–f and Supplementary Data 2). As revealed by over-representation analysis (ORA), module 1 consisted of genes that are upregulated in HSCR-ENCCs with low disease severity, and are largely involved in "negative regulation of neuron differentiation and Notch signaling," such as NOTCH3 and GLI3 (Fig. 2f). Module 2 consisted of genes that show rapid downregulation and are enriched in "cell cycle and synapse organization," such as ORC3 and GPC4. On the other hand, module 3 consisted of genes that are activated at the higher disease severity state, during cell shift from moderate S-HSCR to L-HSCR, and are predominantly "RNA splicing" genes, such as PTBP1 and U2AF2. Lastly, module 4 consisted of genes that are downregulated at the severe disease state and enriched in energy metabolism, such as COX7B and MDH1. Each disease-associated module was enriched with specific biological processes, suggesting that distinct cellular events are interrupted at different disease stages, accounting for the varying disease severities. In particular, perturbation of neurogenesis, proliferation, RNA splicing, and energy metabolism of ENCCs were found to be the key events associated with the disease severity of HSCR.

To complement the ORA analysis, the expression of whole pathways was further scored using an additive model. Genes involved in neurogenesis (GO:0050771), proliferation (GO:0051726), RNA splicing (GO:0033120), and cellular respiration (GO:1901857) in the GO database were used to estimate the overall changes of the key biological processes. From these analyses, we found genes implicated in neurogenesis (Fig. 2g) and energy metabolism genes (Fig. 2j) were mildly repressed in the S-HSCR-ENCCs, but severely downregulated in L-HSCR-ENCCs, leading to low pathway scores in these two categories. Genes implicated in cell proliferation (Fig. 2h) and RNA splicing (Fig. 2i) were also uniquely repressed and activated in S-HSCR and L-HSCR-ENCCs, respectively. HSCR#1, representing the intermediate case, exhibited interrupted RNA splicing and energy metabolism, while the neurogenesis pathway score was similar to that of S-HSCR-ENCCs. The expression levels of representative genes in these cellular processes

are shown in Supplementary Fig. 4. Together, our results revealed that the enhanced RNA splicing accompanied by reduced energy metabolism are associated with a severe form of HSCR.

## Identification of HDAC1 as a key regulator across the HSCR severity axis

We next aimed to determine the pathway(s) that are commonly affected across various patient bases and identify the corresponding "potential driver genes". A total of 118 DEGs are shared between the S- and L-HSCR groups, including 49 downregulated and 69 upregulated genes. We reasoned that the potential driver genes are likely changed along with the disease state and contribute to the higher disease state. Thus, we built a statistical model to assess how these 118 DEGs change along the disease severity axis. From this analysis, we identified 20 genes that were significantly perturbed along the disease severity axis (DEG across PC2, FDR <0.05) and functionally enriched in biological pathways. We then used a local linear regression model to identify the rate of expression changes of these core genes across the severity axis. We found that 14 and 6 core genes were consistently upregulated and downregulated, respectively, in HSCR (Fig. 3a). Based on the STRING database[27], these genes encode proteins that show strong interactions and involvement in neurogenesis and NC development (Fig. 3b). Specifically, HDAC1 was identified as a hub gene in the PPI network that bridging multiple crucial biological processes. Since most of the core genes were upregulated in the HSCR-ENCCs and showed similar expression patterns with HDAC1 (Fig. 3d), we performed motif enrichment analysis to explore the potential transcriptional coactivator effects of HDAC1 on their expression. We found that 13 of the 20 core genes had at least two sites enriched with the HDAC1 DNA-binding motif (Supplementary Fig. 5). Furthermore, 6 of them were predicted to be "high-confidence" direct target genes of HDAC1 (Fig. 3c). Particularly, we deliberately used the non-DEGs as background genes to perform HDAC1 motif enrichment analysis and observed significantly more enrichment in the DEGs compared to the non-DEGs (OR = 3.41, $\chi^2$ $P$ value = 1.60E-04). This suggested the binding of HDAC1 to the human genome was neither universal nor random, and it was associated with the disease state. Using single-cell regulatory network inference and clustering (SCENIC) analysis[28], we established a gene regulatory network (GRN) of HDAC1. HDAC1 exhibited transcript enrichment (orange) paired with concomitant activation of downstream regulatory networks (blue) in various disease states (Fig. 3e). Intriguingly, the HDAC1 GRN was slightly activated in cells of low severity state (S-HSCR) but highly activated in severe disease state (L-HSCR) (Fig. 3e, boxplot). This implies that the upregulation of HDAC1 and activation of its GRN are highly associated with disease severity, highlighting its role in different forms of HSCR.

A whole-transcriptome correlation analysis was performed to identify HDAC1-mediated (activate or repress) genes and pathways. We first evaluated the correlation (co-expression) between HDAC1 and the other genes in control- and HSCR-ENCCs to identify significant alterations in the regulatory architecture of HDAC1 among HSCR-ENCCs. To identify potential direct binding genes of HDAC1, we then performed a motif enrichment analysis on the differentially expressed HDAC1-associated genes between the control- and HSCR-ENCCs. From these analyses, we predicted 5336 and 512 genes to be the HDAC1-activated and repressed genes, respectively, where the activated genes represent the direct targets of the HDAC1 with binding motif(s) in their promoter regions (Fig. 3f). We found that the putative HDAC1-activated genes are enriched in "RNA splicing and protein localization to nucleus" pathways, while the repressed target genes are enriched in in "action potential, regulation of membrane and pH", likely implicated in regulating the ion channels of the cellular membrane (Fig. 3g).

The transcriptomic data suggest that the elevated transactivation activity of HDAC1 would be an underlying cause of HSCR. As revealed

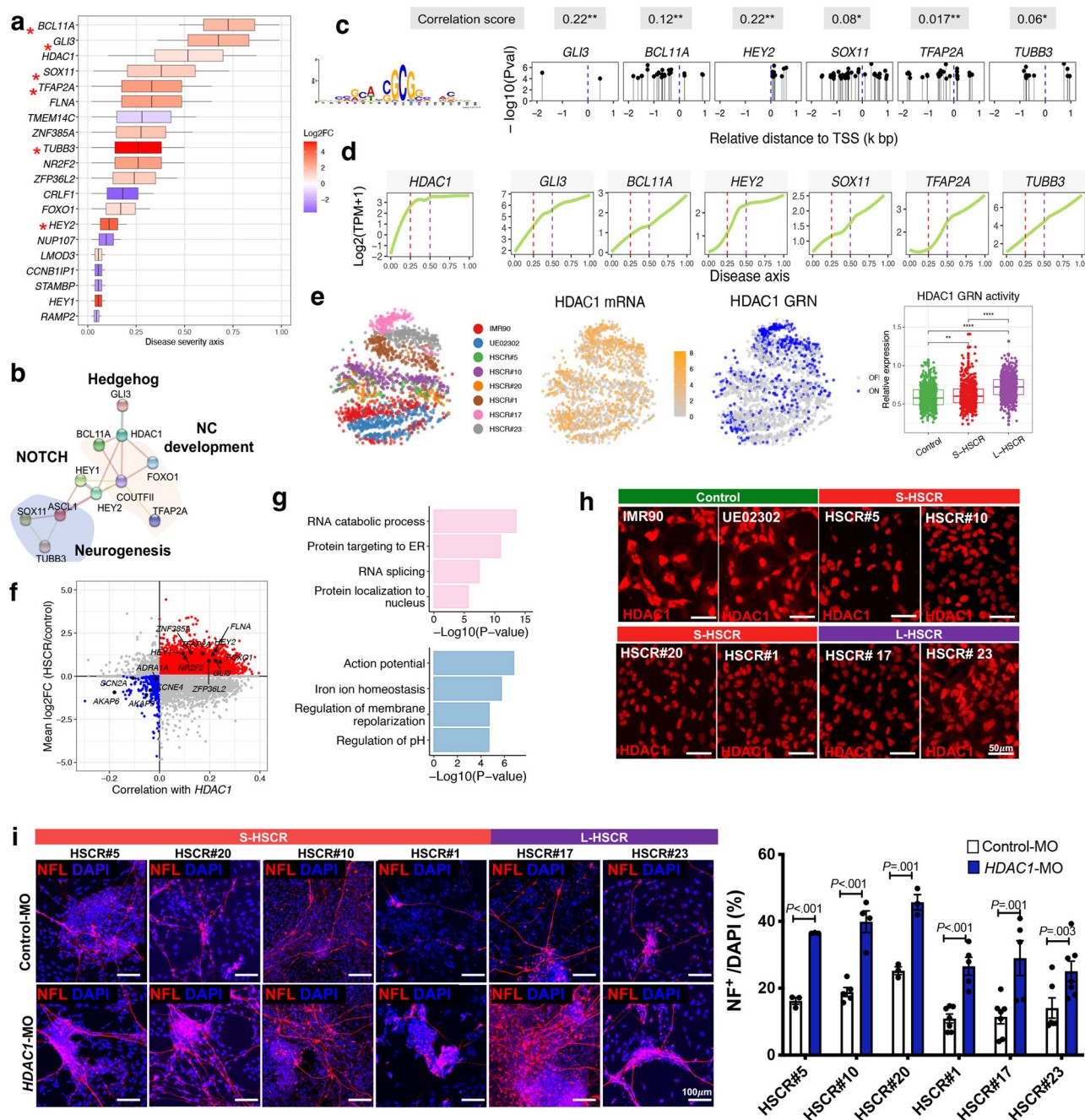

by the scRNA-seq data, the *HDAC1* transcript levels were elevated in ENCCs derived from two S-HSCR (HSCR#20, #1) and two L-HSCR (HSCR #17 & #27) lines (Supplementary Fig. 6a). Therefore, we performed additional biological assays to delineate how HDAC1 may influence the development of hENCCs and determine how the aberrant HDAC1 activity may cause HSCR disease. Western blot analysis revealed that the total protein expression of HDAC1 in control- and all HSCR-ENCCs are highly comparable (Supplementary Fig. 6b). We also examined the repressor function of HDAC1 based on its ability to catalyze the removal of acetyl groups from core histones (H3). As shown in Supplementary Fig. 6b, no obvious deacetylation of the H3 was found in HSCR-ENCCs based on the levels of H3K27Ac (Supplementary Fig. 6b). HDAC1 forms Sin3, NuRD, and CoREST complexes to repress gene expression through promoting histone deacetylation. Therefore, we also examined the potential repressor function of HDAC1 based on the combined expression levels of various subunits in the Sin3, NuRD, and CoREST complexes (Supplementary Table 2), the

inferred Sin3, NuRD, and CoREST complex activities were highly comparable between control and HSCR-ENCCs (Supplementary Fig. 6c). All these data imply that the aberrant HDAC1 activity as seen in HSCR-ENCCs is unlikely due to the enhanced expression of HDAC1 itself nor its repressor function. On the other hand, expression of HDAC1 was mainly found in the nuclei of HSCR-ENCCs, while both cytoplasmic and nuclear expression of HDAC1 were observed in control-ENCCs (Fig. 3h and Supplementary Fig. 6d). Given the overall expression levels of HDAC1 was comparable between the control- and HSCR-ENCCs, the enhanced nuclear localization of HDAC1 may increase the transactivation activity of HDAC1 in HSCR-ENCCs, leading to the aberrant expressions of HDAC1 target genes. To directly demonstrate the impact of the elevated HDAC1 activity in the neuronal differentiation defects of HSCR-ENCCs, we performed the in vitro differentiation assays using HSCR-ENCCs with specific knock-down of HDAC1, where the HSCR-ENCCs were directed to neuronal lineage to generate enteric neuronal progenitors (NPs) in the presence of

**Fig. 3 | Identification of a functionally enriched gene set and HDAC1 as a key regulator along HSCR severity axis. a** Anchoring of the functionally enriched gene set to severity axis with 2136 cells, colored by log₂ fold-change (Log2FC) and HDAC1 targets are marked by asterisks. Log2FC was calculated by comparing all HSCR cases to two controls. Box indicates the confidence interval of activated (red) or repressed (blue) 'period' at the severity axis for each gene. **b** Protein–protein interaction (PPI) network of the core gene set. NC, neural crest. **c** Motif enrichment of HDAC1 at the promoter region (−2 to 1 kb relative to TSS) of its target genes. HDAC1 DNA-binding motif is shown in the left panel. TSS transcription start site. *P* values ($P_{val}$) were calculated by meme FIMO software. **d** Dynamic expression of HDAC1 and its activated target genes along the disease severity axis. Gene expression (log₂(TPM + 1)) was used to fit the smooth line using all 2136 cells. TPM, Transcript per million. Correlations between HDAC1 and its target genes are shown at the bottom panel (calculated by the R cor function). The significance of the correlation testing are marked by an asterisk. *P* values <0.05 and <0.001 are marked by * and **. **e** t-SNE projection of cells based on GRN analysis. mRNA (yellow) and binary regulon activity (active (blue) or inactive (gray)) inferred by SCENIC are shown. The overall activities of HDAC1 targeted GRN in control and S/L-HSCR are shown in the boxplot. t-SNE was generated based on overall regulon activities from the SCENIC analysis. A two-sided Wilcoxon rank-sum test was applied to calculate the significance of the difference between the two groups. All samples (two controls with 754 cells, four S-HSCR with 949 cells, and two L-HSCR with 433 cells) were included for comparison and presented as groups. Boxplot represents the interquartile range (showing median, 25th and 75th percentile, and 1.5. the interquartile range). Tests marked by "****" and "**" represent they are statistically different from the controls of *P* values <0.0001 and <0.01, respectively. ns not significantly different. In all the boxplots, each box represents the interquartile range (IQR, the range between the 25th and 75th percentile) with the mid-point of the data, whiskers indicate the upper and lower value within 1.5 times the IQR. **f** Scatterplot shows putative HDAC1-activated (red) and repressed (blue) targeted genes. Key genes are labeled. The distribution of the correlation between HDAC1 and other genes is shown at the top panel. **g** Gene ontology (GO) enrichment analyses of HDAC1-activated and repressed targeted genes. *P* value and FDR were calculated by clusterProfiler software. **h** Immunocytochemistry shows the reduced cytoplasmic HDAC1 in HSCR-ENCCs. **i** In vitro differentiation assays in the absence or presence of *HDAC1* morpholino (*HDAC1*-MO, 5 μM). Cells were harvested for immunofluorescence staining on day 5 of neuronal differentiation. Data were presented as mean values ± SEM from 3–7 independent experiments. (Student's *t*-test, two-sided). Source data are provided as a Source Data file.

Control-MO or *HDAC1*-MO. Consistently, downregulation of HDAC1 expression level by the addition of morpholinos (*HDAC1*-MO) (Supplementary Fig. 6e), favored the neuronal lineage differentiation of all HSCR-ENCCs, and significantly more NF-expressing NPs were found at the day 5 of differentiation (Fig. 3i). Similar observations were found when HDAC1 was blocked by two different HDAC inhibitors (pan-HDAC inhibitor: VPA and HDAC1 class I inhibitor: CI994). (Supplementary Fig. 6f, g). In summary, aberrantly high HDAC1 transactivation activity is likely associated with the neuronal differentiation defect as observed in HSCR-ENCCs.

## Global analysis of splicing among different HSCR-ENCCs

Alterations in RNA splicing pathways were found to be highly associated with advanced disease states in HSCR (Fig. 2i). Consistent with this, we identified 37 RNA-splicing mediators, including *PTBP1*, *SRSF2*, *U2AF1*, and *U2AF2*, specifically upregulated in severe HSCR cases (Fig. 4a). Subsequent global alternative splicing (AS) analysis also revealed that the sequencing reads in S-HSCR#1 and L-HSCR-ENCCs show significantly higher splicing frequencies (determined by the number of splices divided by uniquely mapped reads number in each cell) than the control-ENCCs (Fig. 4b). The transcript events and local AS events were generated according to GRCh38 genome annotation profile. Further, the percentage of transcripts with sequences spliced in (percent spliced in [PSI]) was estimated for cassette exons, alternative 5′/3′ splice sites, and various AS events by SUPPA2[29]. The most common AS events identified were cassette inclusion and alternative first exons, accounting for more than 60% of the total AS events (Fig. 4c). Except for intron retention, all the other six AS types had significantly higher numbers of events in ENCCs at high disease state (HSCR#1, 17 and 23) than those in low disease states (two controls and three S-HSCR-ENCCs) (*P* < 0.05, *t*-test). Additionally, the overall expression of the 37 RNA-splicing mediators significantly and positively correlated with AS frequency (Fig. 4d). PCA analysis was performed based on the PSI profile of all AS events identified. L-HSCR-ENCCs exhibited distinct AS profiles, with a unique cluster in the PCA (Fig. 4e). This suggested the significant post-transcriptional alteration on RNA splicing in L-HSCR_ENCCs, compared to S- and control-ENCCs. In total, 4735 AS events occurred at significantly higher rates in the L-HSCR-ENCCs than in control-ENCCs, affecting 2063 genes (FDR <0.05). The core AS genes that could contribute (determined by the weight of AS events along specific PC) to the advanced disease state (PC2) are shown in Fig. 4f and many of them are involved in neuronal differentiation and cell metabolism. Polypyrimidine Tract Binding Protein 1 (*PTBP1*) showed the highest association with L-HSCR and were present in multiple AS forms resulting from the inclusion of exon

9 (*PTBP1 [1]*) as well as alternative first exons *PTBP1 [2]* and *[4]*) and retention of the intron (*PTBP1 [3]*) (Fig. 4g and Supplementary Data 3). Different isoforms of PTBP1 have been implicated in neurogenesis[30,31] or glycolysis[32]. In addition, we found that many of the other top differentially spliced genes are the putative targets of PTBP1 [1] (inclusion of exon 9) (Fig. 4f, highlighted in red). For instance, Adenylate Kinase 3 (*AK3*) encodes a GTP:ATP phosphotransferase, which is involved in cell energy metabolism. Ubiquitin Specific Peptidase 14 (USP14) is a deubiquitinating enzyme and is associated with neurodegeneration. The inclusion of exon 4 and exon 3 in *AK3* and *USP14* were found to be significantly enriched in all HSCR and L-HSCR, respectively.

We then further established a multi-layer model to evaluate the clinical outcome of HSCR using the PCs described in Figs. 2c, 4f, which represent the key changes at transcriptional (gene expression) and post-transcriptional (RNA splicing) levels of the cells, respectively. The combination of transcriptional and post-transcriptional profiles explains the measured clinical scores of the HSCR cases, providing a possible picture of the molecular constituents and mechanisms underlying the heterogeneity of HSCR (Fig. 4h).

## Inclusion of *PTBP1* exon 9 was associated with the disrupted cellular metabolism in L-HSCR-ENCCs

Among all AS events, the inclusion of *PTBP1* exon 9 appears to contribute most significantly to the advanced disease state; *PTBP1* was overexpressed in the HSCR, particularly higher in the severe disease cases, with strong enrichment of exon 9 (Fig. 5a and Supplementary Fig. 7). Exon 9 in *PTBP1* encodes a linker between RNA recognition motif 2 (RRM2) and RRM3 in *PTBP1*. It also possesses splicing regulatory activity that are distinct from its RNA-binding activity[31]. Therefore, we further analyzed the relationship between *PTBP1* exon 9 inclusion and other AS events found in HSCR-ENCCs. First, AS events significantly correlated with *PTBP1* exon 9 were selected. Then, the exons targeted by PTBP1 through motif recognition were used for further identification of HSCR-associated AS events. From this, ~2000 and 1000 AS events were predicted to be enhanced and repressed, respectively, by *PTBP1* exon 9 in HSCR-ENCCs (Fig. 5b). Remarkable overlaps were found between our predicted *PTBP1* exon 9 targets and public data[31] (Fig. 5c). Target genes with activated or repressed exons were used to performed gene set enrichment analysis. Predictive scores reflected the strength and direction of regulation were calculated for enriched pathways based on the integrative correlation coefficient (additive model). Overall, the *PTBP1* exon 9-repressed exons represented genes that are related to "metabolic processes" and "biosynthetic process" pathways, while the genes with *PTBP1* exon 9-activated

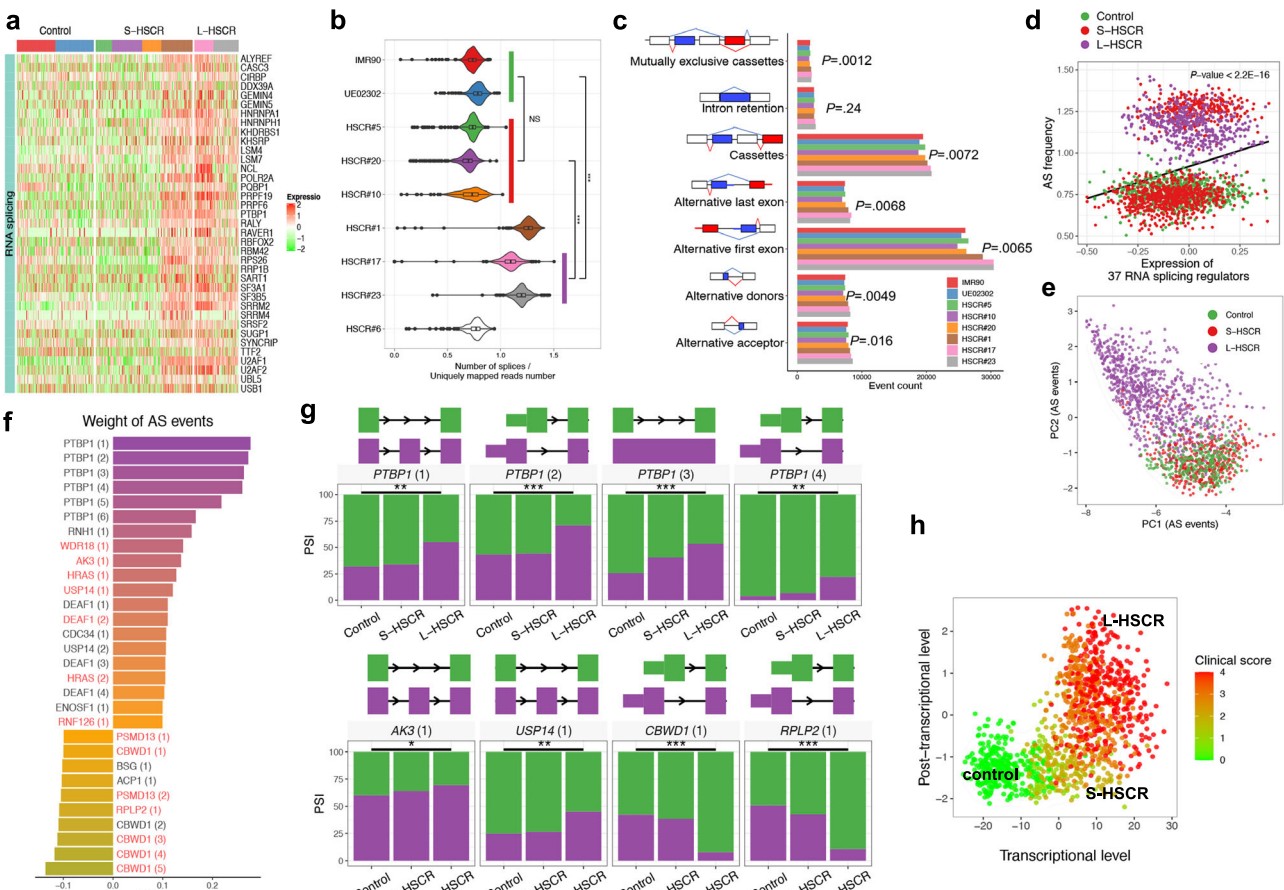

**Fig. 4 | Global analysis of RNA alternative splicing (AS) events in control- and HSCR-ENCCs. a** Heatmap shows the top 37 upregulated AS mediator genes enriched in HSCR#1 and L-HSCR. **b** Normalized number of splices in control- and HSCR-ENCCs. *P* values (two-sided Mann–Whitney *U*-test) from comparing ENCCs derived from the two controls with 754 cells and various HSCR groups (three S-HSCR with 643 cells and two L-HSCR with 433 cells) are shown. The splicing frequency of each cell was estimated by the number of splices divided by the uniquely mapped read number. Tests marked by "***" represent they are statistically different from the controls of *P* values <0.001. ns not significantly different. In the violin plot, each violin represents the kernel probability density of the data at different values. An inner boxplot indicates the interquartile range (IQR, the range between the 25th and 75th percentile) with the mid-point of the data. **c** Number of different types of RNA splicing events in control- and HSCR-ENCCs. ENCCs at high disease states (HSCR#1, 17 and 23) were compared to those in low disease states (two controls and three S-HSCR-ENCCs) (*P* < 0.05, two-sided *t*-test). **d** The 37 RNA-splicing regulator

genes are significantly upregulated and positively correlated with AS frequency in HSCR#1- and two L-HSCR-ENCCs. All 2136 cells were included to fit the linear regression. *P* value shows the significance of the fitting (calculated by R lm function). **e** PCA projection of single cells based on AS events reveals that PC2 is highly associated with disease severity. **f** Top AS events positively and negatively contributed to PC2. Putative targets of *PTBP1 (1)* (Inclusion of exon 9) are labeled in red. **g** Significant differentially splicing events between control and S/L-HSCR (two-sided *t*-test). Two controls, four S-HSCR, and two L-HSCR were included for comparison, where a single PSI value was calculated for each sample. *P* values <0.05 and <0.001 are marked by ** and ***. **h** Transcriptional (gene expression) and post-transcriptional (RNA splicing) levels represent two complementary dimensions describing the progression of HSCR disease. PCs described in Figs. 2c and 4f were selected to represent the transcriptional (gene expression) and post-transcriptional (RNA splicing) levels of cells.

exons were involved in "protein binding", "RNA localization", "axonogenesis", and "axon guidance" pathways (Fig. 5d).

One of the known targets of *PTBP1* is *PKM*, which encodes two key enzymes (pyruvate kinase M [PKM] 1 and PKM2) involved in glycolysis, and they are responsible for converting phosphoenolpyruvate (PEP) to pyruvate as a substrate for oxidative phosphorylation (OXPHOS). Upregulation of *PTBP1* or inclusion of the *PTBP1* exon 9 is associated with the binding of *PTBP1* to the splice sites flanking exon 9 in *PKM* transcripts, resulting in mutual exclusion (MX) of exon 9 and inclusion of exon 10 to generate *PKM2* (Fig. 5e). In order to reveal the potential regulation between *PTBP1* and *PKM*, we performed differential co-expression analysis. In brief, the RNA-binding motif of PTBP1 was obtained from RBP binding motif database (mCrossBase). Then, RNA-binding motif enrichment analysis was performed using the FIMO tool of the MEME suite. Significant RNA recognition motif (RRM) enrichment was found in the *PKM* gene bodies of the HSCR-ENCCs (*P* = 4.9 × 10⁻¹⁷), with *PTBP1* exon 9 inclusion inversely correlated with

*PKM* exon 9 (*PKM1*) (Spearman *r* = 0.536). According to the correlation distribution of *PTBP1* exon 9 inclusion to other AS events (Fig. 5b), we found more than 95% of the correlation values located within −0.4 to 0.4. The relatively low correlation value may be attributed to the low RNA materials in single cells, and thus limits the accuracy and sensitivity of the AS event detection by the algorithms. However, the strong correlation between *PTBP1* exon 9 inclusion and MXE10 events in *PKM* suggests a potential regulation between them (Fig. 5e). In concordant with these findings, subsequent quantitative RT-PCR analysis also showed a significantly high *PTBP1* exon 9 and a low *PKM1* transcript level in L-HSCR-ENCCs, while *PKM2* levels did not show association with any disease state (Fig. 5f).

PKM1 efficiently converts PEP into pyruvate that is preferentially used for OXPHOS, favoring neurogenesis[33]. Consistently, the overall expression of OXPHOS pathway genes were found to be decreased in the HSCR cases as shown in the scRNA-seq data, highly correlated with reduced *PKM1* levels (Fig. 5g). Concordantly, the cellular aerobic

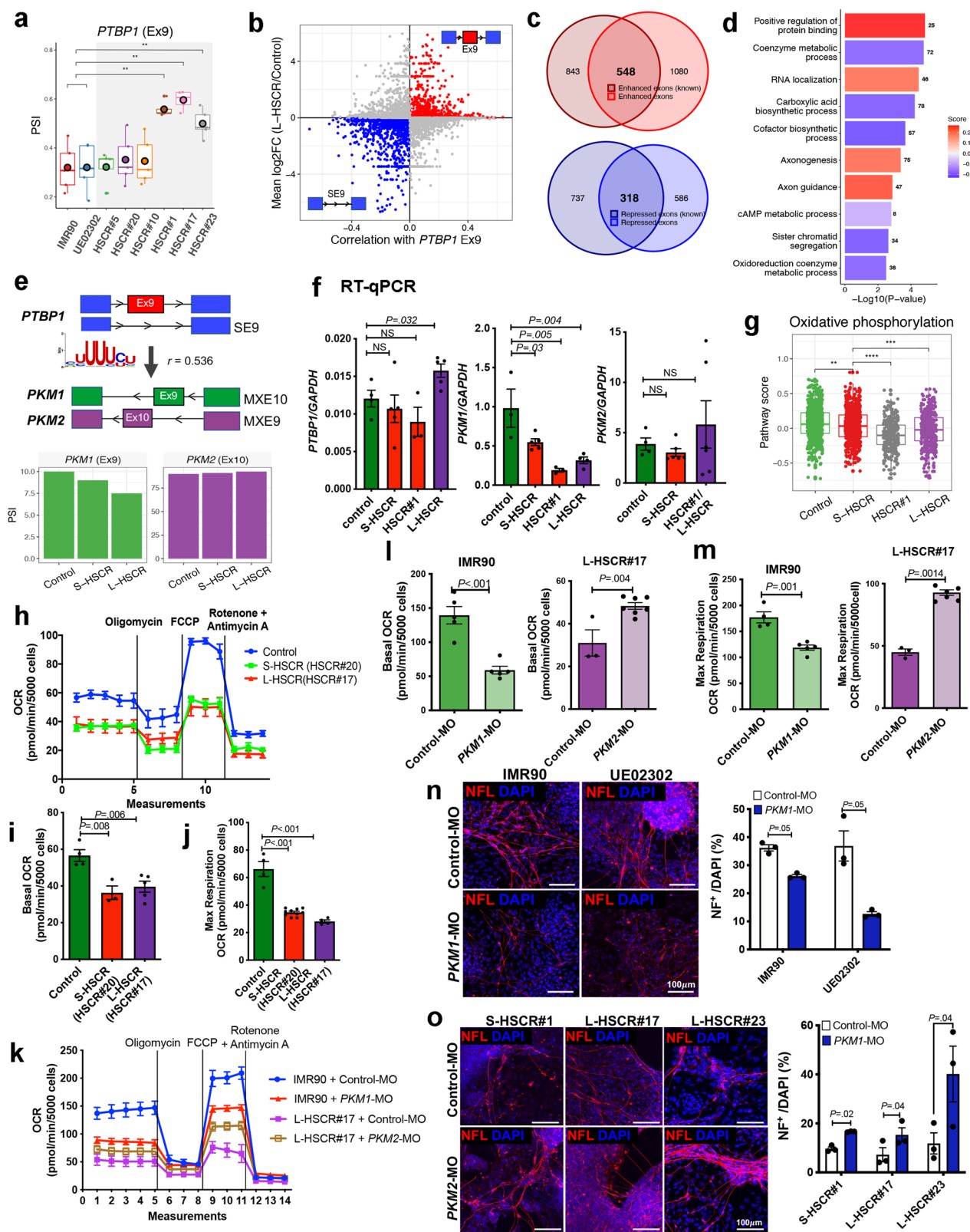

respiration of HSCR-ENCCs was found to be lower than that of the control cells as determined by the oxygen consumption rate (OCR) using Seahorse analyzer. The association of the OXPHOS with the OCR were further illustrated by inducing metabolic stress with the addition of oligomycin, carbonyl cyanide-p-trifluoromethoxyphenylhydrazone (FCCP), and antimycin A/rotenone in succession. In both control- and HSCR-ENCCs, treatment with oligomycin, an ATP synthase inhibitor,

induced a loss of OCR, indicating that the mitochondrial respiration is coupled with ATP production in ENCCs (Fig. 5h). To measure reserve respiratory capacity, which is an indicative parameter of maximal respiratory efficiency of mitochondria, we next added FCCP, an ionophore that induces high proton conductance into the mitochondrial membrane with rapid acceleration of the mitochondrial electron transport chain (ETC). FCCP treatment resulted in a high OCR increase

**Fig. 5 | Inclusion of *PTBP1* exon 9 is associated with dysregulation of cellular metabolic pathways in L-HSCR-ENCCs. a** Boxplot shows the percent spliced in (PSI) of exon 9 (Ex9) inclusion rate in *PTBP1*. Each HSCR-ENCC (HSCR#1 with 306 cells, HSCR#17 with 187 cells, and HSCR#23 with 246 cells) was compared to the merged controls with 754 cells. Each box represents the interquartile range (IQR, the range between the 25th and 75th percentile) with the mid-point of the data, whiskers indicate the upper and lower value within 1.5 times the IQR. Tests marked by "**" represent they are statistically different (two-sided Wilcoxon rank-sum test) from the controls of *P* values <0.01. ns not significantly different. **b** Scatterplot shows the enhanced and repressed exons associated with *PTBP1* Ex9 enriched in L-HSCR. SE9: skipped exon 9. **c** Venn plots shows the PTBP1 Ex9 -enhanced and -repressed exons enriched in L-HSCR overlapping with the known PTBP1-target exons. **d** Gene ontology enrichment analyses of genes enriched with target exons of PTBP1-Ex9 in L-HSCR. *P* value and FDR were calculated by clusterProfiler software. **e** Schematic shows the regulation of PKM1/2 splicing regulated by PTBP1-Ex9. SE: skipped exon; MXE mutually exclusion of exon. Differentially splicing of *PKM1* and *PKM2* between control, S-HSCR, and L-HSCR. Two controls were merged for comparison. **f** RT-qPCR analyses of *PTBP1*, *PKM1*, and *PKM2* expressions in ENCCs derived from the control- and various HSCR-iPSC lines. Data were presented as mean values ± SEM from 3–6 independent experiments. (Student's *t*-test, two-

sided). **g** Boxplot shows the reduced oxidative phosphorylation pathway scores in HSCR-ENCCs as inferred based on the expression of the genes implicated in these pathways. A two-sided Wilcoxon rank-sum test was applied to calculate the significance of the difference between the two groups. All samples (two controls with 754 cells, three S-HSCR with 643 cells, intermediated HSCR#1 with 306 cells, and two L-HSCR with 433 cells) were included for comparison and presented as groups. Tests marked by "****", "***", "**", and "*" represent they are statistically different from the controls of *P* values <0.0001, <0.001, <0.01, and <0.05, respectively. ns not significantly different. Seahorse assays show oxygen consumption rate (OCR) in **h** control (IMR90), S-HSCR, and L-HSCR control and **k** control (IMR90)- and HSCR-ENCCs treated with control-, *PKM1*-, or *PKM2*- morpholinos (MO) (four independent experiments). Quantitative analyses of the **i, l** basal OCR and **j, m** maximum respiratory rate in the control and HSCR-ENCCs with or without morpholino treatment. Data were presented as mean values ± SEM from 3–10 independent experiments. (Student's *t*-test, two-sided). In vitro differentiation assays show NF⁺ neurons derived from **n** control (IMR90 & UE02302)-ENCCs treated with *PKM1*-MO; and **o** HSCR-ENCCs treated with *PKM2*-MO, on day 5 of neuronal differentiation. Data were presented as mean values ± SEM from 3–7 independent experiments. (Student's *t*-test, two-sided). Source data are provided as a Source Data file.

in control-ENCCs, while only a mild increase was observed in the HSCR-ENCCs (Fig. 5h). Both the basal OCR (Fig. 5i) and the maximal respiratory capacity of mitochondria (Fig. 5j) were reduced in HSCR-ENCCs. To demonstrate the involvement of PKM1/PKM2 in this process, particularly in the severe form of HSCR, which showed significant association with *PKM1* expression, we employed morpholinos targeting the splicing sites of *PKM* gene to alter the levels of *PKM1* (Intron 8/ Exon 9) and *PKM2* (Exon 10/Intron 10), respectively, in control- and HSCR-ENCCs. *PKM1*-MO could significantly downregulate the level of *PKM1* transcripts (Supplementary Fig. 8a) and alter the cellular aerobic respiration of ENCCs (Fig. 5k). It reduced both the basal OCR (Fig. 5l) and the maximal respiratory capacity of mitochondria (Fig. 5m) in control-ENCCs. In concordant with these observations, *PKM2*-MO suppressed *PKM2*, accompanied by elevated *PKM1* expression (Supplementary Fig. 8b) and that greatly increased both the basal OCR (Fig. 5l) and maximal respiratory rate (Fig. 5m) of HSCR-ENCCs. More importantly, the levels of *PKM1* and *PKM2* transcripts positively and negatively correlated to the differentiation capacity of ENCCs, respectively. Downregulation of *PKM1* by *PKM1*-MO perturbed the neuronal differentiation of control-ENCCs (Fig. 5n and Supplementary Fig. 8a, c), while upregulation of *PKM1* and suppression of *PKM2* (Supplementary Fig. 8d, e) favored the neurogenic lineage differentiation of HSCR-ENCCs as monitored based on the NF expression on day 5 of differentiation (Fig. 5o). Therefore, the reduced *PKM1* activity in L-HSCR that is associated with the inclusion of *PTBP1 exon 9* likely perturb OXPHOS, leading to the retarded neuronal differentiation of ENCCs.

## Hedgehog-enhanced cellular metabolism promoted the growth and neuronal differentiation of ENCCs

Increasing evidence suggests that neuronal differentiation defect of ENCCs is one of the common problems found in all HSCR patients (S/L-HSCR). Therefore, it is conceivable that rescuing the neuronal differentiation of ENCCs represents a possible treatment strategy. Our previous study demonstrated that activation of the Hedgehog pathway by the addition of a Smoothened agonist (SAG) primes ENCCs to a more advanced cell state and favors neuronal lineage differentiation[8]. Subsequent analysis of the transcriptomes of these SAG-primed ENCCs at single cell level further revealed that the genes implicated in glycolytic and fatty acid oxidation (FAO) pathways are robustly upregulated (Fig. 6a, b). In concordance with these observations, a full activation of Hedgehog pathway by SAG or a partial agonism through applying a low dosage of cyclopamine (Cyc^Low, 1 nM) significantly increased the cellular aerobic respiration of ENCCs as monitored based on the OCR (Fig. 6c), accompanied by significant increases in both

OCR/extracellular acidification rate (ECAR) ratio (Fig. 6d) and the maximum respiration (Fig. 6e). In addition, a specific activation of AMPK-PKM2-PDHA1 pathway was observed upon an addition of SAG or Cyc^Low, and this pathway was blocked by a high dosage of cyclopamine (Cyc^High, 100 nM), which antagonizes the Hedgehog pathway (Fig. 6f). Activation of the AMPK-PKM2 pathway by the full (SAG) or partial (Cyc^Low) agonism of Hedgehog pathway led to an enhanced ATP production, which could be abolished by an addition of PKM2 inhibitor (PKM2i) (Fig. 6g). Both lactate (Fig. 6h) and pyruvate (Fig. 6i) levels were significantly increased upon activation of the Hedgehog pathway, and they served as substrates to support the anaerobic and aerobic cellular metabolism, respectively, resulting in a higher ATP yield. Importantly, the Hedgehog-enhanced metabolic rewiring greatly increased the survival and/or growth of ENCCs, as revealed by the MTT assay (Fig. 6j).

The involvement of the FAO pathway in Hedgehog-enhanced metabolic rewiring was also determined using an FAO-specific inhibitor, Etomoxir (ETO). Unlike glycolysis, a full agonism of the Hedgehog pathway was required for activation of the FAO pathway, where only SAG, but not Cyc^Low, could significantly increase the FAO-associated OCR (Fig. 6k) and maximum respiration (Fig. 6l). In addition, activation of FAO or Hedgehog pathway by GW9508 (a potent and selective G protein-coupled receptors FFA1 (GPR40)) and SAG, respectively, led to a significant enhancement in the mitochondrial activity as monitored by the mitotracker (Fig. 6m, red), while the SAG-treated cells exhibited lower mitochondrial activity when ETO was added, suggesting the involvement of FAO in SAG-mediated mitochondrial activity. More importantly, activation of the FAO pathway conferred ENCCs a greater neuronal differentiation capacity, and significantly more neurons (TUJ1⁺, NF⁺) were found at day 5 of neuronal differentiation when the FAO pathway was activated by the addition of GW9508 (Fig. 6n). Our results support an essential role of the Hedgehog/FAO/mitochondria axis in mediating the neuronal differentiation of ENCCs.

## Hedgehog-induced OXPHOS restored the neuronal differentiation capacity of HSCR-ENCCs

We next sought to determine whether the Hedgehog-induced metabolic rewiring can rescue the differentiation defects of HSCR-ENCCs. The FAO-dependent mitochondria activity in HSCR-ENCCs was elevated upon SAG treatment which was blocked by ETO (Fig. 7a–c). Intriguingly, activation of the Hedgehog/FAO/mitochondria axis could also greatly improve the differentiation capacity of HSCR-ENCCs and significantly more TUJ1⁺ and NF⁺ neurons were detected when FAO pathway was activated by GW9508 (Fig. 7d, e).

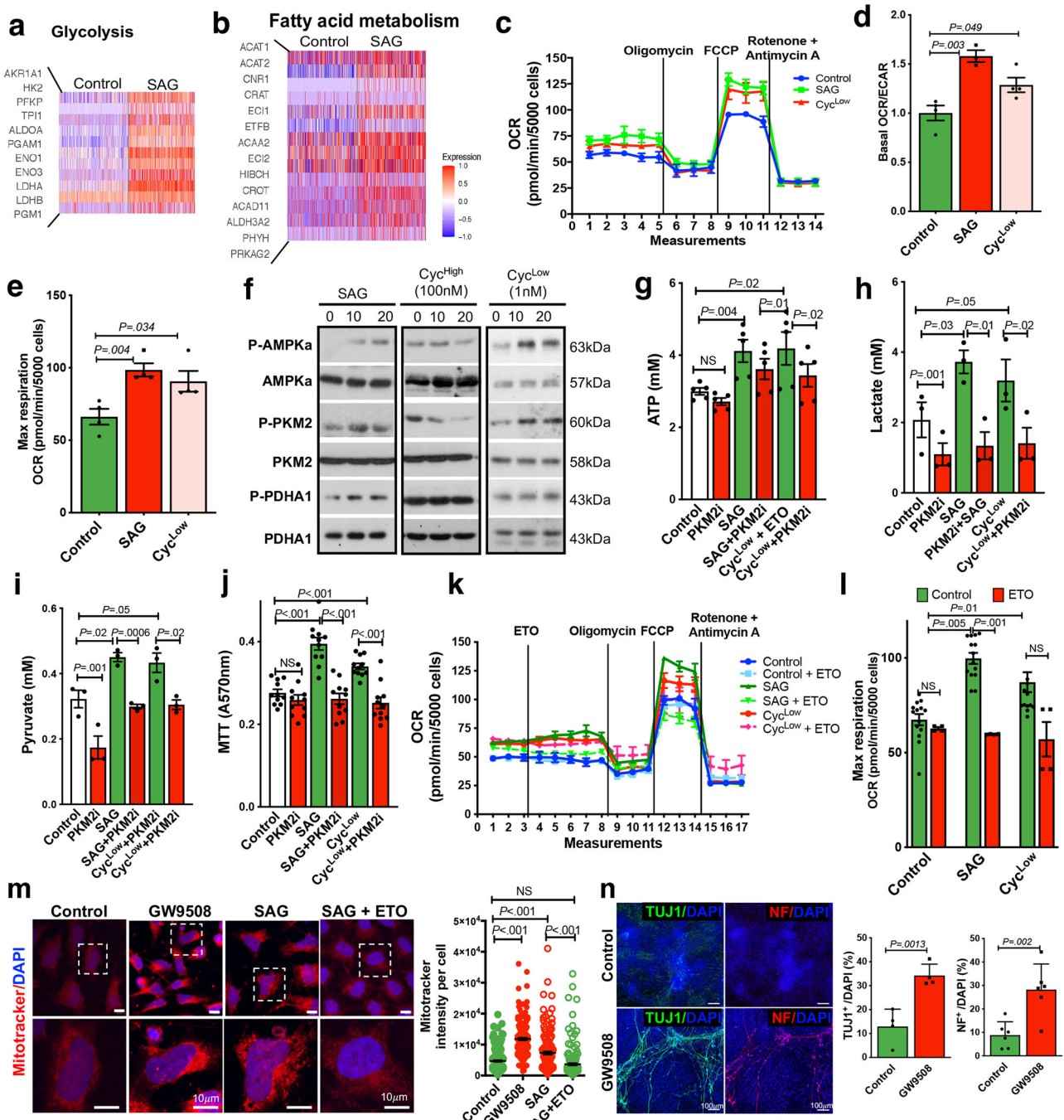

**Fig. 6 | Hedgehog-induced metabolic shift in ENCCs.** Heatmaps show genes implicated in **a** glycolysis and **b** fatty acid metabolism pathways in ENCCs upon the SAG treatment. **c** Seahorse assays of control-ENCCs treated with SAG or cyclopamine (1 nM, Cyc^Low). (Mean ± SEM, $n = 4$). Dot plots from three independent experiments show the changes in **d** the basal oxygen consumption rate (OCR) to extracellular acidification rate (ECAR) ratios and **e** the maximum respiration rates upon different treatments (mean ± SEM, $n = 4$, Student's $t$-test, two-sided). **f** Western blot shows the alternations of AMPK-PKM2-PDHA1 pathway in ENCCs subjected to SAG, cyclopamine (100 nM, Cyc^high and 1 nM, Cyc^Low) treatments. Representative images from three independent analyses are shown. The samples were derived from the same experiment, and the gels/blots were processed in parallel. Bar charts show **g** ATP ($n = 5$), **h** lactate ($n = 3$), **i** pyruvate ($n = 3$), and **j** MTT

($n = 11$) assays in the presence or absence of SAG or Cyc^Low. Bar charts show the mean values ± SEM from independent experiments (Student's $t$-test, two-sided). **k** Seahorse assays of ENCCs treated with SAG and Cyc^Low in the absence or presence of FAO inhibitor (Etomoxir, ETO). Means of OCR ± SEM from three independent experiments are shown. **l** Bar chart shows the maximum respiration rates (mean ± SEM, $n = 4$, Student's $t$-test, two-sided). **m** Mitochondria activities in ENCCs were monitored using the Mitotracker assay. The black line marks the mean values ± SEM from four independent experiments. **n** Immunocytochemistry of TUJ1^+ and NF^+ neurons in day 5 of neuronal differentiation. Bar charts show the mean values ± SEM from 4–6 independent experiments (Student's $t$-test, two-sided). Source data are provided as a Source Data file.

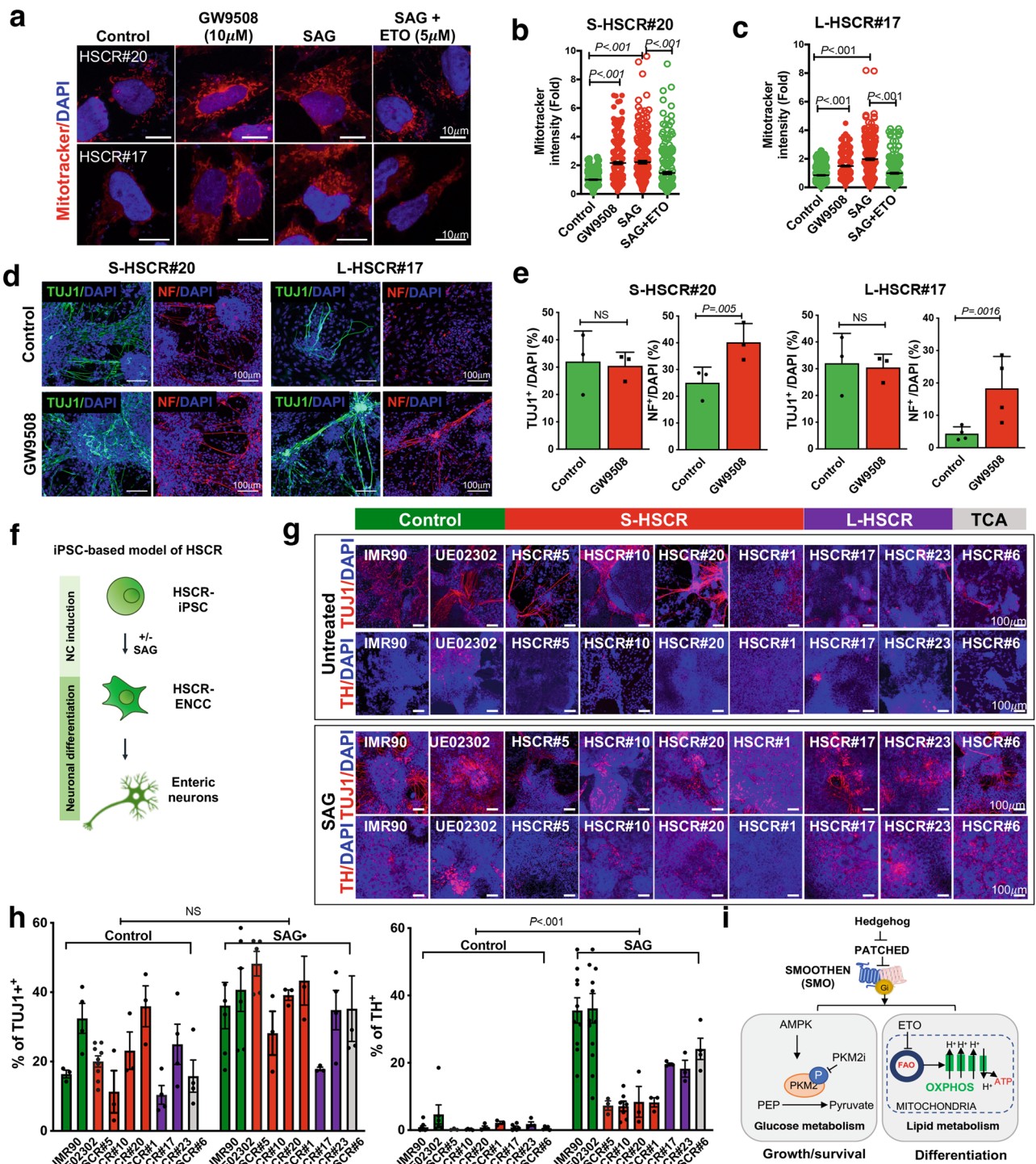

**Fig. 7 | Hedgehog-induced OXPHOS rescues the differentiation defect of HSCR-ENCCs. a** Mitochondria activities in HSCR-ENCCs treated with FAO agonist (GW9508), SAG, and SAG in combination with ETO were monitored using the Mitotracker assay. The corresponding quantitative analyses of mitochondria activities in HSCR#20- and HSCR#17-ENCCs from three independent experiments are shown in (**b**, **c**). Black lines mark the mean values ± SEM. **d** Immunocytochemistry of TUJ1[+] and NF[+] neurons on day 5 of neuronal differentiation. **e** Bar charts show mean values ± SEM from 3–4 independent experiments (Student's *t*-test, two-

sided). **f** Schematic shows the differentiation strategy for the generation of HSCR-ENCCs. **g** Immunocytochemistry of TUJ1[+] and TH[+] neurons on day 9 of the neuronal differentiation of control and HSCR-ENCC with or without pretreatment with SAG. **h** The bar charts show the mean values ± SEM from independent experiments. Data points obtained from independent experiments are shown (two-way ANOVA test). **i** Schematic shows that Hedgehog-mediated glycolysis and FAO regulate the survival and differentiation of ENCCs. Source data are provided as a Source Data file.

Based on the result from the time-course experiment as described previously, the addition of SAG during ENCC induction could maximize the yield and prime cells to the neurogenic lineage with enhanced cellular metabolic rate[8]. Therefore, we employed the same

differentiation strategy to generate HSCR-ENCCs (Fig. 7f). Consistently, SAG-treated HSCR-ENCC-derived TUJ1[+] neurons grew better, exhibited better differentiation competency and required a shorter differentiation time to generate neuronal subtypes, such as tyrosine

hydroxylase (TH) (Fig. 7g, h), while the migration of these cells remained unaffected (Supplementary Fig. 9). In sum, our data suggest that activation of the Hedgehog pathway promotes glycolysis and FAO to support the survival/growth and differentiation of ENCCs (Fig. 7i).

## Discussion

In the present study, we depicted the cellular landscape of HSCR-ENCCs in mild to advanced disease stages by analyzing the transcriptomes of control- and HSCR-ENCCs at the single-cell level. Intriguingly, we identified common core regulatory machineries across different patient bases and built a model network of interacting disease-modifiable factors that detail the complex nature of HSCR (Fig. 8).

Through high-resolution scRNA-seq and systemic multi-patient comparison analyses, we defined the core and disease-modifiable gene sets associated with HSCR pathogenesis. As supported by the clinical and molecular features of the intermediate case (HSCR#1), we observed a pseudo-disease severity axis that can reflect the difference from mild to severe HSCR disease states (S-HSCR to L-HSCR). Additionally, our results suggest that disruption of RNA splicing and energy metabolism pathways was associated with the disease severity axis and would promote more extensive aganglionosis. Importantly, we identified a core gene set with 118 genes representing misregulated molecular machinery in S- and L-HSCR. Specifically, HDAC1 was identified to be the potential regulator of the core gene set, where transcriptional activation of HDAC1 targets were found elevated in HSCR-ENCCs.

Enhanced AS events were found in HSCR-ENCCs, particularly, in the more advanced disease states. It could be the result of the upregulation of RNA splicing mediating factors. Many of the top disease-associated AS genes, such as *PTBP1*, *AK3*, and *USP14*, are associated with neurogenesis and energy metabolism. In particular, the inclusion of exon 9 in *PTBP1* contributed to many HSCR-associated AS events. The inclusion of exon 9 in *PTBP1* was associated with the reduced MXE10 in *PKM* and a significantly lower abundance of *PKM1* in ENCCs in advanced disease states. These changes likely prohibit metabolic rewiring from glycolysis to OXPHOS, which is required for supporting energy needs in cell state transition and differentiation[34,35]. In concordant with the expression data, significantly lower OXPHOS was consistently detected in HSCR-ENCCs and that was highly associated with the poor survival and differentiation capacity of HSCR-ENCCs.

Glycolysis and FAO are essential metabolic pathways for producing pyruvate and acetyl-CoA, respectively, to fuel the tricarboxylic acid (TCA) cycle for ATP generation under an energy-demanding condition. The Hedgehog signaling is frequently coupled with bioenergetics of various cancers, and is associated with the insulin-independent glucose uptake in muscle and brown adipose tissue in obese and diabetic patients[13]. Here, we used Cyc^Low and SAG to achieve a partial agonism and full activation of the Hedgehog pathway and revealed that the Hedgehog-mediated glycolytic and FAO metabolic pathways greatly enhance the survival and neuronal differentiation competency of ENCCs, respectively. These functions are conserved in HSCR-ENCCs, which usually have lower OXPHOS activity. According to the transcriptomic analysis as described previously, activation of the Hedgehog pathway primes ENCCs to neurogenic lineage without altering their neuronal differentiation trajectory[8]. It suggests that Hedgehog/FAO/OXPHOS axis would be the potential mechanism underlying this process, which is consistent with the preference for mitochondrial OXPHOS during the cell state transition and neuronal differentiation. The non-canonical metabolic routing represents a quick and flexible way to provide energy and prepare ENCCs for cell state transition/differentiation, but it likely requires reinforcement by the canonical Hedgehog signaling, which further initiates a transcriptional glycolysis response and modulates the expression of the entire machinery of primary metabolism including the Hedgehog/FAO/OXPHOS axis. As supported by our own transcriptional profiling data of SAG-primed ENCCs and the data from the differentiation assays, the full activation of the Hedgehog pathway by SAG treatment robustly upregulates the expression of genes implicated in various cellular metabolic pathways, and that leads to a sustainable and profound effect on cell state transition and differentiation. In particular, HSCR-ENCCs frequently show low OXPHOS activity, the prolonged activation of Hedgehog is required for improving their survival and differentiation capacity, without altering their migration capacity. Nevertheless, the aberrant activation of the canonical pathway is associated with the pro-oncogenic nature of the Hedgehog pathway and HSCR disease. The use of an FAO agonist, or in combination with a selective partial agonism of the Hedgehog pathway, may represent an alternative approach to restore the functions of ENCCs in HSCR patients and warrants further studies.

In conclusion, our study provides a comprehensive biological picture of the core molecular mechanisms underlying HSCR heterogeneity. In particular, we show that metabolic rewiring is critical for the development of ENCCs and the defective OXPHOS represents an underlying cause of L-HSCR.

## Methods

### Patients

Seven HSCR patients from different subgroups (four S-HSCR, two L-HSCR, and one TCA) were recruited at Queen Mary Hospital, Hong Kong. Each patient was given a disease score according to the length of

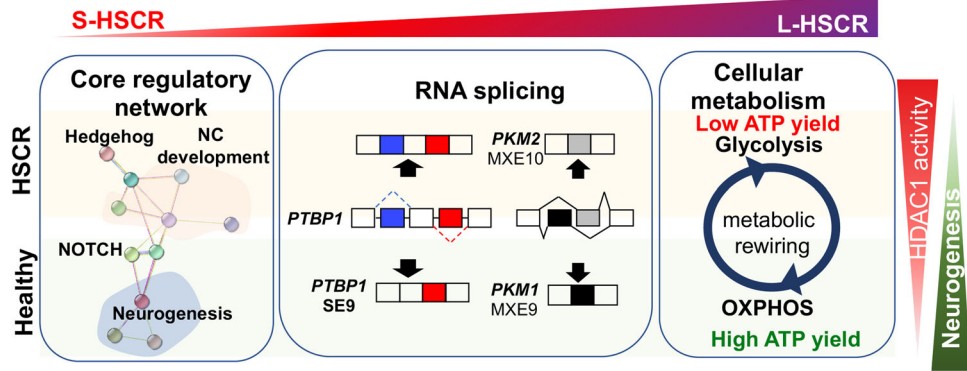

**Fig. 8 | Schematic summaries of the key molecular and cellular processes associated with HSCR pathogenesis and severity.** A core gene set associated with the severity of HSCR was identified. Additional alternations in genes implicated in RNA splicing and oxidative phosphorylation (OXPHOS) were found in L-HSCR cases. High HDAC1 activity is associated with defective neuronal differentiation, as observed in HSCR-ENCCs. Only the representative genes/pathways are shown.

the aganglionic segment (0–5; control = 0 and TCA = 5) (Fig. 1a, b). The *RET* alleles in rs2435357 and variants in the coding region of *RET* gene were identified by sequencing the genomic DNA from the patient's blood. In brief, among the four S-HSCR patients, HSCR#5, #10, and #20 showed relatively mild phenotypes with disease scores 1 or 2, while HSCR#1 had a longer aganglionic segment up to the lower sigmoid, representing the intermediate phenotype between S- and L-HSCR. Skin biopsies were obtained from these patients for the generation of iPSC lines. The study was approved by the institutional review board of The University of Hong Kong together with the Hospital Authority (UW 13-419). Informed consent was obtained from participants.

### Human induced pluripotent stem cells (iPSCs)
A control hiPSC line (IMR90-iPSC) was obtained from WiCell Research Resources (WiCell, WI., RRID:CVCL C434). Another control hiPSC cell line (UE02302) was generated from urine-derived cells of a male individual by episomal reprogramming vectors carrying the four reprogramming factors[36]. All iPSCs used in this study were at the intermediate (35-65) passage numbers and maintained on Matrigel (Corning, 354277)-coated plate with mTeSR1 medium (StemCell Technologies, 85850).

### Generation of iPS cell lines
Fibroblasts were derived from skin biopsies obtained from patients diagnosed with HSCR disease. Fibroblasts were then maintained and expanded in DMEM with 10% FBS. iPSC were generated from fibroblasts using episomal vectors carrying the four reprogramming factors (Oct4, Klf4, Sox2, and c-Myc)[37]. In brief, $1 \times 10^6$ fibroblasts were transfected with episomal vectors using Nucleofector with a transfection kit (Lonza, VPD-1001) and the transfected cells were then replated on a 35 mm gelatin-coated well. The medium was changed 24 h later and replaced daily from thereon. At day 7, cells were dissociated and replated on MEFs in ES medium (DMEM/F12 (Thermo Fisher Scientific,, 10565018) plus 10% KnockOut Replacement Serum (KSR) (Thermo Fisher Scientific, 10828028), 0.5% L-glutamine (Thermo Fisher Scientific, 25030018), 1% NEAA (Thermo Fisher Scientific, 11140050), 0.1% β-mercaptoethanol (Thermo Fisher Scientific, 21985023), 0.5% penicillin-streptomycin (Thermo Fisher Scientific,15140122), and FGF2 (Peprotech, 100-18B). Cells were fed daily with ES medium. Following the appearance of hESC-like colonies after approximately 3–4 weeks, individual colonies were isolated and clonally expanded to establish iPSC lines. All the iPSC lines generated in this study were fully characterized following the standard protocols (Supplementary Fig. 9).

### Derivation of neural crest (NC) from iPSC lines
At day 0, Control or HSCR-iPSCs were seeded on a Matrigel-coated plate ($10^5$ cells cm$^{-2}$) in iPS cell medium containing 10 ng ml$^{-1}$ FGF2 and 10 μM Y-27632 (Tocris Bioscience, 1254). Differentiation was then initiated by replacing iPSC medium with KSR medium, containing KnockOut DMEM (Thermo Fisher Scientific, 10829018) plus 15% KSR, 1% NEAA, 1% L-glutamine, 1X β-mercaptoethanol, 1% penicillin-streptomycin, LDN193189 (100 nM, Reprocell, 04-0074), and SB431542 (10 μM, Abcam, ab120163). The dual SMAD inhibitors and a potent GSK inhibitor were added at different time frames during the NC induction, including LDN193189 (from day 0 to day 3), SB431542 (from day 0 to day 4), and 3 μM CHIR99021 (from day 2 to day 10, Reprocell, 04-0004). The NC cells were finally caudalized with 1 μM retinoic acid (Abcam, ab120728) (from day 6 to day 9). The KSR medium was gradually changed to N2 medium on day 4 by increasing N2 from 25 to 75% from day 4 to 9[7]. The N2 medium contained neural basal medium (Thermo Fisher Scientific, 22103049) and DMEM/F12 in a 1:1 ratio supplemented with 0.5% N2 supplement (Thermo Fisher Scientific, 17502048), 1% B27 supplement (Thermo Fisher Scientific, 17504044), 5 μg ml$^{-1}$ insulin (Thermo Fisher Scientific, 12585014), and 1% penicillin-

streptomycin. The ENCCs were enriched by FACS with antibodies against p75$^{NTR}$ and HNK-1 at day 10 of the differentiation as described in refs. 7, 8, 19. FACS-sorted ENCCs were collected for single-cell RNA sequencing.

### In vitro differentiation of ENCCs to neuronal progenitors (NPs)
Around 40 thousand FACS-enriched ENCCs were seeded as droplets on poly-ornithine/laminin/fibronectin (PO/LM/FN)-coated 24-well plates in N2 medium containing 10 ng ml$^{-1}$ FGF2, 3 μM CHIR99021 and 10 μM Y-27632. The neuronal differentiation started 48 h later and the attached ENCCs were then cultured with N2 medium containing BDNF (10 ng ml$^{-1}$ Peprotech, 450-01), GDNF (10 ng ml$^{-1}$, Peprotech, 450-10) and ascorbic acid (200 μM, Sigma, A4034-100G), NT-3 (10 ng ml$^{-1}$, Peprotech, 450-03), NGF (10 ng ml$^{-1}$, PeproTech, 450-01), and cAMP (1 μM, Sigma, D0260). For valproic acid (VPA) treatment, 1 mM VPA (Sigma, P4543) was added during neuronal differentiation. The culture medium was changed every 2 days. ENCC-derived neurons at differentiation day 5 or 9 were fixed for immunocytochemistry analyses.

### Fluorescence-activated cell sorting (FACS) and flow cytometric analysis
For quantifying the ENCCs, the 10-day-differentiated cells were dissociated with Accutase and then incubated with anti-human antibodies, including APC-HNK-1 (BD Pharmingen, 560845) and FITC-p75$^{NTR}$ (Miltenyi Biotec, 130-091-917) for 30–45 min on ice. To stain for PE-RET (Neuromics, FC15018), the cells were fixed in 4% paraformaldehyde for 10 min at room temperature and permeabilized using 0.1% (w/v) Saponin solution, then washed and blocked in PBS with 2% FBS. The cells are then stained with antibodies for 30–45 min on ice. Approximately $10^6$ cells were stained, and labeled cells were detected using FACSAriaIII (Becton Dickinson Immunocytometry Systems, San Jose, CA, USA). Isotype-matched antibodies were used as controls. FlowJo version 8.2 (Tree Star, Inc.) was used to analyze flow data.

For cell sorting, HNK-1/p75$^{NTR}$ stained cells were washed and resuspended in PBS with 2% FBS. The HNK-1 and p75$^{NTR}$ double-positive cells were enriched using fluorescence-activated cell sorting (FACS) with BD FACSAria III Cell Sorter. The HNK-1 and p75$^{NTR}$ double-positive cells were gated and sorted using the four-way purity mode, and the purity of sorted cells was >96% and evaluated by flow cytometry. The sorted neural crest cells were collected for immunostaining or subsequent experiments. A list of primary antibodies and the working dilutions are provided in Supplementary Table 1.

### Immunofluorescence analysis
For immunocytochemistry, the cells were fixed with 4% paraformaldehyde in PBS at room temperature for 30 min, followed by blocking with 1% bovine serum albumin (BSA) (Thermo Scientific, 23209) with or without 0.1% Triton X-100 (Sigma, T8787) in PBS buffer. Cells were then incubated in primary antibody overnight at 4 °C and host-appropriate Alexa Fluor −488 or 594 secondary antibody (Molecular Probes, Invitrogen) (Supplementary Tables 1, 2) for 1 h at room temperature. Cells were then counterstained with a mounting medium with DAPI (Thermo Scientific, P36931) to detect nuclei. Cells were photographed using Carl Zeiss confocal microscope (LSM810). Quantitative image analysis of differentiated neuronal cultures was performed with the ImageJ plug-in tool. The total signal of the neuronal marker was normalized with the DAPI signal and the values reported in the bar charts represent the mean ± SEM. The same approach was used to quantitate the expression of the neuronal marker in various groups. A minimum of 4000 cells were analyzed per sample.

The nuclear and cytoplasmic HDAC1 expressions were quantified at the single-cell level using the CellProfiler 4[38]. To enhance the measurement precision, cells were imaged using a Carl Zeiss Confocal

microscope LSM810 with a high numerical aperture lens, 63x, and the same imaging settings. Images were saved in a megapixel resolution. A pipeline was set up to measure the HDAC1 expressions in each cell on the images. In brief, the nuclear and cytoplasmic regions were defined using the "IdentifyObjects" modules and the DAPI- and HDAC1-stained images, respectively. The HDAC1 expressions were measured in the defined nuclear and cytoplasmic regions for each cell using the module "MeasureObjectIntensity". For the cell lines where the cytoplasmic region was difficult to define, the cytoplasmic HDAC1 expression was measured in the perinuclear region instead of the entire cytoplasmic region. The nuclear-to-cytoplasmic HDAC1 ratio was calculated and compared between cell lines using one-way ANOVA. For each cell line, more than 200 cells were analyzed.

### Migration assay

FACS-enriched ENCCs were plated on human fibronectin-coated 12-well culture plates (30,000 cells cm$^{-2}$). After 24 h, cells were treated with mitomycin (10 µg ml$^{-1}$) to stop cell proliferation. A wound was created in the center of each well by scratching it with a pipette tip. Cells were allowed to migrate for 18 h. The images of the initial wound and final wound were captured immediately and at 18 h after scratching. The migration distance was obtained by comparing the width of the initial wound created and wound closure during 18 h.

### Morpholino (MO) delivery into ENCCs

Translation-blocking morpholino against *HDAC1* and splice-modifying morpholinos targeting *PKM*: *PKM1*-MO (targeting splice junction between intron 8 and exon 9 of *PKM*) and *PKM2*-MO (targeting splice junction between exon 10 and intron 10 of *PKM*), were transfected into FACS-ENCCs with Endo-Porter PEG (Gene Tools) according to the manufacturer's instructions. In brief, ENCCs was replenished with fresh culture medium (N2 medium containing 10 ng ml$^{-1}$ FGF2 and 3 µM CHIR99021) containing 5 µM morpholinos (*HDAC1*-MO, *PKM1*-MO, *PKM2*-MO, or control-MO) and swirled gently. Then, 6 µl Endo-Porter PEG was added to every 1 ml of medium and swirled immediately to mix. Neuronal differentiation was started 1 day after transfection, the medium was replaced with N2 medium supplemented with GDNF (10 ng ml$^{-1}$) and ascorbic acid (200 µM) and maintained for 5 days. For seahorse analysis, the cellular metabolisms of ENCCs were assayed 48 h after transfection. The oligonucleotide sequences of morpholinos are listed in Supplementary Table 3.

### Reverse transcription-PCR (qRT-PCR)

Total RNA was isolated from iPSC-derived ENCCs using RNeasy Mini Kit (Qiagen); and reverse transcribed in a 20 µl reaction system using HiScript II Q RT SuperMix (Vazyme, R223-01), in accordance with the manufacturer's instructions. diluted cDNA samples were amplified by Luna® Universal qPCR Master Mix (New England BioLabs, M3003) with forward and reverse primers specific for *HDAC1*, *PKMEx2-4*, *PKM1* (*PKM-Ex9*), and *PKM2* (*PKM-Ex10*). The fluorescence signal was measured by ViiA 7 Real-Time PCR System (Thermo Fisher Scientific) at the end of each cycle. Each individual sample was assayed in triplicate and gene expression was normalized with *GAPDH* expression. The primer sequences are listed in Supplementary Table 4.

### Western blotting

FACS-sorted ENCCs were collected and lysed using protein lysis buffer containing 50 mM Tris-HCl, pH7.5, 100 mM NaCl, 1% Triton X-100, 0.1 mM EDTA, 0.5 mM MgCl2, 10% glycerol, protease inhibitor cocktail (Roche) and phosphatase inhibitor cocktail (Roche). About 20 µg of total protein from cell lysates was separated on 12% SDS-polyacrylamide gels and blotted with the corresponding primary antibodies. A list of primary antibodies and working dilutions is provided in Supplementary Table 1. The same membranes were stripped and hybridized with an anti-β-actin monoclonal antibody (Millipore,

MAB 1501) as a protein-loading control. All blots were incubated with secondary horseradish peroxidase-conjugated anti-mouse or anti-rabbit, or anti-goat antibodies (1:2500, DAKO).

### Seahorse analysis

The Agilent Seahorse XFe96 Analyzer (Seahorse Bioscience, Billerica, MA) was used to measure basal, ATP-linked, and maximal oxygen consumption rate (OCR); reserve capacity; and extracellular acidification rate (ECAR). Cellular bioenergetics was performed using XF Cell Mito Stress Test Kit (Agilent Technologies, 103010-100), XF Glycolysis Stress Test Kit (Agilent Technologies, 103020-100), and XF Long Chain Fatty Acid Oxidation Stress Test Kit (Agilent Technologies, 103672-100). About 5 × 10$^4$ of FACS-enriched ENCCs (Passage 1−2) were seeded onto fibronectin-coated Seahorse XF96 V3 PS Culture Microplate (Seahorse Bioscience Inc, Billerica, MA, USA, 101085-004) in N2 medium containing 10 ng ml$^{-1}$ FGF2 and 3 µM CHIR99021. After a 24-h incubation, N2 medium from each well were removed, and washed once with pre-warmed assay medium (Seahorse XF DMEM Medium, Seahorse Bioscience Inc, 103575-10) supplemented with 25 mM glucose, 2 mM L-glutamine, 1 mM sodium pyruvate, 0.5% N2 supplement, 1% B27 supplement, 1% penicillin-streptomycin, 10 ng ml$^{-1}$ FGF2 and 3 µM CHIR99021, then 180 µl of assay medium was added. Cells were incubated in a 37 °C incubator without CO$_2$ for 1 h to allow to pre-equilibrate with the assay medium. Cells were treated with/without PKM2 inhibitor (compound 3k, 1 µM, Selleckchem, D8375), SAG (1 µM), or cyclopamine-KAAD (1 nM) for 1 h.

For Mito Stress Test, tenfold concentrated compounds in the kit of oligomycin (Complex V inhibitor), carbonyl cyanide-4 (tri-fluoromethoxy) phenylhydrazone (FCCP, mitochondrial uncoupler), and a mixture of rotenone (complex I inhibitor) and antimycin A (complex III inhibitor) were loaded into the cartridge to produce final concentrations of 1.2, 2, and 1 µM respectively.

For the Glycolysis stress test, tenfold concentrations of compounds from the kit containing glucose (fuel for glycolysis), oligomycin, and 2-deoxy-D-glucose (2-DG, competitive inhibitor of hexokinase) were loaded into the cartridge to produce the final concentrations of 10, 1.2, and 50 mM, respectively.

For the long-chain fatty acid oxidation stress test, tenfold concentrations of compounds from the kit containing etomoxir (ETO, fatty acid oxidation inhibitor), oligomycin, FCCP, and a mixture of rotenone and antimycin were loaded into the cartridge to produce final concentrations of 5, 1.2, 2, and 1 µM respectively.

Data were analysed using Agilent Seahorse analytics and normalized with the number of cells seeded in each well.

### Lactate and pyruvate assay

Lactate and pyruvate levels in the medium was measured using Lactate Colorimetric Assay Kit (BioVision Incorporated, #K607) and Pyruvate Colorimetric/Fluorometric Assay Kit (BioVision Incorporated, #K609) according to the manufacturer's instructions, respectively. In brief, FACS-enriched ENCCs (Passage 1−2) were seeded at 5 × 10$^5$/well onto a fibronectin-coated 12-well plate and incubated overnight. On the next day, cells were treated with/without PKM2 inhibitor (1 µM), SAG (1 µM), or cyclopamine-KAAD (1 nM) for 1 h. The medium were then collected, and the levels of lactate or pyruvate were measured simultaneously. The levels were normalized to DNA concentration. The experiments were performed in triplicate.

### ATPlite luminescence assay

ATP was detected using the ATPlite Luminescence ATP Detection Assay System (PerkinElmer, #6016943) according to the manufacturer's protocol. In brief, FACS-enriched ENCCs (Passage 1−2) were seeded at 5 × 10$^3$/well onto a fibronectin-coated 96-well plate and incubated overnight. On the next day, cells were treated with PKM2 inhibitor (1 µM), SAG (1 µM), or cyclopamine-KAAD (1 nM) for 1 h. After

incubation, 50 µl lysis buffer (from the ATPlite kit) was added and shaken for 5 min. Then, 50 µL substrate (from the ATPlite kit) was added and incubated for 15 min in the dark. The ATP content was measured with VICTOR Nivo Multimode Microplate Reader (PerkinElmer).

## Mitochondria activity

Mitochondrial activity was measured using MitoTracker Red CMXRos, which is a red-fluorescent dye that stains mitochondria in live cells, and its accumulation is dependent upon membrane potential. Cells with high mitochondrial activity will pick up and accumulate more dye, which can be detected using fluorescent imaging. To measure the mitochondrial activity, FACS-enriched ENCCs (Passage 1–2) were seeded at $1.5 \times 10^5$/well onto a fibronectin-coated 24-well plate and incubated overnight. On the next day, cells were washed once with PBS, and starved in 2 mM glucose NC medium for 2 h. Cells were then treated with GW9508 (10 µM), SAG (1 µM), or ETO (5 µM, 1 h) + SAG (1 µM) for 1 h. About 250 nM MitoTracker Red (Life Technologies, M7514) were then added and incubated for 30 min at 37 °C, 5% $CO_2$. After incubation, cells were washed twice with PBS, and then fixed with 4% paraformaldehyde for 20 min at room temperature. After fixation, cells were stained with DAPI, and imaged using a 63x objective on Carl Zeiss confocal microscope (LSM 800). At least 200 cells were counted in each experiments and 3–5 independent experiments were platformed. Data were shown as means of all replicates with SEM.

## MTT assay

FACS-enriched ENCCs (Passage 1–2) were seeded at $5 \times 10^3$/well onto a fibronectin-coated 96-well plate and incubated overnight. On the next day, cells were treated with PKM2 inhibitor (1 µM), SAG (1 µM), or cyclopamine-KAAD (1 nM) for 1 h. For pyruvate or lactate treatment, cells were washed once with PBS, and starved in 2 mM glucose NC medium for 2 h. Then 2 mM pyruvate, 2 mM pyruvate + 1 µM SAG, 4 mM lactate, or 4 mM lactate + 1 µM SAG were added to the cells, and incubated for 3 or 6 h. About 10 µl of MTT (3-[4,5-dimethylthiazol-2-yl] −2,5-diphenyltetrazolium bromide) labeling reagent (final concentration 0.5 mg/ml, Sigma, M2128) was added to each well for 3 h. Medium with labeling reagents were then removed, and 200 µl DMSO was added to solubilize the purple formazan crystals. The absorbance was measured at 570 nm using VICTOR Nivo Multimode Microplate Reader (PerkinElmer).

## Plate-based scRNA-seq

Plate-based single-cell (sc) RNA sequencing were performed at the Centre of Genomic Science, The University of Hong Kong. For the scRNA-seq of human cells, the Smart-seq® v4 (Clontech) kit was used for first-strand synthesis. Single cells were directly sorted into 4 µl of lysis buffer in a 384-well plate using a FACSAria III flow cytometer (BD Biosciences). First-strand DNA was synthesized within 16 cycles of amplification according to the manufacturer's instructions. cDNA was purified on Agencourt AMPureXP magnetic beads, washed twice with fresh 80% ethanol and eluted in 17 µl of elution buffer. Then, 1 µl of cDNA was checked and quantified on an Agilent Bioanalyzer high-sensitivity DNA chip. Sequencing libraries were produced using Illumina Nextera XT tagmentation according to the manufacturer's instructions except using 150 pg of input cDNA, 5 min of tagmentation and 12 cycles of amplification using the Illumina XT 24 index primer kit. Libraries were cleaned using an equal volume (50 ml) of Agencourt AMPureXP magnetic beads and resuspended in 20 µl of elution buffer. Libraries were checked and quantified on an Agilent Bioanalyzer high-sensitivity DNA chip (size range 150–2000 bp) and by Qubit dsDNA BR (Molecular Probes). Libraries were pooled to a normalized concentration of 1.5 nM and sequenced on an Illumina™ NextSeq 500 using the 150 bp paired-end kit as per the manufacturer's instructions.

## Preprocessing and quality control of smart-seq single-cell RNA sequencing data

Fastq files with paired-end reads were trimmed by Cutadapt (version 1.7) to remove read adapters or low-quality tail bases and then aligned to the ENSEMBL GRCh38 (release 90, https://www.ensembl.org/) human transcriptome by using Bowtie2 (version 2.3.4.1)[39] with options "–sensitive–mp 1,1–np 1 –score-min L,0,−0.1 -I 1 -X 2000 –no-mixed –no-discordant -N 1 -L 25 -k 200," which allows 1 mismatch during sequence alignment. Quality control of the reads of each cell was assessed by using FastQC (version 0.11.6). The gene expression level was quantified by using raw read count number and Transcript per million (TPM) values generated by RSEM (version 1.3.0)[40]. For the quality control of the genes and cells, genes that were detected in less than five cells and cells that expressed with either fewer than 3000 genes or more than 9000 genes, or greater than 20% mitochondrial genes were excluded from the expression matrix. Finally, our study included 2985 high-quality cells for downstream analysis. The distributions of uniquely mapped read count and detected gene number in the cells are shown in Supplementary Data 1 and Supplementary Fig. 2A–C.

## Dimensionality reduction and cell clustering

The R package Seurat (version 3.1.4)[22] implemented in R (version 3.6.2) were used to perform dimensional reduction analysis on iPSC-derived ENCCs RNA-seq data. The "NormalizeData()" function from Seurat was used to normalize the raw counts, and the scale factor was set to 200,000, then followed by "FindVariableFeatures()" with default parameters to calculate highly variable genes for each sample. After performing "JackStraw()", which returned the statistical significance of PCA (principal component analysis) scores, we selected 15 significant PCs to conduct dimension reduction and cell clustering. Then, cells were projected in 2D space using $t$-SNE ($t$-distributed Stochastic Neighbor Embedding) with default parameters. Individual iPSC-derived ENCC was clustered using different resolutions and was finally integrated into five main clusters by the expression of canonical markers.

## Single-cell data integration and batch effect correction

To account for the batch effect among different samples, we used "FindIntegrationAnchors()" in the Seurat package to remove the batch effect and merge samples to one object. In detail, the top 4500 genes with the highest expression and dispersion from each sample were used to find the integration anchors, and then the computed anchoret was applied to perform dataset integration.

## Identification of differentially expressed genes and pathway enrichment analysis

To identify unique differentially expressed genes (DEGs) among each cluster, the "FindAllMarkers()" function from Seurat was used and nonparametric Wilcoxon rank-sum tests were set to evaluate the significance of each individual DEG. The DEG analysis between HSCR and control iPSC-derived ENCCs based on single-cell expression data were performed using monocle (version 2.14.0) R package[26]. The DEGs with adjusted $P$ value less than 0.05 and $\log_2$ fold-change (log2FC) larger than 0.5 were thought to be significant and used in downstream analysis. Gene ontology (GO) term enrichment analyses were performed using clusterProfiler (version 3.14.3) R package[41]. Terms that had an adjusted $P$ value <0.05 was defined as significantly enriched. Since two independent controls were included in the analyses, each HSCR-ENCC was compared to each control separately. Then, we identified the consensus significant DEGs (overlapped DEGs) of each HSCR-ENCC. For example, HSCR#1 consensus DEGs were the overlapped DEGs of HSCR#1/IMR90 and HSCR#1/UE02302. This consensus DE analysis would be able to identify more reliable HSCR DEGs compared to DEGs identified merely based on one control. The two controls were very similar to each other, and only 583 DEGs were identified between the

two controls. Two controls were merged for comparison (calculate *P* value) in the pathway-based analyses (Figs. 2g–j, 3e, 5g).

## Pathway score analysis

In order to measure the activity of whole pathways between samples at the transcriptome level, we applied a simple additive model that ignores the interactions between genes to estimate the overall expression level of the pathways. The genes involved in neurogenesis (GO:0050771), proliferation (GO:0051726), RNA splicing (GO:0033120), and cellular respiration (GO:1901857) in the GO database were used to evaluate the overall changes of the key biological processes.

## Histone deacetylase activity inference

Key components of Sin3, NuRD, and CoREST complexes were obtained from the public database. A geometric average model was applied to estimate the overall activity of histone deacetylase complexes based on the average expression of their subunits. A geometric average was used to set the complexes' activity to zero if any of the subunits was not expressed in single cells.

## Transcriptomic regulatory timing analysis

In order to detect the "switch-like" upregulation or downregulation period of genes along the pseudotime axis, we built a statistical model that probabilistically assigns a region along the axis associated with the positive or negative activation of each gene in the core gene set. Unlike most algorithms that only capture the global changes of gene expression along pseudotime, we intend to infer the local regulatory periods, where gene expression shapely change. This can help us to impute the order of gene regulation along a pseudotime. In brief, we estimated the local changing rate of gene expression along the trajectory for each gene in the core gene set. To measure the local change rate, we divided the pseudotime of the trajectory (0 to 1) into 50 bins. Then, we applied linear regression to obtain the slope and significance of gene expression changes for each bin. Local regressions with *P* value <0.05 were kept and boxplot was used to visualize the concentrated "switch-like" regulatory period of genes along the pseudotime.

## Gene regulatory network (GRN) analysis

Single-cell regulatory network inference and clustering (SCENIC, version 1.1.2)[28] was used to infer transcription factor networks active using scRNA-Seq data. Analysis was performed using default and recommended parameters as directed on the SCENIC vignette (https://github.com/aertslab/SCENIC) using the hg19 RcisTarget database. Kernel density line histograms showing differential AUC score distribution across conditions were plotted with ggplot2 v.3.1.1 using the regulon activity matrix ('3.4_regulonAUC.Rds', an output of the SCENIC workflow) in which columns represent cells and rows the AUC regulon activity. Fold-change (FC) difference between median AUC values was calculated and the highest changed TFs were plotted.

## Differential co-expression analysis

To predict HDAC1-activated and repressed target genes, we applied a three-step strategy to obtain the target genes of HDAC1 during HSCR pathogenesis. In the first step, we identified the significantly correlated genes to HDAC1, which represented the genes that had strong regulatory relationships with HDAC1. In the second step, we utilized the differential expression profile between the control and HSCR-ENCCs to filter out the genes whose expression levels were not significantly disrupted by HDAC1. Finally, to confine the genes to be the actual targets of HDAC1, a public ChIP-seq dataset designed for HDAC1 from a similar cell state (neuronal progenitor) in the Cistrome database[42] was utilized to obtain the direct/indirect binding genes of HDAC1. Activated and repressed target genes were determined by the positive/negative correlation and up/downregulation of the differential expression, respectively. In addition, TF motif enrichment analysis by FIMO tool[43] of MEME suite[44] was used to cross-validate the predicted activated targets of HDAC1.

A similar method was applied to identify the potential target genes/exons of PTBP1 exon 9 based on RNA-binding motif analysis. Firstly, AS events that significantly correlated with PTBP1 exon 9 were selected. Secondly, exons targeted by PTBP1 through motif recognition were kept. Finally, we focused on the AS events that are significantly associated with HSCR disease. RNA-binding motif enrichment analysis was also performed by the FIMO tool[43] of MEME suite[44].

## Alternative splicing (AS) analysis

The number of splices is the total number of splicing events in each read, which is generated by STAR software[45] during read alignment. AS frequency is determined by the number of splices divided by uniquely mapped reads number in each cell, which is a normalized value to evaluate the relative AS level of cells. SUPPA (version 2.3)[29] was used to perform the alternative splicing analysis. Firstly, transcript events and local alternative splicing (AS) events were generated from the ENSEMBL GRCh38 genome annotation (gtf) file. Seven types of AS events were identified by SUPPA, including skipping exon (SE, also known as cassettes), alternative 5′/3′ splice sites (A5/A3, also known as donors/acceptor), mutually exclusive exons (MX), retained intron (RI), and alternative first/last exons (AF/AL). Event count indicates the total number of AS events in a cell. Secondly, the transcript and local AS event inclusion levels (PSI, percent spliced in) were quantified from multiple samples. Lastly, differential splicing for AS events and differential transcript usage between HSCR and control iPSC-derived ENCCs were calculated based on single-cell level PSI value. Clustering and PCA analyses of cells based on PSI value were also performed by the Seurat package. The weight of AS events was derived from PCA analysis, indicating which AS events contribute most to specific PCs.

## Protein–protein interaction (PPI) network analysis

The PPI among the core gene set and HSCR-associated DEGs were obtained from STRING (v11)[27]. We included only the interactions based on manual curation or experimental evidence with a combined score >0.4. Cytoscape[46] was used to plot the network.

## Statistics and reproducibility

The differences among multiple treatment groups were analyzed with a two-sided unpaired Student's *t*-test or one-way analysis of variance followed by a Tukey post-test using GraphPad Prism 9 (GraphPad Software). A *P* value less than 0.05 was interpreted to represent a statistically significant difference. All experiments were replicated at least three times, and data are shown as means with SEM. No specific randomization or blinding protocols were used.

## Reporting summary

Further information on research design is available in the Nature Portfolio Reporting Summary linked to this article.

# Data availability

Raw sequencing data are available in the Sequence Read Archive (SRA) at the NCBI Center with the accession number PRJNA784249 (including one mouse cardiac NCCs and nine humans iPSC-derived Bio-Samples). The processed scRNA-seq datasets are available at https://doi.org/10.5281/zenodo.6104610. Public datasets that are used for comparison in this article can be downloaded from https://doi.org/10.5281/zenodo.5588286 (mouse enteric NCCs, Fig. 1k), http://atlas.gs.washington.edu/mouse-rna (mouse NCCs on enteric and peripheral nervous system, Fig. 1k), and GSE149524 (ENS cells at E15.5, E18.5, and P21). Reference ENSEMBL GRCh38 (release 90) can be downloaded from https://ftp.ensembl.org/pub/release-90/fasta/homo_sapiens/. A

reporting summary for this Article is available as a Supplementary Information file. Source data are provided with this paper.

## Code availability

All annotated codes showing key steps of the analysis are available at https://github.com/ellylab/HSCR-scRNAseq-paper and https://doi.org/10.5281/zenodo.7710574[47].

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

## Acknowledgements

Confocal imaging and RNA sequencing were performed using equipment maintained by the Li Ka Shing Faculty of Medicine Faculty Core and the Centre for PanorOmic Sciences, the University of Hong Kong, respectively. The work described in this paper was substantially supported by the HMRF grants (Project nos: 06173306 and 08192786) from the Health Department and the General Research Fund (GRF: HKU 17108019 and 17101320) from the Research Grant Council of Hong Kong Special Administrative Region, China Hong Kong and a seed funding from Li Dak Sum Research Centre, University of Hong Kong to E.S.-W.N., and a research grant from the Innovative Technology Fund (UIM/299) to P.K.-H.T. and E.S.-W.N.

## Author contributions

K.N.-C.L., S.-T.L., F.P.-L.L., and P.-L.L. established iPSC lines and performed in vitro functional assays. Z.L. and P.C.S. performed and supervised the bioinformatics analyses. E.S.-W.N., C.-C.H., and P.C.S. supervised the project and prepared the manuscript. P.K.-H.T., K.K.-Y.W., and P.H.-Y.C. contributed to the recruitment of patients and provided clinical information. M.-M.G.-B. performed the genetic characterization of the recruited patients and provided genetic data.

## Competing interests

The authors declare no competing interests.
