## [Peer Review File · Nature Communications]

Transcriptomics of Hirschsprung disease patient-derived enteric neural crest cells reveals a role for oxidative phosphorylationReviewers' Comments:

Reviewer #1:

Remarks to the Author:

Although gene expression of HSCR patient tissues have been previously analyzed, these data might reflect the end-stage disease progression, providing limited information about the mechanisms of disease onset. To overcome this limitation, Li et al., utilized iPSC-based models and performed single cell analysis. Their comprehensive bioinformatic analyses collectively provide new mechanistic insight into HDAC1/RNA splicing factors/Hh signaling-mediated OXPHOS involved in HSCR pathogenesis. Additional functional studies would validate some of their claims and further define the relationship of these key regulators.

Patient tissue analysis was provided only as a table summary. Representative histology and some neuronal marker analyses also should be provided.

The authors claimed Cluster 1 as ENCC. However, this cluster seems to highly express glial cell markers. Comparing their data with the published mature enteric glial cell gene expression might provide stronger evidence in support of their claim.

The authors stated (line 421-423) "Therefore, the reduced PKM1 activity in HSCR, at least partly due to the inclusion of PTBP1 exon 9, may perturb OXPHOS, leading to the retarded neuronal differentiation of ENCCs." This claim should be further examined by functional studies through the overexpression and/or knockdown of PKM1.

Interestingly, Hh signaling is known to regulate HDAC activity (PMID 23951168), as shown in Fig 3b. Their relationship could be further investigated using Hh and HDAC inhibitors and agonists in iPSC models. Could HDAC inhibition also influence the expression of RNA splicing factors and OXPHOS pathway genes? As the hierarchical relationship among Hh signaling, HDAC and RNA splicing factors is further defined, the authors should provide a schematic diagram describing how Hh signaling, HDAC, and RNA splicing factors regulate neuronal differentiation and OXPHOS.

Reviewer #2:

Remarks to the Author:

In this study, Li et. al. describe molecular and functional deficits in enteric neural crest cells differentiated from Hirschsprung (HSCR) induced pluripotent stem cells (iPSCs). Using a combination of single cell RNA sequencing and previously established differentiation techniques, the authors identify molecular features in in vitro patient-iPSC derived enteric neural crest cells that correlate with clinical disease severity in patients. Consistent with previous findings, HSCR enteric neural crest cells exhibit reduced migration and neurogenesis potential and their molecular profile is consistent with lower neurogenic capability, altered RNA splicing and energy metabolism. They follow up to describes each of these features in greater detail. Finally, they propose using SHH agonists to rescue these disease phenotypes in HSCR iPSC-derived enteric neural crest cells. While the overall profiling data and analyses are valuable for the field, the manuscript needs to be improved by including additional information, performing further experiments and analyses as outlined below.

General comments:

1. In many instances the description of the performed analyses is vague and broad. There needs to be more specific information about what datasets were used, how they were processed and what was measured. Similarly, the figure legends tend to be very unhelpful in providing information about data presentation, parameters measured, etc. Axes labels of graphs tend to be vague. Number of technical and biological replicated not mentioned. More specific examples are provided in comments below.
2. In many cases correlations are claimed as causations, these wordings need to be fixed throughout

the manuscript. Examples are mentioned in comments below.

3. Figures 3-7 describe different components of HSCR phenotype that correlate in iPSC-derived ENCC. It should be made clear that correlation between these aberrations do not necessarily mean causation. There are instances in which the authors leap into commenting on causal relationship between alternative splicing, neurogenesis, SHH signaling and energy metabolism without substantial evidence to back these claims.

Specific comments:

1. Data needs to be provided around the axial identity of the differentiated ENCCs from different iPSC lines. It is important to assess whether the HSCR mutations impact the axial identity of the NCCs (e.g. vagal vs cranial) using HOX gene expression as a readout.
2. Similar to point 1, it is important to provide data confirming that the differentiated neurons are in fact enteric neurons (as opposed to other NCC-derived neurons such as sensory neurons). This can be established by quantitative analysis lineage specific marker expression.
3. Information about the differentiation stage (day of differentiation) should be provided for NCCs and neurons analyzed or evaluated for each experiment.
4. Line 159, reference missing for the statement about E12.5 mouse embryos.
5. It is unclear if in Figure 1 panels j and k the analysis is only performed on WT hPSC-derived NC or the merged WT and HSCR dataset.
6. In lines 199-202- the interpretation about the number of DEGs is inaccurate. The larger number of altered genes doesn't necessarily imply higher severity. Claim needs to be corrected.
7. In line 262, more detailed information needs to be included about the model. The analysis should be further explained.
8. Line 269 should referred to Fig 3d not 4d.
9. In line 289- 293, the analysis description is vague? Was binding prediction used to filter the data? No data provided on binding prediction in the figure.
10. The main disease relevant phenotype in HSCR that link to the pathophysiology is migration defects (that lead to aganglionosis in newborns). It would be helpful assess migration assays in addition to neuronal differentiation in rescue experiments in figures 3, 6 and 7.
11. The splicing analysis in figure 4 is vaguely described in the results section (lines 337-367). Further details are necessary to assess rigor and accuracy. For example, the description provided for figure 4h is very vague. It is unclear what the x and y axes show. Figure legends overall contain minimal information and lack many crucial details.
12. In line 378-380, it is unclear how the motif analysis and filtering was done.
13. The analysis done in Figure 5d needs to be explained further.
14. Analyses in figure 5 are highly correlational. The wording should be clear to avoid any unsubstantiated implications of causality.
15. In seahorse experiments, details are missing regarding the experiment design, number of replicates, what age cells were tested and what the graphs show.
16. More details necessary regarding the mitochondrial assays. What is GW9508? What is MTT?
17. The justification for testing SHH agonists needs to be explained further.

Reviewer #3:

Remarks to the Author:

The manuscript by Zhixin Li et al "Hedgehog-induced-oxidative-phosphorylation rescues neuronal differentiation defect of human enteric neural crest cells underlying Hirschsprung disease" describes a study carried out on human iPSC-derived enteric neural crest cell model (HSCR-ENCC). The authors aim to identify major changes in molecular pathways and networks associated with disease severity by high-resolution transcriptomic analysis and validate the findings with functional assays.

In a nutshell, the study reveals three major findings. First off, identification of a core gene set of 118

genes commonly dysregulated in all HSCR-ENCC lines, and HDAC1 as a master regulator of these genes. Importantly that increased nuclear localization and transactivation of HDAC1 in HSCR-ENCC is associated with suppressed mitochondrial activity and oxidative phosphorylation. The authors suggest that the elevated transactivation activity of HDAC might be an underlying cause of HSCR.

Secondly, upon the global analysis of splicing among different HSCR-ENCC, association of PTBP1 exon 9 inclusion with the disrupted cellular metabolism in HSCR-ENCC derived from the long segment of a colon is demonstrated.

Finally, basing on the findings above, which indicate a functional link with altered energy metabolism, the authors demonstrate the essential role of Hedgehog signaling mediated fatty acid oxidation and consecutive oxidative phosphorylation in the extent of neuronal differentiation capacity for ENCC.

Concerns/suggestions:

1. The authors claim the provision of "a comprehensive biological picture of the core molecular mechanisms underlying HSCR heterogeneity". Concurring with this in principle, more elaborated analysis of the cross talk between the core molecular mechanisms is needed. For instance, it remains unclear whether proposed approach for restoration of ENCC functions in patients by boosting Hedgehog pathway or fatty acid oxidation is concomitant with the normalization of aberrant HDAC1 activity. In this respect the manuscript would benefit from a graphical abstract depicting the novel findings regarding the pathways in pathogenesis of Hirschsprung disease and their interplay.

2. Lines 330-334: The statement that aberrantly high HDAC1 transactivation activity represents a common cause of neuronal differentiation defect associated with HSCR disease is overstatement. Evidently, inhibition of HDAC1 activity favours the neuronal lineage differentiation.

3. Basic characterization of the iPSC lines is missing. Also, other important details of the iPSCs is missing: how many line per each donor were used (to avoid the impact of line-line variation and understanding the reliability of the lines). Were the lines differentiated in parallel? Have the assays, stainings and blots been replicated 2-3 times?

4. The seahorse data (shown in figs 5-6) should be analysed per donor/line, now the meaning of the dots is unclear. The same is true for ATP, lactate, pyruvate, MTT and mitotracker data which now seem to represent pooled data, still there is considerable overlap in seahorse and mitotracker values between the groups. Based on the current data, it is possible that many diseased lines did not actually differ from controls.

Reviewer #4:

Remarks to the Author:

This manuscript presents single-cell transcriptomic analyses of enteric neural crest cells in both healthy and disease states. It is largely a descriptive study, but the authors make many mechanistic and causal claims that are unsupported by the descriptive data presented. A descriptive analysis of gene expression would be acceptable and appropriate, but making unsupported mechanistic and causal claims from descriptive and correlative data is not.

A few examples of this are provided here. (This is not an exhaustive list.)

1. It is claimed in the abstract and elsewhere that a HDAC1 was shown to be a "master regulator" of a "core gene set of 118 genes." However, no causal evidence is provided. In fact, no clear evidence that HDAC1 regulates these genes is given. The evidence presented is that HDAC1 expression is correlated with the expression of some of these genes, and that a few of the genes have HDAC1 predicted motifs located nearby. This descriptive evidence is nowhere near sufficient to match the broad, causal claims of HDAC1 as a master regulator.

2. It is claimed that there is a "highly confident causal regulation between" PTBP1 and PKM. However,

the only evidence provided is a (modest) correlation between PTBP1 and PKM. This is evidence of a correlation, but certainly not a “highly confident causal regulation.”

3. It is claimed that “PKM1 activity.... due to the inclusion of ptbp1 exon 9...” leads to perturbed OXPHOS, “leading to the retarded neuronal differentiation.” However, none of these links were causally established by evidence in the manuscript. It was not demonstrated that PKM1 was causally regulated by ptbp1 (only correlation). It was not shown that PKM1 expression levels are directly the cause of OXPHOS perturbation (again, only correlation). It was also not demonstrated here (Fig 5) that OXPHOS perturbation is causally linked to neuronal differentiation defects.

These are just a few examples of over-exaggerated causal claims that are not supported by the descriptive, correlative data. Many more examples could be provided. This manuscript would be well served by a wholesale alteration from making causal, mechanistic claims into a more narrowly-focused descriptive analysis of gene expression.

Reviewer #5:

Remarks to the Author:

In their study, Li et al. generated seven patient-derived iPSC-based cell lines to investigate the cellular/molecular bases of Hirschsprung disease (HD). First, they characterized the lines by single cell RNAseq and bioinformatics and by cell biological assays (Figs. 1-2). Then, based on further analysis, the Authors suggest three key findings: (1) aberrantly high HDAC1 transactivation activity is a common cause of neuronal differentiation effects associated with the disease (Fig. 3), (2) reduced PKM1 activity at least partly due to the inclusion of exon 9 in PTBP1 may perturb OXPHOS leading to retarded neuronal differentiation (Figs. 5-6), and (3) Hedgehog-induced OXPHOS can enhance the otherwise impaired survival and differentiation capacity in HD patient-derived cells lines (Figs. 6-7).

While this is an appealing set of conclusions, the manuscript does not generate strong confidence in the results. Despite each of the key findings seeming to suggest some sort of insight into impaired differentiation and/or survival that is associated with the disease, different parts of the study appear to be only loosely connected. In addition, many of the arguments/conclusions lack solid experimental support. Overall, the manuscript gives the impression of multiple undeveloped observations being packaged together. The Hedgehog-induced rescue indicated in the title is only one part of the manuscript and does not require or depend on any other analysis (e.g. HDAC1, splicing, PTBP1, PKM). Perhaps the Authors could focus on this final line of investigation (the Hedgehog-induced rescue) and develop it more fully.

Critique

1. Controls

The Authors argue later in the paper that there are critical transcriptomic differences between control and patient cell lines. However, in Fig. 1h, the Authors show a tSNE plot in which the majority of cells, including control and patient-derived, are grouped together. While this does not exclude the possibility of differences, this suggests patient-derived cells are transcriptomically not very different from controls.

The plot also highlights that the two control lines are just as different from each other as they are from other patient-derived cells. Such a pronounced and potentially uncontrolled variability between controls would fundamentally compromise any subsequent analysis.

The origin of controls further complicates this issue. The Authors generated their patient-derived cell

lines from skin biopsies. By contrast, one of the controls was derived from urine-derived cells (UE02302, male; as described in Methods), whereas the other from fetal fibroblasts (IMR90, female; as described in the original article, Yu et al., Science, 2007).

Clearly, the best practice would be to include multiple control lines in the study that fully match the patient-derived protocols. If this is not possible (e.g. because of subject recruitment constraints), the Authors at least should provide a detailed analysis of the controls (including differential gene expression analysis) showing that they can be appropriately used for this study, and control for their variability in other analyses.

Questions regarding the use of controls also emerge later in the study. At least, in Figs. 5a, S1a and S6a, figure displays suggest that patient-derived data were only compared to UE02302 as control, whereas in Fig. S7, they were only compared to IMR90. In Figs. 2g-j, 3e, 5e-j, 6a-n, S4 only „control“ is displayed, without specifying whether IMR90 or UE02302 or both were used. The Authors should show how their results change depending on which control(s) they use, and clearly describe and explain the use of controls in each legend and the text.

2. HDAC1

The Authors identified multiple genes whose expression changed along the severity axis (this axis is sometimes called disease axis, e.g. Fig. 2d-e versus Fig. 3d, which is additionally confusing). Of these, they quickly singled out HDAC1 as the most relevant hub gene, because PPI network analysis suggested its link to NC development and neurogenesis. The relevances/PPIs of other genes were not discussed. (Peculiarly, the expression level of HDAC1 also appears to be negative in low severity cases, Fig. 3D. Whether this is an error or consequence of an unusual and unexplained scale transformation, it certainly does not help to generate confidence in the results. - The units of expression and other parameters should be always clearly displayed in all plots.)

To support the focus on HDAC1, the Authors showed that (i) another core set genes (whose expression also changed) were upregulated in patient-derived cell lines in a similar fashion to HDAC1, (ii) some of these are predicted to be „high-confidence“ direct targets of HDAC1, and (iii) the presumed gene regulatory network of HDAC1 was slightly activated in low severity disease state but highly activated in severe disease state. Based on these arguments, they suggested that the upregulated expression and elevated transcription activity of HDAC1 would be an underlying cause of the disease.

However, Western blot showed that the total protein expression of HDAC1 was not different between controls and patient-derived cells. These data is only shown in supplementary Fig. S6, but they should be clearly in main figure. (The Western was done only on 4 out of 7 patient-derived lines, and although the main text provides some reasons why this was the case, the results should be shown for all lines.)

As an alternative hypothesis, the Authors then speculate that the nuclear localization of HDAC1 was enhanced, which lead to increased the transactivation activity of HDAC1 in patient-derived cell lines. This speculation is based on the observation that cytosolic HDAC1 signal was not visible in the patient-derived cell lines, only in controls (Fig. 3h). However, this observation does not directly show that nuclear HDAC1 localization was higher in patient-derived cells (even if the Westerns looked similar). It is also unclear if the lack of cytosolic HDAC1 in the patient-derived cell lines was related to the disease, or appeared because of other reasons, e.g. due to the use of different control lines (see first point).

Using an in vitro differentiation assay, the Authors showed that the application of valproic acid (inhibitor of HDAC1 among others) significantly enhanced NF signal in patient-derived cell lines (also

in one of the controls, but not in the other), indicative of enhanced differentiation into a neuronal lineage. However, this experiment does not make a strong case for HDAC1, given that valproic acid could act through drug effectors other than HDAC1.

To conclusively show the involvement of HDAC1 in the disease, the Authors need to gather more solid evidence. For example, they could evaluate the presence/absence of cytosolic signal in matching controls, use quantitative measurements to evaluate the nuclear localization signals, and employ molecular-genetic perturbation experiments to target HDAC1 and confirm their preliminary results using valproic acid (e.g. shRNA or similar approach in patient-derived lines, overexpression in controls).

3. Splicing, PTPB1, PKM, OXPHOS, Hedgehog

Unrelated to HDAC1, the Authors went on to analyze splicing events globally, because the pathway score for RNA splicing was significantly higher in patient-derived cell lines with advanced disease severity (Fig. 2i).

Consistent with this observation, they found enrichment of several splicing factors in patient-driven cell lines. Intriguingly, the enrichment pattern was sharply split between cell lines with higher (enriched) and lower severity scores (not enriched) (Fig. 4a).

Then, they also showed data suggesting that splicing events were more frequent in high severity cases (Fig. 4b), compared the occurrence of different splicing events (Fig. 4c; although it is unclear from the legend and text which comparison do these p values refer to; #17 to controls?, #1 to controls?, #17 to controls?), and performed other bioinformatic analyses (Fig. 4d-e). Finally, they quantified the weight of AS events (Fig. 4f).

Unfortunately, the manuscript lacks sufficient technical descriptions on the splicing calculations, making it hard to judge what the data actually means. The Authors should clearly define (1) number of splices, (2) event count, (3) AS frequency, (4) transcriptional level versus post-transcriptional level, (5) weight of AS events - even if these were output from other softwares they used. For example, while top ranking of PTPB1[1] in the weight of AS events plot is a critical argument to follow up on this variant, the calculation method is not explained neither in this nor in the original paper (42) they cite.

Independently, the reason to choose PTPB1[1] (inclusion of exon 9) as opposed to other PTBP1 splicing events or other upregulated factors appears somewhat arbitrary. For example, the significant intron retention seen in PTPB1 (PTPB1[3]) would likely render the protein dysfunctional, and irrespective of inclusion of exon 9 could have a more significant contribution to the disease. Overall, the path of reasoning from PTPB1[1] to PKM (one target of PTPB1) and through that to OXPHOS, on which the Authors had data, feels constructed.

In any case, based on their data, the Authors suggest that increased expression of PTBP1 and/or increased inclusion of exon 9 in PTBP1 would result in decreased PKM1 (also decreased exon 9 inclusion in PKM1) and increased PKM2 (also increased exon 10 inclusion in PKM2) production. However, this hypothesis itself is not new and the manuscript should include references/discussion to/on previous research on this.

Using RT-qPCR, the Authors show that PKM1 levels were significantly decreased in all patient-derived cell lines and that there were no significant PKM2 changes in either of the lines (Fig. 5f). While this finding somewhat confirms their hypothesis (i.e. PKM1 downregulation in high severity cases), it also raises the question why PKM1 was significantly downregulated in low severity cases, in which PTPB1 was neither upregulated (Fig. 4a) nor its splicing has changed (Fig. 5a). Similarly the question of why alternative splicing is impacted so strongly in severe cases, but is unchanged in less severe cases,

while their previous analyses (Figs. 2 and 3) showed more gradual gene expression changes along the severity axis remains unresolved.

Irrespective how the Authors arrived to OXPHOS, the OCR analysis (Fig. 5g-j) reveals a potentially interesting phenotype in patient-derived cell lines. However, their conclusion this phenotype to be due to „reduced PKM1 activity in HSCR, at least partly due to the inclusion of PTBP1 exon 9, may perturb OXPHOS, leading to the retarded neuronal differentiation of ENCCs“ (lines 421-423) is not backed up by experimental data and speculative at most. In order to support that PTBP1 or PKM1 indeed played a role in this phenotype, the Authors should at least perform PTBP1[1] and PKM1 overexpression experiments and see if those show a rescue in the OCR assay.

Finally, the Authors suddenly jump onto investigating Hedgehog signaling. They even introduce a completely new sequencing data set (Fig. 6a-b), the origin of which is not clarified. Is this coming from the Author's previous study? If yes, does it represent mouse or human data? The Authors should clearly describe (supported by detailed analyses comparing controls) how these data relates to the data they analyzed in their manuscript until this point. Regarding Hedgehog, they then show analyses which are potentially interesting, but unrelated to any of their above analyses; the relationship between this part of the study and HDAC1, splicing, and PTPB1 is not recognizable.

Our detailed point-to-point responses to the reviewers comments:

Reviewer #1:

Although gene expression of HSCR patient tissues have been previously analyzed, these data might reflect the end-stage disease progression, providing limited information about the mechanisms of disease onset. To overcome this limitation, Li et al., utilized iPSC-based models and performed single cell analysis. Their comprehensive bioinformatic analyses collectively provide new mechanistic insight into HDAC1/RNA splicing factors/Hh signaling-mediated OXPPOS involved in HSCR pathogenesis. Additional functional studies would validate some of their claims and further define the relationship of these key regulators.

1. Patient tissue analysis was provided only as a table summary. Representative histology and some neuronal marker analyses also should be provided.

Ans: Since the establishment of enteric aganglionosis as the primary pathology of Hirschsprung disease more than 70 years ago, the diagnosis of Hirschsprung disease has been repeatedly documented and reviewed in textbooks and publications (Ambartsumyan et al., 2020; Kapur, 2009). Diagnosis of Hirschsprung's disease begins with clinical evaluation by a specialist pediatric surgeon/paediatrician and is completed with pathological confirmation of absence of ganglion cells in appropriately performed rectal biopsy specimens. During surgery, removal of diseased bowel provides the opportunity for the pathologist to both confirm aganglionosis of distal bowel, and to evaluate the length of aganglionic bowel, allowing classification into short, long or ultra-long segment aganglionosis. The gold standard of diagnosis of Hirschsprung disease is the pathology report describing details of the histological/histochemical/ immunohistochemical abnormalities of the enteric nervous system. The immunohistochemistry images of random tissue samples, in many cases, are confusing /misleading. These data should be read and interpreted by the pathologists with the support of clinical evaluation. Therefore, including the immunohistochemistry data from different regions of the bowels may not be appropriate and probably misleading. Thus, the full pathology reports are provided to support the diagnosis. The attached pathology reports are official documents issued by accredited pathologists from an internationally recognised institution, representing the final verdict on the pathology of the patients.

It should be pointed out that while the diagnosis of Hirschsprung disease is relatively straightforward in the majority of cases, there could be difficulties in achieving the final diagnosis in some patients, who will require repeated biopsies (De La Torre and Wehrli, 2019). All our patients subsequently had surgery providing post resection specimens for final pathology.

Hirschsprung disease remains a challenging disease, requiring relevant clinical and pathological expertise for diagnosis and characterization. The authors include experienced specialist pediatric surgeons who have been primary carers of the patients studied. The final pathology reports are independent conclusive evidence of the enteric nervous system abnormalities of each patient and as such are superior to selected /representative neuronal images which themselves have no novelty value in this paper.

Please note that we need to protect the privacy of the patients, the pathology reports cannot be published or included in the manuscript, they would only be used as the supporting documents for the review process. The names and identity numbers of the patients are hidden.

References:

Kapur, R. P. Practical pathology and genetics of Hirschsprung's disease. *Semin Pediatr Surg* **18**, 212-223, doi:10.1053/j.sempedsurg.2009.07.003 (2009).

Ambartsumyan, L., Smith, C. & Kapur, R. P. Diagnosis of Hirschsprung Disease. *Pediatr Dev Pathol* **23**, 8-22, doi:10.1177/1093526619892351 (2020).

De La Torre, L. & Wehrli, L. A. Error traps and culture of safety in Hirschsprung disease. *Semin Pediatr Surg* **28**, 151-159, doi:10.1053/j.sempedsurg.2019.04.013 (2019).

2. The authors claimed Cluster 1 as ENCC. However, this cluster seems to highly express glial cell markers. Comparing their data with the published mature enteric glial cell gene expression might provide stronger evidence in support of their claim.

Ans: It was reported that *SOX10*, *ERBB3* and *PLP1* are highly expressed in both undifferentiated bipotent ENS progenitors (namely BPs) and glial cells as described in previously published paper (Lasrado et al., 2017). Consistently, our hiPSC-derived ENCCs express high levels of these markers. Nevertheless, it doesn't mean they are glial cells/glial progenitors. When we compared our iPSC derived ENCCs to mouse ENCCs and their derivatives at E13.0 from the published dataset (Lasrado et al., 2017), the molecular signature of iPSC derived ENCCs (iPSC-ENCCs) are more close to the bipotent progenitors (BP) than the glial (GP) and neuronal (NP).

Correlation heatmap shows the similarities between hiPSC, iPSC-derived ENCCs, and mouse enteric bipotent progenitors (BP), glial progenitor (GP), neuronal progenitors (mNP) and enteric mesothelial fibroblast (ENMFb).

In concordant with this observation, when we compared the iPSC-ENCCs to mouse ENCCs and their derivatives at E15.5, E18.5 and P21 from another published dataset (Morarach et al., 2021), iPSC-ENCCs in Cluster 1 also show the highest similarity with the bipotent (mBP), rather than glial (mGP) progenitors.

Correlation heatmap shows the similarities between hiPSC, iPSC-derived ENCCs, and mouse enteric bipotent progenitors (mBP), glial progenitor (mGP), and mature glial cells (mG), neuronal progenitors (mNP) and mature neurons (mN).

All these data suggest that iPSC-ENCCs are very similar to mouse enteric BP (based on their transcriptomic profiles and their functional properties).

Reference:

Lasrado, R. *et al.* Lineage-dependent spatial and functional organization of the mammalian enteric nervous system. *Science* **356**, 722-726 (2017).
<https://doi.org/doi:10.1126/science.aam7511>

Morarach, K., Mikhailova, A., Knoflach, V., Memic, F., Kumar, R., Li, W., Ernfors, P., and Marklund, U. (2021). Diversification of molecularly defined myenteric neuron classes revealed by single-cell RNA sequencing. *Nature Neuroscience* *24*, 34-46. 10.1038/s41593-020-00736-x.

3. The authors stated (line 421-423) “Therefore, the reduced PKMI activity in HSCR, at least partly due to the inclusion of PTBP1 exon 9, may perturb OXPHOS, leading to the retarded neuronal differentiation of ENCCs.” This claim should be further examined by functional studies through the overexpression and/or knockdown of PKMI.

Ans: We have included the new data to show that the down-regulation of *PKMI* perturbs the neuronal differentiation of control-ENCCs, while the up-regulation of *PKMI* rescues the differentiation defect of HSCR-ENCCs. The links between *PKMI/PKM2* levels with cell metabolism were also demonstrated (Fig. 8h-l).

4. Interestingly, Hh signaling is known to regulate HDAC activity (PMID 23951168), as shown in Fig 3b. Their relationship could be further investigated using Hh and HDAC inhibitors and agonists in iPSC models. Could HDAC inhibition also influence the expression of RNA splicing factors and OXPHOS pathway genes? As the hierarchical relationship among Hh signaling, HDAC and RNA splicing factors is further defined, the authors should provide a schematic diagram describing how Hh signaling, HDAC, and RNA splicing factors regulate neuronal differentiation and OXPHOS.

Ans: The protein-protein interaction (PPI) data shown in Fig. 3b was performed with the core gene set which are commonly dysregulated in ALL HSCR cases and the changes in their expression are associated with disease severity. These core gene sets are implicated in four key pathways and are associated with HSCR or NC development. Given that the dysregulation of OXPHOS is particularly enriched in L-HSCR, OXPHOS pathway associated genes were not included in Fig. 3b. On the other hand, the PPI data shown in Fig. 3b may be considered as the regulatory networks priming the individuals to HSCR, particularly associated with the disease severity. Dysregulation of OXPHOS, via AS (which would be HDAC1 dependent or independent), further interrupts the formation of enteric neurons, leading to more severe form of HSCR (L-HSCR).

There are canonical (Gli-dependent) and non-canonical (Gli-independent) Hedgehog pathways. HDAC1 bind to Gli3 to repress enhancers to prevent premature activation of hedgehog target genes (Lex et al., 2020; Lex et al., 2022). Even though Gli3 usually works as

repressor, the aberrant Gli3 activator function has been associated with HSCR pathogenesis (Liu et al., 2015; Sribudiani et al., 2018). On the other hand, Hh-mediated OXPHOS is regulated via the non-canonical Gli-independent pathway (Teperino et al., 2012) albeit the prolonged SAG treatment will also induce Gli-target genes. Therefore, the PPI data shown in figure 3b only refers to the direct interactions among GLI3 and HDAC1. Dysregulation of GLI-HDAC1 repression causes the premature activation of Hedgehog pathway, and the premature activation of Hh is associated with HSCR pathogenesis (Liu *et al.*, 2015; Ngan et al., 2011; Sribudiani *et al.*, 2018).

Based on the data from transcriptomic and genetic analyses, neuronal differentiation defect of ENCCs is one of the common problems found in all HSCR patients (S/L-HSCR). Therefore, it is conceivable that rescuing the neuronal differentiation of ENCCs represents a possible treatment strategy. As stated in page 14, line 438-442, one of our recent studies revealed that the addition of SAG (Smoothed agonist) during derivation of ENCCs from hiPSC primes the ENCCs to the higher cell state, favoring the subsequent neuronal differentiation (Lau et al., 2019). “Subsequent analysis of the transcriptomes of SAG-primed hiPSC-ENCCs at single cell level further revealed that the genes implicated in glycolytic and fatty acid oxidation (FAO) pathways are robustly upregulated (Fig. 6a & b)”. Therefore, we examined whether activation of the non-canonical (GLI-independent) arm of Hedgehog pathway (glycolytic and FAO pathways) can promote the neurogenic differentiation of hiPSC-ENCCs and data are shown in Fig. 6 and Fig. 7.

In sum, from this study, only GLI3 from Hedgehog pathway was found to be associated with HSCR pathogenesis (one of the core-genes), where PPI analysis suggested a direct interaction between GLI3 and HDAC1. On the other hand, Hedgehog activates FAO pathway through the non-canonical arm (GLI-independent) of Hedgehog pathway, promotes the neurogenic lineage differentiation of hiPSC-ENCCs.

Our data (fig. 2g) indicate that many HDAC1 targets are implicated in RNA splicing. Concordantly, it has been reported that the activity of histone deacetylases (HDACs) influences splice site selection. HDAC inhibition induced histone H4 acetylation and increased RNA Polymerase II (Pol II) processivity along an alternatively spliced element (Hnilicová et al., 2011). A recent review have summarized that HDACs interact with spliceosomal and ribonucleoprotein complexes, actively control the acetylation states of splicing-associated histone marks and splicing factors, and thereby unexpectedly could modulate splicing (Rahhal and Seto, 2019).

Similarly, it has been shown that Hdac1/Hdac2 promotes the transition to OXPHOS by enforcing transcriptional fidelity of metabolic gene programs in mouse cardiac development (Milestone et al., 2020).

To delineate the roles of HDAC1 in various cellular processes, a more comprehensive study is required to carefully analyse the epigenomes of the cells, but it is out of the scope of the current study. In the current study, we aimed to present an overview of the biological pathways associated with HSCR pathogenesis. Even though our data suggest that HDAC1 is a master regulator underlying various HSCR-associated pathways, we do not aim to focus on how HDAC1 interacts with the other pathways. Therefore, instead of providing a diagram showing the hierarchical relationship among these genes/pathways with HDAC1, we prepared a diagram summarizing the pathways associated with HSCR pathogenesis and severity as shown below. It is included in the revised manuscript.

Fig. 8. Schematic summaries the key molecular and cellular processes associated with HSCR pathogenesis and severity. A core gene set associated with the severity of HSCR was identified. Additional alternations in genes implicated in RNA splicing and oxidative phosphorylation (OXPHOS) were found in L-HSCR cases. High HDAC1 activity is associated with the defective neuronal differentiation as observed in HSCR-ENCCs. Only the representative genes/pathways are shown.

References:

- Lex, R. K. *et al.* GLI transcriptional repression is inert prior to Hedgehog pathway activation. *Nat Commun* **13**, 808 (2022). <https://doi.org/10.1038/s41467-022-28485-4>
- Lex, R. K. *et al.* GLI transcriptional repression regulates tissue-specific enhancer activity in response to Hedgehog signaling. *Elife* **9** (2020). <https://doi.org/10.7554/eLife.50670>
- Sribudiani, Y. *et al.* Identification of Variants in RET and IHH Pathway Members in a Large Family With History of Hirschsprung Disease. *Gastroenterology* **155**, 118-129.e116 (2018). <https://doi.org/10.1053/j.gastro.2018.03.034>
- Liu, J. A. *et al.* Identification of GLI Mutations in Patients with Hirschsprung Disease That Disrupt Enteric Nervous System Development in Mice. *Gastroenterology* (2015). <https://doi.org/10.1053/j.gastro.2015.07.060>
- Ngan, E.S., Garcia-Barcelo, M.M., Yip, B.H., Poon, H.C., Lau, S.T., Kwok, C.K., Sat, E., Sham, M.H., Wong, K.K., Wainwright, B.J., et al. (2011). Hedgehog/Notch-induced premature gliogenesis represents a new disease mechanism for Hirschsprung disease in mice and humans. *J Clin Invest* *121*, 3467-3478. [10.1172/JCI43737](https://doi.org/10.1172/JCI43737).
- Lau, S.T., Li, Z., Pui-Ling Lai, F., Nga-Chu Lui, K., Li, P., Munera, J.O., Pan, G., Mahe, M.M., Hui, C.C., Wells, J.M., and Ngan, E.S. (2019). Activation of Hedgehog Signaling Promotes Development of Mouse and Human Enteric Neural Crest Cells, Based on Single-Cell Transcriptome Analyses. *Gastroenterology* *157*, 1556-1571 e1555. [10.1053/j.gastro.2019.08.019](https://doi.org/10.1053/j.gastro.2019.08.019).
- Garcia-Barcelo, M.M., Tang, C.S., Ngan, E.S., Lui, V.C., Chen, Y., So, M.T., Leon, T.Y., Miao, X.P., Shum, C.K., Liu, F.Q., et al. (2009). Genome-wide association study identifies NRG1 as a susceptibility locus for Hirschsprung's disease. *Proc Natl Acad Sci U S A* *106*, 2694-2699. [10.1073/pnas.0809630105](https://doi.org/10.1073/pnas.0809630105).
- Gui, H., Schriemer, D., Cheng, W.W., Chauhan, R.K., Antiñolo, G., Berrios, C., Bleda, M., Brooks, A.S., Brouwer, R.W.W., Burns, A.J., et al. (2017). Whole exome sequencing coupled with unbiased functional analysis reveals new Hirschsprung disease genes. *Genome Biology* *18*, 48. [10.1186/s13059-017-1174-6](https://doi.org/10.1186/s13059-017-1174-6).
- Jiang, Q., Arnold, S., Heanue, T., Kilambi, K.P., Doan, B., Kapoor, A., Ling, A.Y., Sosa, M.X., Guy, M., Jiang, Q., et al. (2015). Functional loss of semaphorin 3C and/or semaphorin 3D and

their epistatic interaction with *ret* are critical to Hirschsprung disease liability. *Am J Hum Genet* 96, 581-596. 10.1016/j.ajhg.2015.02.014.

Hnilicová, J., Hozeifi, S., Dušková, E., Icha, J., Tománková, T., and Staněk, D. (2011). Histone deacetylase activity modulates alternative splicing. *PLoS One* 6, e16727. 10.1371/journal.pone.0016727.

Rahhal, R., and Seto, E. (2019). Emerging roles of histone modifications and HDACs in RNA splicing. *Nucleic Acids Research* 47, 4911-4926. 10.1093/nar/gkz292.

Milstone, Z.J., Saheera, S., Bourke, L.M., Shpilka, T., Haynes, C.M., and Trivedi, C.M. (2020). Histone deacetylases 1 and 2 silence cryptic transcription to promote mitochondrial function during cardiogenesis. *Sci Adv* 6, eaax5150. 10.1126/sciadv.aax5150.

Reviewer #2:

In this study, Li et. al. describe molecular and functional deficits in enteric neural crest cells differentiated from Hirschsprung (HSCR) induced pluripotent stem cells (iPSCs). Using a combination of single cell RNA sequencing and previously established differentiation techniques, the authors identify molecular features in in vitro patient-iPSC derived enteric neural crest cells that correlate with clinical disease severity in patients. Consistent with previous findings, HSCR enteric neural crest cells exhibit reduced migration and neurogenesis potential and their molecular profile is consistent with lower neurogenic capability, altered RNA splicing and energy metabolism. They follow up to describes each of these features in greater detail. Finally, they propose using SHH agonists to rescue these disease phenotypes in HSCR iPSC-derived enteric neural crest cells. While the overall profiling data and analyses are valuable for the field, the manuscript needs to be improved by including additional information, performing further experiments and analyses as outlined below.

General comments:

1. In many instances the description of the performed analyses is vague and broad. There needs to be more specific information about what datasets were used, how they were processed and what was measured. Similarly, the figure legends tend to be very unhelpful in providing information about data presentation, parameters measured, etc. Axes labels of graphs tend to be vague. Number of technical and biological replicated not mentioned. More specific examples are provided in comments below.

Ans: We have revised the figure legends and text to include the experimental details. Please refer to our detailed responses below.

2. In many cases correlations are claimed as causations, these wordings need to be fixed throughout the manuscript. Examples are mentioned in comments below.

Ans: The first part of paper is mainly bioinformatic oriented and many conclusions were based on how statistically significant the associations are and based on the published information based on the literatures. To prevent confusing, we have revised the wordings accordingly.

3. Figures 3-7 describe different components of HSCR phenotype that correlate in iPSC-derived ENCC. It should be made clear that correlation between these aberrations do not necessarily mean causation. There are instances in which the authors leap into commenting on causal relationship between alternative splicing, neurogenesis, SHH signaling and energy metabolism without substantial evidence to back these claims.

Ans: We have provided additional data to support our claims on the causal links between alternative splicing for generation of PKM1 and PKM2, cellular metabolism and neurogenesis. The rescue experiments shown in Figures 6 and 7 also demonstrated the implications of the non-canonical Hedgehog signaling with cellular metabolism (OXPHOS) and neurogenesis in control and HSCR-ENCCs.

Specific comments:

1. Data needs to be provided around the axial identity of the differentiated ENCCs from

different iPSC lines. It is important to assess whether the HSCR mutations impact the axial identity of the NCCs (e.g. vagal vs cranial) using HOX gene expression as a readout.

Ans: ENCCs derived from all iPSC express various HOX genes (*HOXB2*, *HOXB3*, *HOXB4* and *HOXB5*) resembling vagal/enteric NCCs as shown in the plot below and as described previously (Lai, 2017).

Expression of *HOXB2*, *HOXB3*, *HOXB4*, *HOXB5* in ENCCs derived from control and HSCR-iPSCs.

Reference:

Lai, F.P., Lau, S.T., Wong, J.W.L., Gui, H., Wang, X.R., Zhou, T., Lai, W.H., Tse, H.F., Tam, P.K., Garcia-Barcelo, M.M., Ngan, E.S.W. (2017). Correction of Hirschsprung-associated mutations in human induced pluripotent stem cells, via CRISPR/Cas9, restores neural crest cell function. *Gastroenterology* 153, 139-153. 10.1053/j.gastro.2017.03.014.

2. *Similar to point 1, it is important to provide data confirming that the differentiated neurons are in fact enteric neurons (as opposed to other NCC-derived neurons such as sensory neurons). This can be established by quantitative analysis lineage specific marker expression.*

Ans: Neuronal cells were obtained after culturing hiPSC-ENCCs in neuronal differentiation medium for 20 and 40 days, namely hNP-20 and hNP-40, respectively, as described previously (Lau *et al.*, 2019). When we compared these neuronal derivatives from hiPSC-ENCCs (hNP-20 and hNP-40) to neurons from three main neuron subtypes (enteric, sympathetic and sensory) in the peripheral nervous system (PNS) (<http://mousebrain.org/adolescent/celltypes.html>) (Zeisel *et al.*, 2018). Consistently, neurons derived from hiPSC-ENCCs (both hNP-20 and hNP-40) show the highest similarity to the ENS neurons as demonstrated by the UMAP analysis and heatmap as shown below:

UMAP analysis comparing the neuronal derivatives of hiPSC-ENCCs (hNP-D20, hNP-4) with mouse enteric, sensory and sympathetic neurons.

Heatmap shows the expression of marker genes in the neuronal derivatives of hiPSC-ENCCs (hNP-D20, hNP-4), mouse enteric, sensory and sympathetic neurons, and reveals that hiPSC-ENCCs highly similar to the mouse enteric neurons.

References:

- Lau, S. T. *et al.* Activation of Hedgehog Signaling Promotes Development of Mouse and Human Enteric Neural Crest Cells, Based on Single-Cell Transcriptome Analyses. *Gastroenterology* **157**, 1556-1571 e1555 (2019). <https://doi.org:10.1053/j.gastro.2019.08.019>
- Zeisel, A. *et al.* Molecular Architecture of the Mouse Nervous System. *Cell* **174**, 999-1014.e1022 (2018). <https://doi.org:10.1016/j.cell.2018.06.021>

3. *Information about the differentiation stage (day of differentiation) should be provided for NCCs and neurons analyzed or evaluated for each experiment.*

Ans: This information was added in the figure legends of the revised manuscript. As stated in the method (page 20, line 606), ENCCs were enriched by FACS with antibodies against p75^{NTR} and HNK-1 at day 10 of the differentiation as described(Lai, 2017; Lai et al., 2021; Lau *et al.*,

2019). As for the neurons, unless specified, all neurons were harvested after culturing the neuronal differentiation for 5 days or 9 days.

References:

Lai, F.P., Lau, S.T., Wong, J.W.L., Gui, H., Wang, X.R., Zhou, T., Lai, W.H., Tse, H.F., Tam, P.K., Garcia-Barcelo, M.M., Ngan, E.S.W. (2017). Correction of Hirschsprung-associated mutations in human induced pluripotent stem cells, via CRISPR/Cas9, restores neural crest cell function. *Gastroenterology* 153, 139-153. 10.1053/j.gastro.2017.03.014.

Lai, F.P., Li, Z., Zhou, T., Leung, A.O.W., Lau, S.T., Lui, K.N., Wong, W.Y., Sham, P.C., Hui, C.C., and Ngan, E.S. (2021). Ciliary protein Kif7 regulates Gli and Ezh2 for initiating the neuronal differentiation of enteric neural crest cells during development. *Sci Adv* 7, eabf7472. 10.1126/sciadv.abf7472.

Lau, S.T., Li, Z., Pui-Ling Lai, F., Nga-Chu Lui, K., Li, P., Munera, J.O., Pan, G., Mahe, M.M., Hui, C.C., Wells, J.M., and Ngan, E.S. (2019). Activation of Hedgehog Signaling Promotes Development of Mouse and Human Enteric Neural Crest Cells, Based on Single-Cell Transcriptome Analyses. *Gastroenterology* 157, 1556-1571 e1555. 10.1053/j.gastro.2019.08.019.

4. *Line 159, reference missing for the statement about E12.5 mouse embryos.*

Ans: A reference (Ref. #21) was cited at the end of the sentence.

5. *It is unclear if in Figure 1 panels j and k the analysis is only performed on WT hPSC-derived NC or the merged WT and HSCR dataset.*

Ans: In the analyses of panels j and k, we merged WT and HSCR dataset for the comparison to confirm our iPSC derived NCCs were highly similar to the enteric NC. WT- and HSCR-ENCCs showed overall similarities with each other, and they were all predicted to have the greatest similarity to the bipotent progenitor (BP) cells by comparing each iPSC-derived ENCCs with the previous published datasets as described in text.

6. *In lines 199-202- the interpretation about the number of DEGs is inaccurate. The larger number of altered genes doesn't necessarily imply higher severity. Claim needs to be corrected.*

Ans: We did not claim more DEGs lead to higher severity in HSCR. We just described that we found more DEGs in advanced disease states through multiple approaches, "implying that more genes and pathways are disrupted in more advanced disease states".

7. *In line 262, more detailed information needs to be included about the model. The analysis should be further explained.*

Ans: The details of the model was included in the methodology section (page 27, Transcriptomic regulatory timing analysis).

8. *Line 269 should referred to Fig 3d not 4d.*

Ans: Amendment has been made.

9. In line 289- 293, the analysis description is vague? Was binding prediction used to filter the data? No data provided on binding prediction in the figure.

Ans: Yes, motif binding analysis was used to filter the activated target genes of HDAC1. It was reported HDAC1 can act as a transcription factor and bind to DNA sequences (promoter and enhancer) to initiate or enhance the gene expression. Therefore, additional filtering based on motif binding was used to reduce the false positive predictions. As stated in line 291, “where the activated genes represent the direct targets of the HDAC1 with binding motif(s) in their promoter regions (Fig. 3f)”. However, for the repressed target genes of HDAC1, we did not perform the motif analysis, because the binding of HDAC1 complex to chromatin is not clear. The data shown in Figure 3f was already filtered by motif analysis.

10. The main disease relevant phenotype in HSCR that link to the pathophysiology is migration defects (that lead to aganglionosis in newborns). It would be helpful assess migration assays in addition to neuronal differentiation in rescue experiments in figures 3, 6 and 7.

Ans: Yes, we did examine the effects of SAG and HDACi on migration of hENCCs using wound healing assays as described in the Fig. 1d. Consistently, we did not observe any significant changes in cell migration. Following is the result of migration assay using SAG-primed ENCCs (as shown in Fig. 7f). SAG-primed ENCCs exhibited better differentiation capacity (Fig. 7g), while their migration capacity is similar to the untreated cells.

Representative images of wound healing assays with unprimed and SAG-primed hENCCs. Bar chart shows the quantitative data from 3-8 independent experiments.

11. The splicing analysis in figure 4 is vaguely described in the results section (lines 337-367). Further details are necessary to assess rigor and accuracy. For example, the description provided for figure 4h is very vague. It is unclear what the x and y axes show. Figure legends overall contain minimal information and lack many crucial details.

Ans: The detailed descriptions have been included in the revised manuscript (page 12, line 340-356), legends (page 36, line 1148) and methodology. (page 29, Alternative splicing (AS) analysis)

12. In line 378-380, it is unclear how the motif analysis and filtering was done.

Ans: The method was described in the methodology section “Differential co-expression analysis”. In brief, RNA binding motif of PTBP1 was obtained from the published database. Then, RNA binding motif enrichment analysis was performed using FIMO tool of MEME suite. Information has been included in the methodology section. (page 28, line 896, Differential co-expression analysis)

13. The analysis done in Figure 5d needs to be explained further.

Ans: A description of the analysis have been included in the revised manuscript: “Target genes with activated or repressed exons were used to performed gene set enrichment analysis. Predictive scores reflected the strength and direction of regulation were calculated for the enriched pathways based on the integrative correlation coefficient (additive model).”(page 13, line 386)

14. Analyses in figure 5 are highly correlational. The wording should be clear to avoid any unsubstantiated implications of causality.

Ans: Amendment has been made.

15. In seahorse experiments, details are missing regarding the experiment design, number of replicates, what age cells were tested and what the graphs show.

Ans: All these information were available in the Method section (page 23) or in the main text.

- All experiments were replicated at least three times, and data are shown as means with SEM (page 30, statistical analysis section).
- ENCCs (Passage 1 -2) were seeded onto fibronectin-coated plate for Seahorse assay (Page 23, line 735).
- Seahorse assay is a standard assay used to measure the cellular metabolism. Seahorse XF platform can measure the flux of oxygen, the oxygen consumption rate [OCR], and the flux of protons, the extracellular acidification rate [ECAR], in the medium immediately surrounding cells in a microplate. With the specific inhibitors, the involvement of an ATP synthase and mitochondrial electron transport chain (ETC) can be tested. The details were described in the text (page 14, lines 415-433).
- Oxygen consumption rate (OCR) is measured before and after the addition of inhibitors to derive several parameters of mitochondrial respiration. Initially, baseline cellular OCR is measured, from which basal respiration and maximal respiratory can be derived

by subtracting non-mitochondrial respiration as shown below. All the calculations were made following the standard protocol provided by the manufacturer.

Basically, our data showed that HSCR-ENCCs exhibited lower OCR (lower OXPHOS), while addition of SAG can increase OXPHOS, via enhancing the fatty acid oxidation (FAO) Hedgehog/FAO/mitochondria axis, favoring the neuronal differentiation of HSCR-ENCCs. The details were also described in the text (pages 14-15).

16. More details necessary regarding the mitochondrial assays. What is GW9508? What is MTT?

Ans:

- We used a standard assay for measuring the mitochondrial activity (the mitoTracker Red). The experimental details are available in Method section (page 25). In brief, MitoTracker Red CMXRos is a red-fluorescent dye that stains mitochondria in live cells and its accumulation is dependent upon membrane potential. In other words, cells with high mitochondrial activity will pick up and accumulate more dye, which can be detected using fluorescent imaging.
- GW9508 is an agonist for activation of FAO pathway (a potent and selective G protein-coupled receptors FFA1 (GPR40)). As stated on page 15, line 475, “activation of FAO or Hedgehog pathway by GW9508 and SAG, respectively”.
- MTT is a chemical, 3-[4,5-dimethylthiazol-2-yl]-2,5-diphenyltetrazolium bromide. The MTT assay (also called cell proliferation assay) is used to measure cellular metabolic activity as an indicator of cell viability, proliferation and cytotoxicity.

17. *The justification for testing SHH agonists needs to be explained further.*

Ans: Based on the data from the transcriptomic and genetic analyses, neuronal differentiation defect of ENCCs is one of the common problems found in all HSCR patients (S/L-HSCR). Therefore, it is conceivable that rescuing the neuronal differentiation of ENCCs represents a possible treatment strategy. As stated in page 15, lines 451-455, one of our recent studies revealed that the addition of SAG (Smoothed agonist) during derivation of ENCCs from hiPSC primes the ENCCs to the higher cell state, favoring the subsequent neuronal differentiation (Lau *et al.*, 2019). “Subsequent analysis of the transcriptomes of SAG-primed hiPSC-ENCCs at single cell level further revealed that the genes implicated in glycolytic and fatty acid oxidation (FAO) pathways are robustly upregulated (Fig. 6a & b)“. Therefore, we examined whether activation of the non-canonical (GLI-independent) arm of Hedgehog pathway (glycolytic and FAO pathways) can promote neurogenic differentiation of hiPSC-ENCCs and data are shown in Fig. 6 and Fig. 7.

We have clarified this point in the revised manuscript (page 14, lines 448-450)

Reviewer #3:

The manuscript by Zhixin Li et al “Hedgehog-induced-oxidative-phosphorylation rescues neuronal differentiation defect of human enteric neural crest cells underlying Hirschsprung disease” describes a study carried out on human iPSC-derived enteric neural crest cell model (HSCR-ENCC). The authors aim to identify major changes in molecular pathways and networks associated with disease severity by high-resolution transcriptomic analysis and validate the findings with functional assays.

In a nutshell, the study reveals three major findings. First off, identification of a core gene set of 118 genes commonly dysregulated in all HSCR-ENCC lines, and HDAC1 as a master regulator of these genes. Importantly that increased nuclear localization and transactivation of HDAC1 in HSCR-ENCC is associated with suppressed mitochondrial activity and oxidative phosphorylation. The authors suggest that the elevated transactivation activity of HDAC might be an underlying cause of HSCR.

Secondly, upon the global analysis of splicing among different HSCR-ENCC, association of PTBP1 exon 9 inclusion with the disrupted cellular metabolism in HSCR-ENCC derived from the long segment of a colon is demonstrated.

Finally, basing on the findings above, which indicate a functional link with altered energy metabolism, the authors demonstrate the essential role of Hedgehog signaling mediated fatty acid oxidation and consecutive oxidative phosphorylation in the extent of neuronal differentiation capacity for ENCC.

Concerns/suggestions:

1. The authors claim the provision of “a comprehensive biological picture of the core molecular mechanisms underlying HSCR heterogeneity”. Concurring with this in principle, more elaborated analysis of the cross talk between the core molecular mechanisms is needed. For instance, it remains unclear whether proposed approach for restoration of ENCC functions in patients by boosting Hedgehog pathway or fatty acid oxidation is concomitant with the normalization of aberrant HDAC1 activity. In this respect the manuscript would benefit from a graphical abstract depicting the novel findings regarding the pathways in pathogenesis of Hirschsprung disease and their interplay.

Ans: We have prepared a diagram summarizing the pathways associated with HSCR pathogenesis and severity as shown below. It is included in the revised manuscript.

Fig. 8. Schematic summaries the key molecular and cellular processes associated with HSCR pathogenesis and severity. A core gene set associated with the severity of HSCR was identified. Additional alternations in genes implicated in RNA splicing and oxidative phosphorylation (OXPHOS) were found in L-HSCR cases. High HDAC1 activity is associated with the defective neuronal differentiation as observed in HSCR-ENCCs. Only the representative genes/pathways are shown.

Based on the current data and previously published articles, the canonical (Gli-dependent) Hedgehog pathway is associated with HSCR pathogenesis. HDAC1 can bind to Gli3 to repress enhancers and prevent premature activation of hedgehog target genes (Lex *et al.*, 2020; Lex *et al.*, 2022), while the aberrant Gli3 activator function and premature activation of Hedgehog are associated with HSCR pathogenesis (Liu *et al.*, 2015; Ngan *et al.*, 2011; Sribudiani *et al.*, 2018). On the other hand, the non-canonical (Gli-independent pathway) Hedgehog pathway favors the neuronal differentiation of hENCCs (Lau *et al.*, 2019) and rescues the differentiation defects of HSCR cells, via enhancing the OXPHOS (not through HDAC1). Thus, the two arms of Hedgehog pathway possess different biological roles during ENS development.

References:

- Lex, R. K. *et al.* GLI transcriptional repression is inert prior to Hedgehog pathway activation. *Nat Commun* **13**, 808 (2022). <https://doi.org/10.1038/s41467-022-28485-4>
- Lex, R. K. *et al.* GLI transcriptional repression regulates tissue-specific enhancer activity in response to Hedgehog signaling. *Elife* **9** (2020). <https://doi.org/10.7554/eLife.50670>
- Sribudiani, Y. *et al.* Identification of Variants in RET and IHH Pathway Members in a Large Family With History of Hirschsprung Disease. *Gastroenterology* **155**, 118-129.e116 (2018). <https://doi.org/10.1053/j.gastro.2018.03.034>
- Liu, J. A. *et al.* Identification of GLI Mutations in Patients with Hirschsprung Disease That Disrupt Enteric Nervous System Development in Mice. *Gastroenterology* (2015). <https://doi.org/10.1053/j.gastro.2015.07.060>
- Ngan, E.S., Garcia-Barcelo, M.M., Yip, B.H., Poon, H.C., Lau, S.T., Kwok, C.K., Sat, E., Sham, M.H., Wong, K.K., Wainwright, B.J., et al. (2011). Hedgehog/Notch-induced premature gliogenesis represents a new disease mechanism for Hirschsprung disease in mice and humans. *J Clin Invest* **121**, 3467-3478. [10.1172/JCI43737](https://doi.org/10.1172/JCI43737).
- Lau, S.T., Li, Z., Pui-Ling Lai, F., Nga-Chu Lui, K., Li, P., Munera, J.O., Pan, G., Mahe, M.M., Hui, C.C., Wells, J.M., and Ngan, E.S. (2019). Activation of Hedgehog Signaling Promotes Development of Mouse and Human Enteric Neural Crest Cells, Based on Single-Cell Transcriptome Analyses. *Gastroenterology* **157**, 1556-1571 e1555. [10.1053/j.gastro.2019.08.019](https://doi.org/10.1053/j.gastro.2019.08.019).

2. Lines 330-334: The statement that aberrantly high HDAC1 transactivation activity represents a common cause of neuronal differentiation defect associated with HSCR disease is overstatement. Evidently, inhibition of HDAC1 activity favours the neuronal lineage differentiation.

Ans: We have revised the sentence to “aberrantly high HDAC1 transactivation activity is likely associated with the neuronal differentiation defect as observed in HSCR-ENCCs”.

3. Basic characterization of the iPSC lines is missing. Also, other important details of the iPSCs is missing: how many line per each donor were used (to avoid the impact of line-line variation and understanding the reliability of the lines). Were the lines differentiated in parallel? Have the assays, stainings and blots been replicated 2-3 times?

Ans: Some of our iPSC lines have been used for other projects and the characterization data have been published previously(Lai, 2017).

We followed the gold standards for characterization of all the iPSC lines:

1. Immunocytochemistry to detect the expression of stem cell markers including SOX2, OCT4, NANOG, SSEA-4 and TRA-1-60, etc.
2. Methylation assay to confirm demethylation of *NANOG* promoter.
3. Karyotyping
4. Bulk RNAseq to analyze the global transcriptomes and data were then compared to the public dataset (human ES/iPSC).
5. Mycoplasma test (all lines are negative to mycoplasma before used)

Please refer to the data enclosed below (Supplementary Fig. 9)

Characterization of HSCR-iPSC lines. a-g: a' Immunocytochemistry analysis on the expression of stem cell markers; b' Promoter of *NANOG* is unmethylated; c' Normal karyotype in HSCR-iPSC. h. PCA analysis of bulk RNA-seq data shows the transcriptomic profiles of all HSCR-iPSC lines are highly comparable to that of hES/hIPSC.

For this study, we only selected one iPSC clone from each patient for the analysis. All the clones selected for this study have normal karyotype and the expression profiles highly resemble to the other hES/hiPSC lines based on the characterization data listed above.

We understand that ideally, more clones should be selected, but it is extremely expensive to perform scRNA-seq using SMARTer platform (which allows us to analyze the RNA splicing). To balance the cost and the potential artifacts due to the variations among clones/individuals, we included more lines derived from different individuals rather than multiple clones from the same individual (e.g. 4 S-HSCR and 2 L-HSCR) in this study. To minimize the potential variations, we also enriched ENCCs by FACS for the both transcriptomic and function analyses. As shown in Fig. 1h, the ENCCs derived from different patient lines are highly similar to each other. Similarly, the yields of ENCCs derived from all the iPSC lines were very comparable (Fig. S1), and hiPSC-ENCCs from the same disease group also exhibited similar differentiation and migration capacities. More importantly, confound effects associated with genetic background, genders or line effects, were removed before the analyses.

The differentiation assays were performed many times (at least 5-8 times) for different parts of the study. Technically, it is impossible to run the differentiation assays of all lines at the same time, but the differentiation assays were performed more the less in parallel. It is just like all the other studies, the data present here are the means of values from multiple independent experiments.

As stated in the method section (page 30, Statistical analysis),” All experiments were replicated at least three times, and data are shown as means with SEM. “

4. The seahorse data (shown in figs 5-6) should be analysed per donor/line, now the meaning of the dots is unclear. The same is true for ATP, lactate, pyruvate, MTT and mitotracker data which now seem to represent pooled data, still there is considerable overlap in seahorse and mitotracker values between the groups. Based on the current data, it is possible that many diseased lines did not actually differ from controls.

Ans: The cellular metabolism and mitochondrial activity vary along cell state (cells at different stages of cell cycle). Therefore, we usually included 6-8 replicates for the Seahorse and MTT assays. As for mitotracker assay, we analyzed the mitochondrial activity at single cell level, at least 200 cells were counted in each experiment. For all the assays, 3-5 independent experiments were performed. Data are shown as means of all replicates with SEM.

Given that there are many replicates included for the statistical analysis, it is not surprised to see overlaps among groups, but with the significant *P*-values. It is also true for the scRNA-seq data. In addition, we did not pool the data from different patient lines for all functional assays. All the data shown in figure 6 were obtained from the assays using the control-cells (IMR90), in presence or absence of various agonists/inhibitors.

Reviewer #4:

This manuscript presents single-cell transcriptomic analyses of enteric neural crest cells in both healthy and disease states. It is largely a descriptive study, but the authors make many mechanistic and causal claims that are unsupported by the descriptive data presented. A descriptive analysis of gene expression would be acceptable and appropriate, but making unsupported mechanistic and causal claims from descriptive and correlative data is not.

A few examples of this are provided here. (This is not an exhaustive list.)

1. It is claimed in the abstract and elsewhere that a HDAC1 was shown to be a “master regulator” of a “core gene set of 118 genes.” However, no causal evidence is provided. In fact, no clear evidence that HDAC1 regulates these genes is given. The evidence presented is that HDAC1 expression is correlated with the expression of some of these genes, and that a few of the genes have HDAC1 predicted motifs located nearby. This descriptive evidence is nowhere near sufficient to match the broad, causal claims of HDAC1 as a master regulator.

Ans: We apologize for the wordings that we used. Given that it is the bioinformatics study, the conclusions we drawn were based on how significant (statistically) the data are. We have changed the wording from “master” to “potential”.

2. It is claimed that there is a “highly confident causal regulation between” PTBP1 and PKM. However, the only evidence provided is a (modest) correlation between PTBP1 and PKM. This is evidence of a correlation, but certainly not a “highly confident causal regulation.”

Ans: Same as above, we have revised the wordings.

3. It is claimed that “PKM1 activity.... due to the inclusion of ptbp1 exon 9....” leads to perturbed OXPHOS, “leading to the retarded neuronal differentiation.” However, none of these links were causally established by evidence in the manuscript. It was not demonstrated that PKM1 was causally regulated by ptbp1 (only correlation). It was not shown that PKM1 expression levels are directly the cause of OXPHOS perturbation (again, only correlation). It was also not demonstrated here (Fig 5) that OXPHOS perturbation is causally linked to neuronal differentiation defects.

Ans: The PTBP1 mediated RNA splicing of PKM genes is well established (Gueroussov et al., 2015), this cellular process is universal and not cell-context specific.

Additional experiments were performed to demonstrate the link between PKM1 and OXPHOS and the data are included in the revised manuscript (Fig. 8h-l).

These are just a few examples of over-exaggerated causal claims that are not supported by the descriptive, correlative data. Many more examples could be provided. This manuscript would be well served by a wholesale alteration from making causal, mechanistic claims into a more narrowly-focused descriptive analysis of gene expression.

Ans: We have provided additional data to support our claims on the causal links between alternative splicing for generation of PKM1 and PKM2, cellular metabolism and neurogenesis. The rescue experiments shown in Figures 6 and 7 also demonstrated the implications of the

non-canonical Hedgehog signaling with cellular metabolism (OXPHOS) and neurogenesis in control and HSCR-ENCCs.

References:

Gueroussov, S., Gonatopoulos-Pournatzis, T., Irimia, M., Raj, B., Lin, Z.Y., Gingras, A.C., and Blencowe, B.J. (2015). An alternative splicing event amplifies evolutionary differences between vertebrates. *Science* 349, 868-873. [10.1126/science.aaa8381](https://doi.org/10.1126/science.aaa8381).

Reviewer #5:

In their study, Li et al. generated seven patient-derived iPSC-based cell lines to investigate the cellular/molecular bases of Hirschsprung disease (HD). First, they characterized the lines by single cell RNAseq and bioinformatics and by cell biological assays (Figs. 1-2). Then, based on further analysis, the Authors suggest three key findings: (1) aberrantly high HDAC1 transactivation activity is a common cause of neuronal differentiation effects associated with the disease (Fig. 3), (2) reduced PKM1 activity at least partly due to the inclusion of exon 9 in PTBP1 may perturb OXPHOS leading to retarded neuronal differentiation (Figs. 5-6), and (3) Hedgehog-induced OXPHOS can enhance the otherwise impaired survival and differentiation capacity in HD patient-derived cells lines (Figs. 6-7).

While this is an appealing set of conclusions, the manuscript does not generate strong confidence in the results. Despite each of the key findings seeming to suggest some sort of insight into impaired differentiation and/or survival that is associated with the disease, different parts of the study appear to be only loosely connected. In addition, many of the arguments/conclusions lack solid experimental support. Overall, the manuscript gives the impression of multiple undeveloped observations being packaged together. The Hedgehog-induced rescue indicated in the title is only one part of the manuscript and does not require or depend on any other analysis (e.g. HDAC1, splicing, PTBP1, PKM). Perhaps the Authors could focus on this final line of investigation (the Hedgehog-induced rescue) and develop it more fully.

Ans: It is an unbiased transcriptomic profiling study of a multigenic disease. The aim of this study was to give an overview of the key biological pathways implicated in the pathology and disease severity of HSCR. It is expected that multiple biological pathways are involved and account for the heterogeneity of the disease. Like many other genome-wide study, it is impossible to address the function of every candidate gene, most likely, the genes work together to affect the biological pathways, leading to disease. Under this framework, we showed the dysregulation of HDAC1, alternative splicing and metabolism are the key pathways based on the bioinformatic analyses. In the revised manuscript, additional experiments were performed to demonstrate the causal links between alternative splicing of PKM1 and PKM2, cellular metabolism and neurogenesis in control and HSCR-ENCCs.

Critique**1. Controls**

1.1 The Authors argue later in the paper that there are critical transcriptomic differences between control and patient cell lines. However, in Fig. 1h, the Authors show a tSNE plot in which the majority of cells, including control and patient-derived, are grouped together. While this does not exclude the possibility of differences, this suggests patient-derived cells are transcriptomically not very different from controls.

Ans: It is well known that great individual effects exist in human data which make every iPSC line differ from each other (see the tSNE plot below), but they are mostly due to the differences in genetic background and genders, etc. Therefore, we first removed these individual effects and made the cells comparable, as stated in page 7, line 187-188, "Genes that were uniquely expressed in one sample were excluded for downstream analysis to remove individual effects.". Then, we aimed to identify the key cluster for the subsequent analyses, so we mapped the cells at a low resolution in order to identify the shared cell types/cells at similar cell states among

cells from different individuals (data shown in Fig. 1h). Fig. 1h suggests the variation of cell types/states is the dominant variation in cells after batch/individual effect removal. However, when looking at cells in Cluster 1 in Fig. 1h, we can still see cells from different control/HSCR-ENCCs have different distributions, suggesting disease severity become the most significant variation within Cluster 1. Therefore, we next focused on Cluster 1 (with molecular signatures resembling bipotent ENCCs) to identify the genes/pathways associated with disease severity. In our paper, we have carefully characterized the variations of individual, batch effect, cell type/state and disease severity.

t-SNE projection of all 3,342 individual cells before removing individual/batch cofounders.

1.2 The plot also highlights that the two control lines are just as different from each other as they are from other patient-derived cells. Such a pronounced and potentially uncontrolled variability between controls would fundamentally compromise any subsequent analysis.

Ans: Again, it is normal to see difference among cells from the two control lines. However, Fig. 2a-c suggest that the two control lines are very similar: Cluster together in Cladogram (Fig. 2b) and in PCA plot (Fig. 2b). Similarly, they have similar distribution on PCA plot (Fig. 2c, bottom), as well as similar activities for key pathways (Fig. 2g-j, neurogenesis, proliferation, RNA splicing and energy metabolism). All these data suggested that the two control were very comparable. Therefore, the two control lines were included in the study. In all our subsequent analysis, we compared each HSCR-ENCC to the cells derived from the two control lines independently. Only the genes significantly different from that in the cells derived from BOTH control lines were considered as DEGs. All the downstream analyses followed this rule and was stated in page 7, line 194, we stated “Next, we compared each HSCR-ENCCs with those from the two controls”. This strategy can significantly reduce the number of false positive DEGs.

1.3 The origin of controls further complicates this issue. The Authors generated their patient-derived cell lines from skin biopsies. By contrast, one of the controls was derived from from urine-derived cells (UE02302, male; as described in Methods), whereas the other from fetal fibroblasts (IMR90, female; as described in the original article, Yu et al., Science, 2007).

Ans: These two control iPSC lines included in this study were carefully selected, based on data obtained from a series of functional and transcriptomic analyses. We included the two control lines with different genetic background and genders to match with those found in the patient group (mixed gender with different genetic background).

Even though the two control iPSC lines were derived from different sources, after reprogramming, they exhibit similar expression profiles as shown in the PCA plot below. In concordance with this observation, ENCCs derived from the two control iPSC lines tended to be clustered together (Fig. 2a & b) and shared great similarities (Fig. 3e).

PCA plot of fibroblast (hFb), iPSC and ENCCs from control and HSCR groups and three hESC lines. All the iPSC (control and HSCR) lines are clustered together and in close proximity to four hESC lines (H7, HUES1, HUES8 and HUES9). Bulk RNA-seq of hESC lines were downloaded from NCBI database (<https://www.ncbi.nlm.nih.gov/geo/query/acc.cgi?acc=GSE102311>). Bulk RNA-seq samples of other lines were previously sequenced by our group.

1.4 Clearly, the best practice would be to include multiple control lines in the study that fully match the patient-derived protocols. If this is not possible (e.g. because of subject recruitment constrains), the Authors at least should provide a detailed analysis of the controls (including differential gene expression analysis) showing that they can be appropriately used for this study, and control for their variability in other analyses.

Ans: Yes, we have carefully addressed this issue in our analyses. There were genes uniquely expressed in each control, we considered them as individual effect and removed them for downstream analyses. For the differential gene expression analysis, we compared each HSCR-ENCC to cells from the two control lines separately. Genes in HSCR-ENCCs significantly different from that of the cells from BOTH of these two control lines were considered as true DEGs. This information is available in the text.

1.5 Questions regarding the use of controls also emerge later in the study. At least, in Figs. 5a, S1a and S6a, figure displays suggest that patient-derived data were only compared to UE02302 as control, whereas in Fig. S7, they were only compared to IMR90. In Figs. 2g-j, 3e, 5e-j, 6a-n, S4 only „control“ is displayed, without specifying whether IMR90 or UE02302 or both were used. The Authors should show how their results change depending on which control(s) they use, and clearly describe and explain the use of controls in each legend and the text.

Ans: The two controls were very similar. For all bioinformatic and RT-qPCR analyses, two controls were merged for comparison (Fig. 2g-j, 3e, 5a-g, 6a & b) and HSCR-ENCCs were compared to the two controls (merged). Information has been updated in the figures and/or figure legends.

2. HDAC1

2.1 The Authors identified multiple genes whose expression changed along the severity axis (this axis is sometimes called disease axis, e.g. Fig. 2d-e versus Fig. 3d, which is additionally confusing). Of these, they quickly singled out HDAC1 as the most relevant hub gene, because

PPI network analysis suggested its link to NC development and neurogenesis. The relevances/PPIs of other genes were not discussed. (Peculiarly, the expression level of HDAC1 also appears to be negative in low severity cases, Fig. 3D. Whether this is an error or consequence of an unusual and unexplained scale transformation, it certainly does not help to generate confidence in the results. - The units of expression and other parameters should be always clearly displayed in all plots.)

Ans: Pseudotime disease axis was first proposed by Lang *et al* to mathematically describe the dynamic changes of disease state of cells based on gene expression profile (Lang et al., 2019). We selected HDAC1 as the key regulator based on the bioinformatic analyses of (1) PPI, (2) co-expression and (3) motif enrichment. The function of HDAC1 on neuronal differentiation was further experimentally validated (Fig. 3).

In Fig. 3a, each HSCR-ENCC was compared to the merged controls, the changes in the gene expression presented as log₂FC and marked by color. HDAC1 showed no significant difference in expression in mild-HSCR cases, but it doesn't mean HDAC1 is not expressed. As shown in Fig. 3h and Supplementary Fig.6a, HDAC1 is expressed in all control and HSCR-ENCCs.

To prevent confusion, we used "disease severity axis" in plot.

We have included the axis scale in the figure legend of Fig. 3d, "Gene expression (log₂(TPM+1)) was used to fit the smooth line. TPM, Transcript per million."

2.2. To support the focus on HDAC1, the Authors showed that (i) another core set genes (whose expression also changed) were upregulated in patient-derived cell lines in a similar fashion to HDAC1, (ii) some of these are predicted to be „high-confidence“ direct targets of HDAC1, and (iii) the presumed gene regulatory network of HDAC1 was slightly activated in low severity disease state but highly activated in severe disease state. Based on these arguments, they suggested that the upregulated expression and elevated transcription activity of HDAC1 would be an underlying cause of the disease.

However, Western blot showed that the total protein expression of HDAC1 was not different between controls and patient-derived cells. These data is only shown in supplementary Fig. S6, but they should be clearly in main figure. (The Western was done only on 4 out of 7 patient-derived lines, and although the main text provides some reasons why this was the case, the results should be shown for all lines.)

Ans: Our Western blot (WB) data did not detect any significant elevation of HDAC1 expression in HSCR-ENCCs, so we have excluded the elevated HDAC1 expression is associated with HSCR pathogenesis. Given that all the main figures are very busy, we decided not to include the negative data in the main figure.

Indeed, we have also examined the expression of HDAC1 in all HSCR-ENCCs, basically, they are very similar to each other. As requested by the reviewers, we included ALL of them in the revised manuscript (Supplementary Fig. 6b).

2.3 As an alternative hypothesis, the Authors then speculate that the nuclear localization of HDAC1 was enhanced, which lead to increased the transactivation activity of HDAC1 in patient-derived cell lines. This speculation is based on the observation that cytosolic HDAC1

signal was not visible in the patient-derived cell lines, only in controls (Fig. 3h). However, this observation does not directly show that nuclear HDAC1 localization was higher in patient-derived cells (even if the Westerns looked similar). It is also unclear if the lack of cytosolic HDAC1 in the patient-derived cell lines was related to the disease, or appeared because of other reasons, e.g. due to the use of different control lines (see first point).

Ans: Please note that TWO control lines were tested for HDAC1 localization (Fig. 3h), both nuclear and cytosolic HDAC1 were detected in ENCCs from the TWO control lines. (please also refer to our responses below)

2.4 Using an in vitro differentiation assay, the Authors showed that the application of valproic acid (inhibitor of HDAC1 among others) significantly enhanced NF signal in patient-derived cell lines (also in one of the controls, but not in the other), indicative of enhanced differentiation into a neuronal lineage. However, this experiment does not make a strong case for HDAC1, given that valproic acid could act through drug effectors other than HDAC1.

Ans: Please note that TWO control lines were tested for VPA. Again, ENCCs from the two control lines are highly comparable based on their transcriptomes and the cellular phenotypes.

2.5 To conclusively show the involvement of HDAC1 in the disease, the Authors need to gather more solid evidence. For example, they could evaluate the presence/absence of cytosolic signal in matching controls, use quantitative measurements to evaluate the nuclear localization signals, and employ molecular-genetic perturbation experiments to target HDAC1 and confirm their preliminary results using valproic acid (e.g. shRNA or similar approach in patient-derived lines, overexpression in controls).

Ans: To better match with the scRNA-seq data, we only used the FACS-sorted ENCCs (without culturing) for the analysis of cellular localization of HDAC1. Therefore, we needed to pool multiple FACS-sorted cells for WB and technically, it is impossible to get sufficient cells for fractionation analysis. Alternatively, we used CellProfiler to quantify the nuclear and cytoplasmic HDAC1 expressions **at the single cell level** as described previously (Stirling et al., 2021). To enhance the measurement precision, cells were imaged using a Carl Zeiss Confocal microscope LSM810 with a high numerical aperture lens, 63x, and the same imaging settings. Images were saved in a megapixel resolution. A pipeline was set up to measure the HDAC1 expressions in each cell on the images. In brief, the nuclear and cytoplasmic regions were defined using the “IdentifyObjects” modules and the DAPI- and HDAC1-stained images, respectively. The HDAC1 expressions were measured in the defined nuclear and cytoplasmic regions for each cell using the module “MeasureObjectIntensity”. For the cell lines where the cytoplasmic region was difficult to define, the cytoplasmic HDAC1 expression was measured in the perinuclear region instead of the entire cytoplasmic region. The nuclear-to-cytoplasmic HDAC1 ratio was calculated and compared between cell lines using one-way ANOVA. For each cell line, more than 200 cells were analysed.

As shown in the graph below, ENCCs from the two control lines exhibited very similar nuclear to cytosolic HDAC1 signal, which is significantly lower than those in HSCR-ENCCs. Please note that WB only shows the **total expression** in a pool of cells, while the current approach can analyze the HDAC1 expression **at single cell level**. Given that the total HDAC1 expression levels in all ENCCs (both control and HSCR) are similar, the enhanced nuclear-to-cytoplasmic HDAC1 ratio is associated with HSCR disease.

Followings are the quantitative result:

Quantitative data of HDAC1 expression in nucleus and cytoplasm. Data show the ratio of nuclear to cytosolic HDAC1 signal in each cell (each dot). The lines represent the median values of nuclear to cytosolic ratio in each group.

We have also blocked HDAC1 activity using two different HDAC inhibitors (pan-HDAC inhibitor: VPA and HDAC1 class I inhibitor: CI994) (figures shown as below) and knocking down *HDAC1* specifically with morpholinos (Fig. 3i). All the data showed that inhibition/knock-down HDAC1 favors the neuronal differentiation of ALL HSCR-ENCCs.

In vitro differentiation assays in absence or presence of HDAC inhibitors: **(A)** Valproic acid (VPA, 1mM) and **(B)** HDAC class I inhibitor (CI994, 500nM). Cells were harvested for immunofluorescence staining at day 5 of neuronal differentiation. Chart shows the quantitative data.

Reference:

Stirling, D.R., Swain-Bowden, M.J., Lucas, A.M., Carpenter, A.E., Cimini, B.A., and Goodman, A. (2021). CellProfiler 4: improvements in speed, utility and usability. *BMC Bioinformatics* 22, 433. 10.1186/s12859-021-04344-9.

3. Splicing, *PTPBI*, *PKM*, *OXPPOS*, *Hedgehog*

3.1 Unrelated to *HDAC1*, the Authors went on to analyze splicing events globally, because the pathway score for RNA splicing was significantly higher in patient-derived cell lines with advanced disease severity (Fig. 2i).

Ans: In this study, we aimed to use transcriptomic analysis to reveal the key pathways implicated in **HSCR pathogenesis and severity**. As stated in the introduction, “It is generally believed that the majority of HSCR cases are caused by cumulative actions of multiple genetic variants, each with minor effects. While each individual may carry different constellations of

genetic variants, they likely converge on shared pathogenic pathways, leading to HSCR disease. On the other hand, additional modifiers likely determine disease severity and account for the disease spectrum.“ Therefore, we first gave an overview of the transcriptomic profiles of control, S-HSCR and L-HSCR, and what are the pathways enriched in S-HSCR and L-HSCR (Fig. 2). Then, we further identified the core gene sets (commonly dysregulated in all HSCR cases) and the key genes which changed along the severity axis (i.e. associated with disease severity (Fig. 3). Based on the data shown in Fig. 2 and Fig. 3, the overall expression of RNA splicing pathway was elevated in advanced disease state (Fig. 2i) and the RNA splicing related genes were predicted to be activated by HDAC1 (Fig. 3g). Therefore, the global splicing events were analyzed (Fig. 4).

3.2 Then, they also showed data suggesting that splicing events were more frequent in high severity cases (Fig. 4b), compared the occurrence of different splicing events (Fig. 4c; although it is unclear from the legend and text which comparison do these p values refer to; #17 to controls?, #1 to controls?, #17 to controls?), and performed other bioinformatic analyses (Fig. 4d-e). Finally, they quantified the weight of AS events (Fig. 4f).

Ans: The cells used for comparison was stated in manuscript (page 12, lines 343-344) and in the figure legend of Fig. 4c. “Except for intron retention, all the other six AS types had significantly higher numbers of events in ENCCs at high disease state (HSCR#1, 17 and 23) than those in low disease states (two controls and 3 S-HSCR-ENCCs) ($P < 0.05$, t -test).”

3.3 Unfortunately, the manuscript lacks sufficient technical descriptions on the splicing calculations, making it hard to judge what the data actually means. The Authors should clearly define (1) number of splices, (2) event count, (3) AS frequency, (4) transcriptional level versus post-transcriptional level, (5) weight of AS events - even if these were output from other softwares they used. For example, while top ranking of PTPB1[1] in the weight of AS events plot is a critical argument to follow up on this variant, the calculation method is not explained neither in this nor in the original paper (42) they cite.

Ans: **(1)** Number of splices is total number of splicing events in each read which is generated by STAR software (Dobin et al., 2013) during read alignment. Seven alternative splicing (AS) events were identified by SUPPA (Trincado et al., 2018), including Skipping Exon (SE, also known as cassettes), Alternative 5'/3' Splice Sites (A5/A3, also known as donors/acceptor), Mutually Exclusive Exons (MX), Retained Intron (RI) and Alternative First/Last Exons (AF/AL). **(2)** Event count is the total number of AS events in a cell. **(3)** AS frequency was determined by the number of splices divided by the uniquely mapped read number in each cell, which is a normalized value to evaluate relative AS level of the cells. **(4)** Transcriptional level refers to the gene expression of the cells versus RNA splicing represents the post-transcriptional level of cells. **(5)** Weight of AS events was derived from the PCA analysis, indicating which AS events contribute most to the specific PCs. Detailed description was added to methodology section (page 29, Alternative splicing (AS) analysis).

3.4 Independently, the reason to choose PTPB1[1] (inclusion of exon 9) as opposed to other PTPB1 splicing events or other upregulated factors appears somewhat arbitrary. For example, the significant intron retention seen in PTPB1 (PTPB1[3]) would likely render the protein dysfunctional, and irrespective of inclusion of exon 9 could have a more significant contribution to the disease. Overall, the path of reasoning from PTPB1[1] to PKM (one target of PTPB1) and through that to OXPHOS, on which the Authors had data, feels constructed.

Ans: Please note that the bioinformatic study can nicely provide a global view of the changes. Nevertheless, it remains challenging to functionally validate every single DEG identified from the bioinformatic study. Therefore, we could only select few genes along the same biological pathway for functional validation.

The selection of *PTBP1* Ex9 for further analyses based on its top score in AS event weighting analysis (Fig. 4f) and the functional data from the literatures that highlight their biological relevance to cellular metabolism. Dysregulation of cellular metabolism was consistently found associated with L-HSCR, so its potential link between PKM pathway was further assessed.

According to the published functional studies, PTBP1 is a RNA binding protein, the most important structures are RNA binding domains. Even though intron retention seen in PTBP1 (PTBP1[3]) renders the protein dysfunctional, it is not clear whether it indeed affects the RNA binding function of PTBP1 (there is no supporting report published so far). However, exon 9 in *PTBP1* encodes a linker between RNA recognition motif 2 (RRM2) and RRM3 in *PTBP1*, and it was reported that inclusion of Exon 9 changes the RNA binding profiles of PTBP1 (Gueroussov *et al.*, 2015). Considering the most significant weight in PCA analysis (Fig. 4f) and known splicing function of PTBP1 exon9, PTPB1[1] (inclusion of exon 9) was considered as a critical AS event and selected for the downstream analysis and functional validation.

3.5 In any case, based on their data, the Authors suggest that increased expression of PTBP1 and/or increased inclusion of exon 9 in PTBP1 would result in decreased PKM1 (also decreased exon 9 inclusion in PKM1) and increased PKM2 (also increased exon 10 inclusion in PKM2) production. However, this hypothesis itself is not new and the manuscript should include references/discussion to/on previous research on this.

Ans: A reference was cited in the manuscript “Exon 9 in *PTBP1* encodes a linker between RNA recognition motif 2 (RRM2) and RRM3 in *PTBP1*. It also possesses splicing regulatory activity that are distinct from its RNA-binding activity (Gueroussov *et al.*, 2015). Therefore, we further analyzed the relationship between *PTBP1* exon 9 inclusion and other AS events found in HSCR-ENCCs.” (page 13, line 378) And “One of the known targets of *PTBP1* is *PKM*, which encodes two key enzymes (pyruvate kinase M [PKM] 1 and PKM2) involved in glycolysis, and they are responsible for converting phosphoenolpyruvate (PEP) to pyruvate as a substrate for oxidative phosphorylation (OXPHOS).” (page 13, line 394)

3.6 Using RT-qPCR, the Authors show that PKM1 levels were significantly decreased in all patient-derived cell lines and that there were no significant PKM2 changes in either of the lines (Fig. 5f). While this finding somewhat confirms their hypothesis (i.e. PKM1 downregulation in high severity cases), it also raises the question why PKM1 was significantly downregulated in low severity cases, in which PTPB1 was neither upregulated (Fig. 4a) nor its splicing has changed (Fig. 5a). Similarly the question of why alternative splicing is impacted so strongly in severe cases, but is unchanged in less severe cases, while their previous analyses (Figs. 2 and 3) showed more gradual gene expression changes along the severity axis remains unresolved.

Ans: We indeed examined the *PTBPI* Ex9 level with multiple batches of ENCCs from S-HSCR (HSCR#5, #20, #10 and #1), but we consistently could not detect any significant increase in *PTBPI* Ex9 transcripts by RT-qPCR, while *PKM1* was significantly down-regulated in these cells. Therefore, it is conceivable that *PTBPI* Ex9 is only one of the splicing regulators determining the *PKM1* level. *PKM1* would be regulated by multiple factors. Based on our association study, the association of *PTBPI*-Ex9 with *PKM1* and metabolism in S-HSCR is less significant than those in L-HSCR. The scRNA-seq is a very sensitive approach with high analytic power for detecting the subtle association(s), while the RT-qPCR was performed with pool of cells with lower sensitivity. This may explain why the association between *PTBPI*-Ex9 and *PKM1* could not be demonstrated by RT-qPCR in S-HSCR. As stated in the text (page 14, line 441), we believe “the reduced *PKM1* activity in HSCR, at least **partly due to** the inclusion of *PTBPI* exon 9”

Fig. 2 and Fig. 3 show that more genes are dysregulated at transcriptional level in severe cases, while Fig. 4 shows more AS events (post-transcriptional level) in severe cases. As shown in Fig. 4h, data at transcriptional and post-transcriptional level are two layers reflecting the disease severity even though they are not totally independent. As stated in the text, “The combination of transcriptional and post-transcriptional profiles explains the measured clinical scores of the HSCR cases, providing a comprehensive picture of the molecular constituents and mechanisms underlying the heterogeneity of HSCR” (page 12, line 368).

3.7 Irrespective how the Authors arrived to OXPHOS, the OCR analysis (Fig. 5g-j) reveals a potentially interesting phenotype in patient-derived cell lines. However, their conclusion this phenotype to be due to „reduced PKM1 activity in HSCR, at least partly due to the inclusion of PTBPI exon 9, may perturb OXPHOS, leading to the retarded neuronal differentiation of ENCCs“ (lines 421-423) is not backed up by experimental data and speculative at most. In order to support that PTBPI or PKM1 indeed played a role in this phenotype, the Authors should at least perform PTBPI[1] and PKM1 overexpression experiments and see if those show a rescue in the OCR assay.

Ans: The *PTBPI* mediated RNA splicing of *PKM* genes is well established (Gueroussov *et al.*, 2015), this cellular process is universal and not cell-context specific. Therefore, additional experiments were performed to directly demonstrate the link between *PKM1*, OXPHOS and neuronal differentiation of ENCCs from control and severe HSCR groups, and the data are included in the revised manuscript (Fig. 8h-l).

References:

Gueroussov, S., Gonatopoulos-Pournatzis, T., Irimia, M., Raj, B., Lin, Z.Y., Gingras, A.C., and Blencowe, B.J. (2015). An alternative splicing event amplifies evolutionary differences between vertebrates. *Science* 349, 868-873. 10.1126/science.aaa8381.

3.8 Finally, the Authors suddenly jump onto investigating Hedgehog signaling. They even introduce a completely new sequencing data set (Fig. 6a-b), the origin of which is not clarified. Is this coming from the Author's previous study? If yes, does it represent mouse or human data? The Authors should clearly describe (supported by detailed analyses comparing controls) how these data relates to the data they analyzed in their manuscript until this point. Regarding Hedgehog, they then show analyses which are potentially interesting, but unrelated to any of

their above analyses; the relationship between this part of the study and HDAC1, splicing, and PTPB1 is not recognizable.

Ans: Based on the data from the transcriptomic and genetic analyses, neuronal differentiation defect of ENCCs is one of the common problems found in all HSCR patients (S/L-HSCR). Therefore, it is conceivable that rescuing the neuronal differentiation of ENCCs represents a possible treatment strategy. As stated in page 15, lines 451-453, one of our recent studies revealed that the addition of SAG (Smoothed agonist) during derivation of ENCCs from hiPSC primes the ENCCs to the higher cell state, favoring the subsequent neuronal differentiation (Lau *et al.*, 2019). “Subsequent analysis of the transcriptomes of SAG-primed hiPSC-ENCCs at single cell level further revealed that the genes implicated in glycolytic and fatty acid oxidation (FAO) pathways are robustly upregulated (Fig. 6a & b)“. Therefore, we examined whether activation of the non-canonical (GLI-independent) arm of Hedgehog pathway (glycolytic and FAO pathways) can promote neurogenic differentiation of hiPSC-ENCCs and data are shown in Fig. 6 and Fig. 7.

We have clarified this point in the revised manuscript (page 14, lines 448-450).

Reviewers' Comments:

Reviewer #1:

Remarks to the Author:

The authors have improved the manuscript by addressing my main concerns. As for claiming Cluster 1 as ENCC, the authors provided transcriptome analysis data in their response letter to reviewer, but they were not included in the revised manuscript. They should include and discuss these data in the revised manuscript.

Reviewer #2:

Remarks to the Author:

This revised manuscript by Li et al addresses many of the comments and concerns raised previously but a few issues remain to be resolved:

General issues:

1. New data and analysis included in response to comments should be added to the revised manuscript
2. Additional details provided regarding approach, methodologies and information on the number and type of replicates should be provided in the results section and in figure legends

Specific issues:

1. In response to my previous comment #6, the wording needs to be adjusted. "implying that more genes and pathways are disrupted" is a misleading statement and doesn't add additional information to the sentence.
 2. In response to comment #7, a brief description of the approach should be included in the results section as well
 3. New data provided in response to comment #10 shows t]lack of change in hENCC migration. This is an important observation and should be included and discussed in the main manuscript.
 4. A brief description of the analysis provided in response to comment #12 should be added to the results section
 5. In response to comment #15, the number of replicates, whether they were biological or technical and the statistic should be added to figure legends. Were these ENCCs aged matched (From the same day of differentiation) across all cell lines? were these cultures equally pure? these are important considerations given the differences in efficiency of hENCC induction and the dynamic nature of how these cells progress through differentiation.
-
2. In response to comment #7, a brief description of the approach should be included in the results section as well
 3. New data provided in response to comment #10 shows t]lack of change in ENCC migration. This is an important observation ad should be noted in the main manuscript.
 4. A brief description of the analysis provided in response to comment #12 should be added to the results section
 5. In response to comment #15, the number of replicates, whether they were biological or technical and the statistic should be added to figure legends. Were these ENCCs aged matched (From the same day of differentiation) across all cell lines? were these cultures equally pure? these are important considerations given the differences in efficiency of hENCC induction and the dynamic nature of how these cells progress through differentiation.

Reviewer #3:

Remarks to the Author:

This paper identifies a gene set dysregulated in HSCR-ENCC lines and HDAC1 as a master regulator of

these genes. The increased transactivation of HDAC1 in HSCR-ENCC is associated with suppressed mitochondrial activity and oxidative phosphorylation, suggesting that the increased transactivation activity of HDAC may cause of HSCR. Another key finding is association of PTBP1 exon 9 inclusion with the disrupted cellular metabolism in HSCR-ENCC derived from the long segment of a colon. Thus, there appears to be a functional link with altered energy metabolism, and Hedgehog signaling mediated fatty acid oxidation and consecutive oxidative phosphorylation has a key role in the extent of neuronal differentiation capacity for ENCC.

The authors have performed a large set of additional analyses and provided more methodological details. While the number of patient cases and iPSC lines is very small, and fusion of control for several comparisons weakens a bit the reliability of the data and conclusion, the manuscript is of significance in its field compared to established literature.

Reviewer #4:

Remarks to the Author:

In my previous review I was very skeptical about a number of causal claims unsupported by the evidence provided. After revisions, I am just as skeptical as before. A few word choices have been revised here and there. But there remain numerous causal claims, repeated throughout the manuscript, for which no causal data are present.

As one paradigmatic example (there are various others), I was previously concerned about claims that HDAC1 is a "master regulator" of a core set of genes, when no causal data was presented. After revisions, the claim is now that HDAC1 is a "key regulator" and an "overarching regulator." But still no causal evidence is provided.

Reviewer #5:

Remarks to the Author:

While the revision patches some of the shortcomings of the original submission, it fails to appropriately address critical points and most of my original critiques remain. Overall, the manuscript still does not describe a coherent study.

With regard to the Authors' specific responses (numbering is based on the points they used in the rebuttal letter):

1.1 - The authors response does not adequately address this concern beyond what was already apparent from the original ms.

1.2 - In their response, the Authors state that „In all our subsequent analysis, we compared each HSCR-ENCC to the cells derived from the two control lines independently. Only the genes significantly different from that in the cells derived from BOTH control lines were considered as DEGs.“ - This is a critical information that should have been included in the manuscript, and clearly explained how this was technically done. In the ms, the rather undefined “Next, we compared each HSCR-ENCCs with those from the two controls” statement remains unchanged.

1.3 - In their response, the authors state that „These two control iPSC lines included in this study were carefully selected, based on data obtained from a series of functional and transcriptomic analyses.“ However, these selection criteria are not defined in the rebuttal letter nor are documented in the manuscript.

1.4 - The authors did not provide the requested differential gene expression analysis between the controls. Without further understanding of the differences between the two controls, removal of all the

uniquely expressed genes from the controls is just as concerning. In addition, the removed/remaining genes and their numbers in each controls are not identified/documented. In addition, in their response, they write that „Genes in HSCR-ENCCs significantly different from that of the cells from BOTH of these two control lines were considered as true DEGs. This information is available in the text.“ Yet, this information is not apparent in the text and importantly, is missing from the Methods section „Identification of differentially expressed genes and pathway enrichment analysis“.

1.5 - It remains undefined how the two controls were „merged“. Were the individual values averaged or by other method? Either way, this approach departs from the approach the authors suggested in response to 1.4, which suggest that relevant genes should be significantly different from BOTH controls separately. The use of the different approaches is not described and confusing.

2.1 - In their response, the Authors refer to Lang et al., 2019, the citation of which (e.g. journal, pages, PUBMED ID, ...) is not included and the concern remains unresolved. Here, again the Authors state that „each HSCR-ENCC was compared to the merged controls“, which contradicts that „Genes in HSCR-ENCCs significantly different from that of the cells from BOTH of these two control lines were considered as true DEGs.“ - See above.

2.2 - Thank you for providing this information.

2.3 & 2.4 - In their responses, the Authors highlight that TWO controls were used; this was, however, neither questioned nor it provides a resolution to the concerns.

2.5 - The nuclear:cytosolic intensity data could provide support for the Authors argument and should have been incorporated in the manuscript (currently it is only in the rebuttal letter). Further, thank you for including the MO analysis. However, it is puzzling why the VPA data was removed from and the CI994 data not included in the manuscript. Both are only shown now in the rebuttal letter, but should have been included in the manuscript, main or supplementary. In addition, the MO sequences (HDAC1 and PKMs) should be included/documented in the ms.

3.1 - This item separated from the subsequent text did not highlight a concern from my part that needed to be answered.

3.2 - The Authors response somewhat clarifies this issue. Nonetheless it is still unclear if HSCR #1, 17 and 23 were separately compared to controls (how were they merged?) or together (how were they merged?).

3.3 - This response should have been fully incorporated to Methods / AS analysis, instead of describing in the rebuttal letter.

3.4 & 3.5 - It is fine to choose one target for analysis; however, the authors response does not lend further and/or independent support for the presumed, exceptional importance of PKM and OXPHOS among other potentially disease-relevant pathways controlled by PTBP1. (It would have been also OK, if the Authors simply state and describe/discuss an interest in these pathways and their potential modulators.)

3.6 - This is a critical problem; in their rebuttal letter, the Authors provide an explanation, but do not convincingly resolve this problem. Even if experimental solution cannot be found, the problem and their explanation/interpretation should be clearly discussed in the Discussion, not in the rebuttal letter.

3.7 - Thank you for the PKM1/2 experiments. These lend greater credibility for the involvement of PKM1/2 in the found phenotype. However, the Authors now unexplainably removed the basal OCR and max respiration data presented in original Fig. 5 h-j, which provide the basis for these experiments - they should have kept these in, even if the control-MO data is somewhat comparable with the original

data set.

3.8 - In their response, the Authors did not clearly address my questions. Their quote in the rebuttal letter, however, seems to imply that the results were reused from a previous study and the lines were hiPSC-derived (in the cited article they analyzed both mice and human lines). Peculiarly, although their quote in the rebuttal letter includes „hiPSC“, this is omitted from the quoted part in the manuscript. Concerns of adequate descriptions (e.g. reuse of data) and quality controls and how they are related to the rest of the manuscript (HDAC1, splicing, PTPB1) are not addressed.

Authors Response

Point-by-point response to the reviewers' comments:

REVIEWER COMMENTS

Reviewer #1 (Remarks to the Author):

The authors have improved the manuscript by addressing my main concerns. As for claiming Cluster 1 as ENCC, the authors provided transcriptome analysis data in their response letter to reviewer, but they were not included in the revised manuscript. They should include and discuss these data in the revised manuscript.

Ans: We thank the reviewer for the positive comment. We have included this information in our revised manuscript (line 171-173) and supplementary figures 2i&j.

Reviewer #2 (Remarks to the Author):

This revised manuscript by Li et al addresses many of the comments and concerns raised previously but a few issues remain to be resolved:

General issues:

1. New data and analysis included in response to comments should be added to the revised manuscript

Ans: All new data and analyses have been added to our revised manuscript (Supplementary Fig. S2e-i, S6f-g, S7b&c, S9).

2. Additional details provided regarding approach, methodologies and information on the number and type of replicates should be provided in the results section and in figure legends

Ans: *all key information has been added to results section and figure legends (number and type of replicates needed).*

Ans: We have included the information in the figures and figure legends accordingly.

Specific issues:

1. In response to my previous comment #6, the wording needs to be adjusted. "implying that more genes and pathways are disrupted" is a misleading statement and doesn't add additional information to the sentence.

Ans: We have deleted this misleading statement in our revised manuscript (line 206).

2. In response to comment #7, a brief description of the approach should be included in the results section as well

Ans: We have described the rationale and the approach in the result section of the revised manuscript (line 265).

3. *New data provided in response to comment #10 shows lack of change in hENCC migration. This is an important observation and should be included and discussed in the main manuscript.*

Ans: The data have been added in the revised manuscript (lines 517 & 574 and Supplementary Fig. 9).

4. *A brief description of the analysis provided in response to comment #12 should be added to the results section*

Ans: A brief description has been added to our revised manuscript (line 412).

5. *In response to comment #15, the number of replicates, whether they were biological or technical and the statistic should be added to figure legends. Were these ENCCs aged matched (From the same day of differentiation) across all cell lines? were these cultures equally pure? these are important considerations given the differences in efficiency of hENCC induction and the dynamic nature of how these cells progress through differentiation.*

Ans: The information about the number of replicates was added in the figures and figure legends. The information about the statistical analyses was included in the method section (line 975) “The differences among multiple treatment groups were analyzed with a two-sided unpaired Student’s t test or one-way analysis of variance followed by Tukey post-test using GraphPad Prism 9 (GraphPad Software).” The same approach was used for all the analyses with the functional data. The details of statistic methods for the transcriptomic analyses were also included in the corresponding figures.

As stated in the method (line 656 & Supplementary Fig.1), the ENCCs were enriched by FACS with antibodies against NC-specific markers (HNK-1 and p75^{NTR}) before used in all the assays. We have always added “FACS-enriched” in line 140 to clarify this point in our revised manuscript.

Reviewer #3 (Remarks to the Author):

This paper identifies a gene set dysregulated in HSCR-ENCC lines and HDAC1 as a master regulator of these genes. The increased transactivation of HDAC1 in HSCR-ENCC is associated with suppressed mitochondrial activity and oxidative phosphorylation, suggesting that the increased transactivation activity of HDAC may cause of HSCR. Another key finding is association of PTBP1 exon 9 inclusion with the disrupted cellular metabolism in HSCR-ENCC derived from the long segment of a colon. Thus, there appears to be a functional link with altered energy metabolism, and Hedgehog signaling mediated fatty acid oxidation and consecutive oxidative phosphorylation has a key role in the extent of neuronal differentiation capacity for ENCC.

The authors have performed a large set of additional analyses and provided more methodological details. While the number of patient cases and iPSC lines is very small, and fusion of control for several comparisons weakens a bit the reliability of the data and conclusion, the manuscript is of significance in its field compared to established literature.

Ans: We thank the reviewer for the positive comment.

Please note that HSCR is a rare disease with an incidence of one in 5000 newborns where 80% of them are S-HSCR and many of L-HSCR cases are caused by the coding region mutation of a single gene with a defined pathway. In this study, we have analyzed 7 HSCR-iPSC lines including 2 L-HSCR and 1 TCA, where there is no known coding region mutation in all the S-HSCR and L-HSCR lines. Under this stringent selection criterion, we believe that we have included the key cases (patient without known genetic cause) and covered the full disease spectrum (S-HSCR, intermediate and L-HSCR).

Reviewer #4 (Remarks to the Author):

In my previous review I was very skeptical about a number of causal claims unsupported by the evidence provided. After revisions, I am just as skeptical as before. A few word choices have been revised here and there. But there remain numerous causal claims, repeated throughout the manuscript, for which no causal data are present.

As one paradigmatic example (there are various others), I was previously concerned about claims that HDAC1 is a "master regulator" of a core set of genes, when no causal data was presented. After revisions, the claim is now that HDAC1 is a "key regulator" and an "overarching regulator." But still no causal evidence is provided.

Ans: Our study started from the unbiased and global bioinformatic analyses, we identified a gene set of 118 genes **commonly dysregulated** in all HSCR-ENCCs and predicted HDAC1 as a key upstream regulator across various patient groups that are highly associated with the HSCR pathology. The bioinformatic analytic method (described in "Differential co-expression analysis" section in methodology) that we adopted to predict HDAC1 as a key regulator has been widely applied in single-cell RNA-seq data analysis. It's a variety of the algorithms in SCENIC (Aibar et al., 2017) which has been cited for ~2,000 times. We have employed multiple approaches to support the roles of HDAC1 across different patient groups, and there are multiple lines of evidence suggesting that HDAC1 is a key regulator:

1. Experimental approaches:

We did provide experimental data in our study. For examples, we used HDAC1 morpholinos, HDAC1 inhibitors (VPA, CI994), FAO pathway agonist (GW9508) and PKM1/2 morpholinos to reveal the causal effects and potential mechanisms of regulations of these key regulators and pathways. All these data suggest that HDAC1, OXPHOS and RNA splicing (on *PKM*) affect the differentiation of the control and HSCR-ENCCs.

2. Evidence from the bioinformatic study through integrating the scRNAseq data with the genetic data:

In this study, we have identified a common gene set which represented the shared molecular mechanisms underlying S- and L-HSCR, with 118 genes widely distributed in a series of functional pathways involved in NC development, migration, Notch signaling, neurogenesis, axonogenesis, synapse organization, RNA splicing, energy metabolism, and DNA/protein modification on (enrichment at FDR < 0.05). We also found that the 93 S-HSCR-unique DEGs had strong interactions with the common gene set or independently integrated into the core pathways (Notch signaling, migration, neurogenesis, and

axonogenesis). When we incorporated the 117 HSCR candidate genes previously identified in various genetic screens (Garcia-Barcelo, et al, 2009, Gui, et al, 2017, Jiang, Q. et al, 2015) into this PPI network. A total of 66 of these genes were significantly enriched, with strong interactions with the HSCR-DEGs in 6 core pathways (OR=24.4, χ^2 P-value=4.73 \times 10⁻²²). Integration of data from the independent genetic screens further suggest the biological relevance of the DEGs identified from the current study.

Interactions between HSCR associated DEGs and HSCR candidate genes identified in various genetic screens. Colored by type of DEG sets. HSCR candidate genes are labelled in red. The common DEGs with 118 genes widely distributed in a series of functional pathways involved in NC development, migration, Notch signaling, neurogenesis, axonogenesis, synapse organization, RNA splicing, energy metabolism, and DNA/protein modification on (enrichment at FDR < 0.05). A total of 66 of HSCR candidate genes were significantly enriched, with strong interactions with the HSCR-DEGs in 6 core pathways (OR=24.4, χ^2 P-value=4.73 \times 10⁻²²).

Please note that most of these HSCR related genes (318 of 346) in the figure, including RNA splicing factors and OXPHOS pathway genes, show strong interaction (experimentally_determined_interaction score > 0.1) with HDAC1 based on STRING database, they would be functionally or physically interacting with HDAC1 (empirically demonstrated). Therefore, it is conceivable that HDAC1 is implicated in many of these biological pathways and interacting with many genes associated with the HSCR pathogenesis.

3. Based on the independent published data:

Our data indicated that many HDAC1 targets are implicated in many biological processes that are interrupted in HSCR. For instance, RNA splicing (fig. 2g): it has been reported that the activity of histone deacetylases (HDACs) influences splice site selection. HDAC inhibition induced histone H4 acetylation and increased RNA Polymerase II (Pol II) processivity along

an alternatively spliced element (Hnilicová et al., 2011). A recent review has summarized that HDACs interact with spliceosomal and ribonucleoprotein complexes, actively control the acetylation states of splicing-associated histone marks and splicing factors, and thereby unexpectedly could modulate splicing (Rahhal and Seto, 2019). Similarly, it has been shown that Hdac1/Hdac2 promotes the transition to OXPHOS by enforcing transcriptional fidelity of metabolic gene programs in mouse cardiac development (Milstone et al., 2020).

In the current study, we aimed to present an overview of the biological pathways associated with HSCR pathogenesis. Even though our data suggest that HDAC1 is a key regulator underlying various HSCR-associated pathways, we did not aim to focus on how HDAC1 interacts with the other pathways. To delineate the roles of HDAC1 in various cellular processes, a more comprehensive study is required to carefully analyze the epigenomes of the cells, but it is out of the scope of the current study.

References:

- Garcia-Barcelo, M. M. et al. Genome-wide association study identifies NRG1 as a susceptibility locus for Hirschsprung's disease. *Proc Natl Acad Sci U S A* 106, 2694-2699 (2009). <https://doi.org:10.1073/pnas.0809630105>
- Gui, H. et al. Whole exome sequencing coupled with unbiased functional analysis reveals new Hirschsprung disease genes. *Genome Biology* 18, 48 (2017). <https://doi.org:10.1186/s13059-017-1174-6>
- Jiang, Q. et al. Functional loss of semaphorin 3C and/or semaphorin 3D and their epistatic interaction with ret are critical to Hirschsprung disease liability. *Am J Hum Genet* 96, 581-596 (2015). <https://doi.org:10.1016/j.ajhg.2015.02.014>
- Aibar, S., González-Blas, C., Moerman, T. et al. SCENIC: single-cell regulatory network inference and clustering. *Nat Methods* 14, 1083–1086 (2017).
- Hnilicová, J., Hozeifi, S., Dušková, E., Icha, J., Tománková, T., and Staněk, D. (2011). Histone deacetylase activity modulates alternative splicing. *PLoS One* 6, e16727. [10.1371/journal.pone.0016727](https://doi.org:10.1371/journal.pone.0016727).
- Rahhal, R., and Seto, E. (2019). Emerging roles of histone modifications and HDACs in RNA splicing. *Nucleic Acids Research* 47, 4911-4926. [10.1093/nar/gkz292](https://doi.org:10.1093/nar/gkz292).
- Milstone, Z.J., Saheera, S., Bourke, L.M., Shpilka, T., Haynes, C.M., and Trivedi, C.M. (2020). Histone deacetylases 1 and 2 silence cryptic transcription to promote mitochondrial function during cardiogenesis. *Sci Adv* 6, eaax5150. [10.1126/sciadv.aax5150](https://doi.org:10.1126/sciadv.aax5150).

Reviewer #5 (Remarks to the Author):

While the revision patches some of the shortcomings of the original submission, it fails to appropriately address critical points and most of my original critiques remain. Overall, the manuscript still does not describe a coherent study.

With regard to the Authors' specific responses (numbering is based on the points they used in the rebuttal letter):

1.1 - The authors response does not adequately address this concern beyond what was already apparent from the original ms.

Ans: We believe that we didn't interpret your question correctly last time. We would like to make our point clear. In Fig. 1h, we aimed to show the 5 main cell types/states found in both the control and the patient derived ENCCs. Please note that **both the control and the patient cells are ENCCs**, they were enriched by FACS expressing similar ENCC markers. Definitely, they should look similar to each other in the tSNE plot where only different cell types/states (Cluster 2-5) will show the bigger differences (shown as the distinct clusters). Even though the Cluster 1 of the control and the patient cells were clustered together, they were transcriptionally different with differential expressed genes (DEGs) when we analyzed them at a higher resolution (please refer to the Supplementary Fig 2k, where the two controls are located close to each other and at the bottom of the Cluster 1 and separated from the other patient cells). In this study, we needed to compare the gene expression profiles of the same type of cells (ENCCs) at the similar cell state, not comparing ENCCs with their neuronal or glial derivatives. It is why we focused on Cluster 1 and Fig 1h was used to demonstrate that we are comparing ENCCs at the same cell state (rather than comparing apples with oranges).

1.2 - In their response, the Authors state that „In all our subsequent analysis, we compared each HSCR-ENCC to the cells derived from the two control lines independently. Only the genes significantly different from that in the cells derived from BOTH control lines were considered as DEGs.“ - This is a critical information that should have been included in the manuscript, and clearly explained how this was technically done. In the ms, the rather undefined “Next, we compared each HSCR-ENCCs with those from the two controls” statement remains unchanged.

Ans: We have added the description of the analytic method in the methodology section (Identification of differentially expressed genes and pathway enrichment analysis, line 884). “Since two independent controls were included for the analyses, each HSCR-ENCC was compared to each control separately. Then, we identified the consensus significant DEGs (overlapped DEGs) of each HSCR-ENCC. For example, HSCR#1 consensus DEGs were the overlapped DEGs of HSCR#1/IMR90 and HSCR#1/UE02302. This consensus DE analysis would be able to identify more reliable HSCR DEGs compared to DEGs identified merely based on one control. The two controls were very similar from each other and only 583 DEGs were identified between the two controls. For convenience, two controls were merged for comparison (calculate *P*-value) in several visualizations (Fig. 2g-j, 3e, 5a-g).”

1.3 - In their response, the authors state that „These two control iPSC lines included in this study were carefully selected, based on data obtained from a series of functional and transcriptomic analyses.“ However, these selection criteria are not defined in the rebuttal letter nor are documented in the manuscript.

Ans: (1) We have provided the data from the migration and *in vitro* differentiation assays of ENCCs derived from the two control lines (Figure 1d-g) where the control cells show the high capacity to migrate and differentiate, fulfilling the two major characteristics of ENCCs. Importantly, these assays also showed that the cells derived from the two control lines are highly comparable. (2) In addition, transcriptomic analyses also showed that these two control iPSC lines exhibit a similar molecular profile as found in the other human iPSC and ES lines (Supplementary figure 9). (3) We have also performed additional transcriptomic analyses and compared the ENCCs and their neuronal derivatives derived from these control lines with mouse enteric, sensory and sympathetic neurons and demonstrated that these control ENCCs and their neuronal derivatives possess the molecular signatures highly similar to the mouse bipotent ENCCs and enteric neurons, respectively (Supplementary Figure 2d-2i). All these data support the use of these two control lines in this study and all the data are available in our manuscript.

1.4 - The authors did not provide the requested differential gene expression analysis between the controls. Without further understanding of the differences between the two controls, removal of all the uniquely expressed genes from the controls is just as concerning. In addition, the removed/remaining genes and their numbers in each controls are not identified/documentated. In addition, in their response, they write that „Genes in HSCR-ENCCs significantly different from that of the cells from BOTH of these two control lines were considered as true DEGs. This information is available in the text.“ Yet, this information is not apparent in the text and importantly, is missing from the Methods section „Identification of differentially expressed genes and pathway enrichment analysis“.

Ans: We have added a sentence (line 202) and the descriptions of the analytic method in the methodology section (Identification of differentially expressed genes and pathway enrichment analysis, line 884) to clarify how the DEGs were defined.

We have performed DE analysis and identified 583 DEGs between the two controls, they are enriched in the pathways implicated in housekeeping pathways, such as cytoplasmic translation, ribosome biogenesis, rRNA processing, etc. We did not see any significant enrichment of these pathways in the patient cells. More importantly, according to our functional data, we did not see differentiation and migration defect in the two controls, further suggesting that these DEGs are not related to the HSCR-associated phenotypes. Therefore, we excluded them for the subsequent analyses. (please also see below (our response to 2.1) for the additional information).

1.5 - It remains undefined how the two controls were „merged“. Were the individual values averaged or by other method? Either way, this approach departs from the approach the authors suggested in response to 1.4, which suggest that relevant genes should be significantly different from BOTH controls separately. The use of the different approaches is not described and confusing.

Ans: For DEG analysis, we did NOT merge control cells, but performed the consensus DE analysis. This helps us to identify the most reliable consensus DEGs. Since we did not find obvious difference between the two controls on neurogenesis, proliferation and RNA splicing, etc. in the downstream visualization (Fig. 2g-j, 3e, 5a-g), we merged the two controls to simplify the visualization. “merged cells” is not a new analysis, it’s just for visualization convenience. “merged cells” is a simple combination of cells to calculate the *P*-value(s).

2.1 - In their response, the Authors refer to Lang et al., 2019, the citation of which (e.g. journal, pages, PUBMED ID, ...) is not included and the concern remains unresolved. Here, again the Authors state that „each HSCR-ENCC was compared to the merged controls“, which contradicts that „Genes in HSCR-ENCCs significantly different from that of the cells from BOTH of these two control lines were considered as true DEGs.“ - See above.

Ans: We apologize for not including the details of the reference. Please see below for the details.

We would like to clarify that the DEGs were obtained by comparing each HSCR-ENCC with each control individually, only the genes significantly different from the two controls were considered as DEGs. Given that the two controls are very similar, we combined them for the pathway-based analyses simply by combining all the cells (dots) together (Fig. 2g-j, 3e, 5g). Consistently, significant differences were observed between the controls (merged controls) and the samples (merged samples). These observations further support that (1) the two controls are very similar; (2) the differences among the two controls are not related to HSCR associated phenotypes. Similar approach was used for the samples (3 S-HSCR #5, #10 & #20, and 2 L-HSCR, #17 & #23). As illustrated in Fig. 2c, cells on PC1 in each group exhibited a similar transcription profile. Thus, we merged them for visualization to highlight the uniqueness of each group. Importantly, it is a very useful approach to show the differences are not due to the individual variations among lines, but the **common differences found among groups**. The variations among samples in each group were reflected by the *P*-values obtained from the statistical analyses. In the text and the figures, we have clearly listed out the DEGs found in each patient line and the merged data were presented as groups. In all cases, ALL the samples (2 control and 6 HSCR) were used for the analyses.

Reference:

- Lang C, Campbell KR, Ryan BJ, Carling P, Attar M, Vowles J, Perestenko OV, Bowden R, Baig F, Kasten M, Hu MT, Cowley SA, Webber C, Wade-Martins R. Single-Cell Sequencing of iPSC-Dopamine Neurons Reconstructs Disease Progression and Identifies HDAC4 as a Regulator of Parkinson Cell Phenotypes. *Cell Stem Cell*. 2019 Jan 3;24(1):93-106.e6. doi: 10.1016/j.stem.2018.10.023. Epub 2018 Nov 29. PMID: 30503143; PMCID: PMC6327112.

2.2 - Thank you for providing this information.

2.3 & 2.4 - In their responses, the Authors highlight that TWO controls were used; this was, however, neither questioned nor it provides a resolution to the concerns.

Ans: The control cells always can differentiate robustly. The reviewer agree that there was a small difference in one control but not the other (IMR90) after the VPA treatment. As shown in the figure, IMR90 exhibited 60-70% of cells expressing NF even without treatment, which is a very high percentage (usually 20% or below in the patient cells). It is not surprised that the percentage cannot be pushed up further after addition of VPA. On the other hand, UE02302 showed a lower percentage (around 50%) of cells expressing NF, and a slight increase was observed (to 70-80%) upon VPA treatment. In both control lines, the maximum yield of NF+ neurons were 80%, highly

comparable. We said the two control lines are very similar, but NOT identical (with different genetic background). This is the reason why two controls and multiple samples were included in the study. Whenever we are handling human samples with different genetic backgrounds, we can only just focus on the common phenotypes and pathways found in each group, not the subtle variations among individual lines within the groups. In addition, the fold differences between the treated and untreated cells in the patient group are much dramatic when compared to the difference found in this control.

2.5 - The nuclear:cytosolic intensity data could provide support for the Authors argument and should have been incorporated in the manuscript (currently it is only in the rebuttal letter). Further, thank you for including the MO analysis. However, it is puzzling why the VPA data was removed from and the CI994 data not included in the manuscript. Both are only shown now in the rebuttal letter, but should have been included in the manuscript, main or supplementary. In addition, the MO sequences (HDAC1 and PKMs) should be included/documentated in the ms.

Ans: The nuclear:cytosolic intensity data was included in Fig. S6b & S6d.

MO, VPA and CI994 are referring to the same point and the data are redundant. The main figures are busy enough, but as requested by the reviewer, we included the VPA and CI994 data in Fig. S6 f & S6g.

MO sequences (HDAC1 and PKMs) were included in the Supplementary table 3.

3.1 - This item separated from the subsequent text did not highlight a concern from my part that needed to be answered.

Ans: it's good if it is not a concern.

3.2 - The Authors response somewhat clarifies this issue. Nonetheless it is still unclear if HSCR #1, 17 and 23 were separately compared to controls (how were they merged?) or together (how were they merged?).

Ans: We already described the details in the figure legend of Fig. 4. "ENCCs at high disease state (HSCR#1, 17 and 23) were compared to those in low disease states (two controls and 3 S-HSCR-ENCCs) ($P < 0.05$, t-test)". The splicing events and numbers in ENCCs at high disease state (HSCR#1, 17 and 23) were very comparable (please refer to the individual data shown in Fig 4a and 4b), so these samples were merged (pooled) for comparison as stated in the figure legend, simply pooling all the cells from these samples together for comparative analyses. It is just a simple group comparison.

As mentioned above, whenever the samples are labeled as control/HSCR without numbering, they are referring to the pooled samples.

3.3 - This response should have been fully incorporated to Methods / AS analysis, instead of describing in the rebuttal letter.

Ans: The information was already included in method section (Alternative splicing (AS) analysis) in our previous version.

3.4 & 3.5 - *It is fine to choose one target for analysis; however, the authors response does not lend further and/or independent support for the presumed, exceptional importance of PKM and OXPHOS among other potentially disease-relevant pathways controlled by PTBP1. (It would have been also OK, if the Authors simply state and describe/discuss an interest in these pathways and their potential modulators.)*

Ans: The information was available in the manuscript.

1. “Among all AS events, the inclusion of *PTBP1* exon 9 appears to contribute most significantly to the advanced disease state; *PTBP1* was overexpressed in the HSCR, particularly higher in the severe disease cases, with strong enrichment of exon 9” to justify why *PTBP1* exon9 was chosen.
2. “One of the known targets of *PTBP1* is *PKM*, which encodes two key enzymes (pyruvate kinase M [*PKM*] 1 and *PKM2*) involved in glycolysis, and they are responsible for converting phosphoenolpyruvate (PEP) to pyruvate as a substrate for oxidative phosphorylation (OXPHOS).” and “*PKM1* efficiently converts PEP into pyruvate that is preferentially used for OXPHOS, favoring neurogenesis.” to highlight the link between *PTBP1*, *PKM* and OXPHOS and neurogenesis (the major defects found in patient cells).

We also added a description to explain further the underlying meaning of Fig. 5e “In order to reveal the potential regulation between *PTBP1* and *PKM*, we performed differential co-expression analysis. In brief, RNA binding motif of *PTBP1* was obtained from RBP binding motif database (mCrossBase). Then, RNA binding motif enrichment analysis was performed using FIMO tool of MEME suite.” in the revised manuscript lines 411-415. We hope it will help to clarify why *PTBP1*-*PKM1*-OXPHOS pathway was selected.

3.6 - *This is a critical problem; in their rebuttal letter, the Authors provide an explanation, but do not convincingly resolve this problem. Even if experimental solution cannot be found, the problem and their explanation/interpretation should be clearly discussed in the Discussion, not in the rebuttal letter.*

Ans: Please note that we are handling human samples with heterogenous genetic backgrounds and not the mouse with specific knockout of a single gene. The expression analysis here is a correlation study that allows us to identify the potential key pathways/genes and the subsequent functional validation will address their biological functions. *PTBP1* exon 9 is NOT significantly changed in S-HSCR based on our scRNAseq and RT-qPCR analyses. On the other hand, *PKM1* alone is significantly down-regulated in S-HSCR cells based on the RT-qPCR data. It means that the *PTBP1*-*PKM1* pathway is significantly changed in L-HSCR, while ONLY *PKM1* is significantly changed in S-HSCR. The roles of *PKM1* in cellular metabolism and neuronal differentiation of ENCCs in control, S-HSCR and L-HSCR groups were demonstrated in Fig. 5k-o. Therefore, *PKM1* is important in the control and the diseased cells. The down-regulation of *PKM1* is associated with the *PTBP1*-Ex9 in **L-HSCR** (with the support of the scRNAseq and RT-qPCR data), but this correlation was **not seen in S-HSCR**. We believe we should follow this logic to interpret the data, rather than over-interpreting the data to explain why dysregulation of *PTBP1*-Ex9 -*PKM1* was not found in S-HSCR and what other genes are involved in regulation of *PKM1* in S-HSCR, particularly, we did not have any data on this area.

We have revised the statement to “the reduced PKM1 activity in L-HSCR that is associated with the inclusion of *PTBP1* exon 9 likely perturb OXPHOS, leading to the retarded neuronal differentiation of ENCCs.” to prevent confusion. (line 458).

3.7 - Thank you for the PKM1/2 experiments. These lend greater credibility for the involvement of PKM1/2 in the found phenotype. However, the Authors now unexplainably removed the basal OCR and max respiration data presented in original Fig. 5 h-j, which provide the basis for these experiments - they should have kept these in, even if the control-MO data is somewhat comparable with the original data set.

Ans: The data were added back to the revised manuscript.

*3.8 - In their response, the Authors did not clearly address my questions. Their quote in the rebuttal letter, however, seems to imply that the results were reused from a previous study and the lines were hiPSC-derived (in the cited article they analyzed both mice and human lines). Peculiarly, although their quote in the rebuttal letter includes „hiPSC“, this is omitted from the quoted part in the manuscript. Concerns of adequate descriptions (e.g. reuse of data) and quality controls and how they are related to the rest of the manuscript (*HDAC1*, splicing, *PTBP1*) are not addressed.*

Ans: We have clearly mentioned in text why the use of SAG was studied.

Line 461 “Increasing evidence suggests that neuronal differentiation defect of ENCCs is one of the common problems found in all HSCR patients (S/L-HSCR). Therefore, it is conceivable that rescuing the neuronal differentiation of ENCCs represents a possible treatment strategy. Our previous study demonstrated that activation of Hedgehog pathway by addition of a Smoothened agonist (SAG) primes ENCCs to a more advanced cell state and favors the neuronal lineage differentiation (*Lau et al, 2019*, reference #8 in text)”. It is true for both hiPSC derived ENCCs and mouse ENCCs as stated in the cited reference.

Therefore, we based on our previous observations to further understand how SAG primes ENCCs to the neuronal lineage by analyzing the transcriptomics of ENCCs and SAG-primed ENCCs (Fig. 6 a & 6b). We found that (1) the metabolic pathway genes are consistently up-regulated in the primed ENCCs as shown in their transcriptomes and (2) by the functional assays. The enhanced metabolic pathways, particularly OXPHOS can boost up the differentiation capacity of ENCCs, primed ENCCs are more competent to make neurons (fig.6). Therefore, the use of SAG to rescue the differentiation defects of HSCR-ENCCs was tested (fig. 7).

To understand the link between other findings, please go through the following key findings in order:

- (1) the reduced cellular metabolism (OXPHOS) is a common problem in both S-HSCR and L-HSCR (Fig 2, 5h-5o)
- (2) *PTBP1*-*PKM1* is dysregulated in L-HSCR, *PKM1* is down-regulated in S-HSCR and the upregulation of *PKM1* improves the differentiation capacity of both S-HSCR and L-HSCR ENCCs via boosting up the OXPHOS (Fig. 2 & 5).
- (3) SAG primed ENCCs to neurogenic lineage via inducing OXPHOS and favors the neuronal lineage differentiation.

Therefore, we tested the potential therapeutic value of SAG to “rescue” the differentiation defects of the patient cells (Fig. 7) and showed that SAG enhances OXPHOS and primes ENCCs to neurogenic lineage differentiation in control and patient groups.

As for HDAC1, our data indicate that many HDAC1 targets are implicated in RNA splicing (fig. 2g). Concordantly, it has been reported that the activity of histone deacetylases (HDACs) influences splice site selection. HDAC inhibition induced histone H4 acetylation and increased RNA Polymerase II (Pol II) processivity along an alternatively spliced element (Hnilicová, J. et al. 2011). A recent review has summarized that HDACs interact with spliceosomal and ribonucleoprotein complexes, actively control the acetylation states of splicing-associated histone marks and splicing factors, and thereby unexpectedly could modulate splicing (Rahhal & Seto, 2019). Thus, we consider HDAC1 is an upstream regulator and its aberrant activity causes the disease partially via perturbing AS/OXPHOS, leading to the neuronal differentiation defects as seen in HSCR-ENCCs. SAG, on the other hand, can induce OXPHOS and favor the neuronal differentiation. Through “correcting” the downstream problem (OXPHOS), the neuronal differentiation capacity of HSCR-ENCCs can be improved. In other words, through mapping the cellular processes along the same pathway, we discovered the potential therapeutic targets.

We added a schematic diagram (Fig.8) to summarize these points.

References:

1. Lau, S. T. et al. Activation of Hedgehog Signaling Promotes Development of Mouse and Human Enteric Neural Crest Cells, Based on Single-Cell Transcriptome Analyses. *Gastroenterology* 157, 1556-1571 e1555 (2019). <https://doi.org:10.1053/j.gastro.2019.08.019>
2. Hnilicová, J. et al. Histone deacetylase activity modulates alternative splicing. *PLoS One* 6, e16727 (2011). <https://doi.org:10.1371/journal.pone.0016727>
3. Rahhal, R. & Seto, E. Emerging roles of histone modifications and HDACs in RNA splicing. *Nucleic Acids Research* 47, 4911-4926 (2019). <https://doi.org:10.1093/nar/gkz292>

Reviewers' Comments:

Reviewer #2:

Remarks to the Author:

The authors have substantially improved the manuscript and have adequately addressed my concerns. There are still instances where the explanation on methodology and interpretations are provided only in the rebuttal letter. It is important that all these information are provided in the manuscript text and methods where possible.

Reviewer #4:

Remarks to the Author:

In the first two rounds of review, I requested that the authors either provide causal evidence for many of their causal claims, or remove the causal claims altogether. After multiple rounds of revision, the authors have not done either.

As one paradigmatic example, the manuscript still claims that HDAC1 is a key and overarching regulator of a core set of genes.

In their rebuttal the authors report that the tool for correlative inference that they use has been cited 2,000 times. This is an impressive citation count, but does not constitute causal evidence. The authors also state that they use morpholinos, etc., which is true, but they do not use them address the claim(s) in question.

Therefore, the manuscript still makes numerous causal claims (including the above) that are not supported by causal evidence.

Reviewer #2 (Remarks to the Author):

The authors have substantially improved the manuscript and have adequately addressed my concerns. There are still instances where the explanation on methodology and interpretations are provided only in the rebuttal letter. It is important that all these information are provided in the manuscript text and methods where possible.

Ans: All the key information has been included in the manuscript, method and Supplementary data/figures.

Reviewer #4 (Remarks to the Author):

In the first two rounds of review, I requested that the authors either provide causal evidence for many of their causal claims, or remove the causal claims altogether. After multiple rounds of revision, the authors have not done either.

As one paradigmatic example, the manuscript still claims that HDAC1 is a key and overarching regulator of a core set of genes.

In their rebuttal the authors report that the tool for correlative inference that they use has been cited 2,000 times. This is an impressive citation count, but does not constitute causal evidence. The authors also state that they use morpholinos, etc., which is true, but they do not use them address the claim(s) in question.

Therefore, the manuscript still makes numerous causal claims (including the above) that are not supported by causal evidence.

Ans: Amendments have made according to Editor's suggestions.